# On the Stability-Plasticity Dilemma in Continual Meta-Learning: Theory and Algorithm

**Qi Chen**[*]
Laval University

**Changjian Shui**
McGill University

**Ligong Han**
Rutgers University

**Mario Marchand**
Laval University

## Abstract

We focus on Continual Meta-Learning (CML), which targets accumulating and exploiting meta-knowledge on a sequence of non-i.i.d. tasks. The primary challenge is to strike a balance between stability and plasticity, where a model should be stable to avoid catastrophic forgetting in previous tasks and plastic to learn generalizable concepts from new tasks. To address this, we formulate the CML objective as controlling the average excess risk upper bound of the task sequence, which reflects the trade-off between forgetting and generalization. Based on the objective, we introduce a unified theoretical framework for CML in both static and shifting environments, providing guarantees for various task-specific learning algorithms. Moreover, we first present a rigorous analysis of a bi-level trade-off in shifting environments. To approach the optimal trade-off, we propose a novel algorithm that dynamically adjusts the meta-parameter and its learning rate w.r.t environment change. Empirical evaluations on synthetic and real datasets illustrate the effectiveness of the proposed theory and algorithm.

## 1 Introduction

An essential goal in real-time intelligent systems is to balance stability (preserve past knowledge; minimize catastrophic forgetting [1]) and plasticity (rapidly learn from new experiences; generalize quickly [2]). For addressing this dilemma, a promising direction is to incorporate meta-learning [3, 4] with Continual Learning (CL) [5–7], which constitutes the Continual Meta-Learning (CML) [8–10] problem. Specifically, CML sequentially learns a meta-model from few-shot tasks as the common prior. Then, for new tasks, it quickly adapts task-specific models with this prior. CML effectively enables balancing the *task-level* trade-off between learning new tasks and retaining previously acquired knowledge since the performance on previous tasks can be recovered with few additional samples.

Despite being intuitive and technically sound, most related works in CML have focused on improving empirical performance. There is still much room for exploration in the theoretical aspects. In particular, there is a lack of rigorous understanding on *(1) which factors are important for stability and plasticity, (2) how to address the dilemma effectively by considering these factors.*

To theoretically understand these questions, we need to characterize the sequential task-generating process. In fact, many CML approaches are based on the implicit assumption that the non-i.i.d. tasks are originated from a static task environment $\tau$, where $\mu_t \sim \tau, \forall t$. In addition, recent works reveal that catastrophic forgetting often occurs when transferring a meta-model to a new environment [11, 12] and start trying to empirically address the forgetting in shifting environments, such as [7, 13–16]. We take this case into consideration and formally define a more general CML setting, where at each time $t$ the task $\mu_t$ is generated from a possibly shifting environment $\tau_t$ with $\mu_t \sim \tau_t$ (the environment is

---

[*]Correspondence to: qi.chen.1@ulaval.ca

37th Conference on Neural Information Processing Systems (NeurIPS 2023).

static if $\tau_t = \tau, \forall t$). We found a *meta-level* trade-off exists in shifting environments, which further induces difficulty in controlling the task-level learning-forgetting trade-off, as discussed in Sec. 4.2.

In this paper, we formally study the *bi-level* (task- and meta-) learning-forgetting trade-off in CML to fill the theoretical gap. Compared with previous works, our contribution highlights are as follows:

**Unified theoretical framework** We first introduce a novel and unified theoretical framework for CML in both static and shifting task environments. For each task, the excess risk reflects the true model performance but is intractable. Therefore, we derive a unified form of the excess risk upper bound for various base learners, which contains optimization error with the generalization gap and could be estimated from observations. Based on these, we propose to control the Average Excess Risk (AER) upper bound of all the tasks as the learning objective in CML (shown in Sec. 3). This upper bound can be further decomposed as the optimal trade-off and the algorithmic regret.

**Understanding the bi-level learning-forgetting trade-off** We discuss the bi-level trade-off in shifting environments with an illustrative example in Sec. 4.1. We theoretically identify in Theorem 5.1 that: (1) Inside each environment, the task-level trade-off is affected by the task similarities, which reflects on the slot diameters and variances. (2) The path length and changing points representing environment similarities and non-stationarity dominate the meta-level trade-off. (3) Task and sample numbers are important to generalization. The optimal trade-off is achieved with a minimal upper bound of AER, so it's helpful to consider these factors for optimizing the meta-parameter sequence.

**Theoretically grounded algorithm** We propose a novel algorithm (Algo. 1) for addressing the bi-level trade-off in shifting environments, which dynamically adjusts both the meta-parameter and its learning rate when an environment change is detected. Hence, the proposed algorithm can implicitly control the AER by minimizing the dynamic regret to approach the optimal trade-off. We derive a general bound for the proposed algorithm in Theorem 5.1. Improved complexity factors and rates theoretically demonstrate the validity of the proposed algorithm for balancing the trade-off, as illustrated in Theorem 6.1 and 6.2. Furthermore, empirical results on various datasets show improvements in estimated bounds and superior performance compared to baselines, demonstrating that the proposed algorithm can incrementally learn from new environments while maintaining good performance on the previous ones, i.e., a well-balanced trade-off.

## 2 Related work

Due to the page limit, we only briefly discuss the most related works in this section. A more detailed discussion and comprehensive investigation of various settings are provided in Appendix H.

**Continual learning** Traditional CL approaches mainly focus on addressing *catastrophic forgetting*. Some methods use *regularization* techniques, such as Elastic Weight Consolidation (EWC) [17] and Synaptic Intelligence (SI) [18], that aim to protect important parameters of a neural network from being modified during training on new tasks. Other methods like *replay-based* approach [19, 20] store or replay previous task examples to aid in preserving knowledge of old tasks. Another line of work is *parameter isolation* or *dynamic architecture* methods, which involve creating new network pathways or modules for each new task [21, 22]. Riemer et al. [7] first conceptualize the learning forgetting trade-off in CL as temporally maximizing transfer and minimizing interference by enforcing gradient alignment across examples, which is implemented using experience replay and Reptile [23] to approximate the gradient alignment objective with a first-order Taylor expansion.

**Meta-learning** Meta-learning aims for *quick generalization*, allowing for efficient learning of new tasks with limited data. The foundational framework of *statistical meta-learning* by Baxter [24] assumes tasks are i.i.d. sampled from a task environment. Amit and Meir [25] proposed PAC Bayes bounds with a joint training algorithm, which is not straightforwardly extendable to sequential learning. Chen et al. [26] derived information-theoretic bounds for MAML-like algorithms suitable for sequential few-shot learning. These works assume i.i.d. conditions for both tasks and in-task data. To break the assumption, *online meta-learning* introduces two main approaches for learning non-i.i.d. sequential tasks. *Batch-Within-Online* (BWO)[9, 27] methods, similar to CML, learn each task under a statistical batch setting. However, the theoretical analysis is limited, and none addresses shifting task environments. *Online-Within-Online* (OWO) [28–31] methods employ online algorithms as both base and meta learners. Khodak et al. [31] were the first to consider shifting environments for OWO, providing average regret bounds for gradient-based learners.

In contrast, we consider various statistical batch algorithms in addition to the gradient-based ones as base learners. We further propose a fine-grained algorithm w.r.t environment change that has an improved rate over the bound in [31] (although not directly comparable) for gradient-based base learners, as discussed in Sec. 6.2. This improvement implies a better learning-forgetting trade-off not considered in previous meta-learning literature.

**Continual meta-learning** FTML [9] extends MAML [4] to sequential learning using Follow The Leader (FTL) as the meta-learner, which requires storing all the previous tasks. MOCA [32] incorporates Bayesian Online Changing-point Detection (BOCD) to identify unknown task boundaries during the continual meta-learning process. The above methods can address the task-level trade-off under a static task environment. However, a meta-level trade-off exists when facing shifting environments. He et al. [8] combines MAML with Bayes Gradient Descent (BGD), slowly updating meta-parameters of small variance to stabilize the learning from non-i.i.d. sequential tasks. Caccia et al. [13] address the environment shift with a soft modulation on meta-updates with the empirical loss. [14], [15], and [16] incorporate additional memory with different environment shift detection methods to address the meta-level trade-off. [14] and [15] model the meta-parameter distribution with a Dirichlet Process Mixture Model (DPMM) and Dynamic Gaussian Mixture Model (DGMM), respectively, which can detect new environments and store the corresponding $K$ meta-parameters with $\mathcal{O}(K)$ memory. [16] grows $K$ subnets by detecting the environment shift with BOCD and stores $M$ tasks from previous environments, thus having a memory complexity of $\mathcal{O}(K + M)$.

Instead of the memory-based approaches, we focus on the more challenging fully online experimental setting first proposed by Caccia et al. [13]. Our algorithm consumes $\mathcal{O}(1)$ memory and better balances the meta-level trade-off in the online setting. In addition, we consider different levels of non-stationarity and more environment shifts compared to previous works.

**Theory on the stability-plasticity dilemma** Raghavan and Balaprakash [33] explicitly study the task-level trade-off in continual learning by formulating the problem as a two-player sequential game and prove the existence of a balance point for each task. However, the cost function is based on empirical loss without considering generalization on unseen data. Based on NTK [34] regime, the generalization and forgetting property of orthogonal gradient descent [35] are separately studied in [36] and [37] for continual learning. *To the best of our knowledge, we are the first theoretical work studying the stability-plasticity dilemma in CML.* We also first provide a theoretical framework for CML in static and shifting environments, offering a formal understanding of the bi-level trade-off.

## 3 Problem setup

Let us consider a sequence of different task distributions $\{\mu_t\}_{t=1}^{T}$ defined on the same example space $\mathcal{Z} = \mathcal{X} \times \mathcal{Y}$ with $T \to \infty$. Specifically, at time $t$, the task $\mu_t$ is generated from a possibly shifting environment $\tau_t$ with $\mu_t \sim \tau_t$, which is static if $\tau_t = \tau, \forall t$. Then a dataset of $m_t$ examples $S_t = \{Z_i\}_{i=1}^{m_t}$ are i.i.d. sampled from $\mu_t$, where $S_t \sim \mu_t^{m_t}$. In CML, we have two kinds of parameters: model parameter and meta-parameter, which are learned through a base learner and a meta learner, respectively. A visual representation of the CML process is provided in Fig. 1. Let the meta-parameters be defined on the same support $\mathcal{U}$ for all the tasks. Additionally, we assume that the same parametric hypothesis space $\mathcal{W}$ is used across all tasks with non-negative bounded loss function $\ell : \mathcal{W} \times \mathcal{Z} \to [0, 1]$. Given any model parameter $w \in \mathcal{W}$, the true risk and empirical risk of $t$-th task are defined as $\mathcal{L}_{\mu_t}(w) \stackrel{\text{def}}{=} \mathbb{E}_{Z \sim \mu_t} \ell(w, Z)$ and $\mathcal{L}_{S_t}(w) \stackrel{\text{def}}{=} \frac{1}{m_t} \sum_{i=1}^{m_t} \ell(w, Z_i)$. To motivate the CML setting mentioned above, we also provide an analysis of how to apply CML to real-world examples like online recommendation systems in Appendix G.2.

### 3.1 Base learner

At time $t$, base learner $\mathcal{A}$ takes the task data $S_t$ and the meta-parameter $u_t \in \mathcal{U}$ that represents the prior knowledge learned from previous tasks as the input then outputs the model parameter $W_t = \mathcal{A}(u_t, S_t)$. Specifically, $\mathcal{A}$ is characterized by a conditional distribution[2] $P_{W_t|S_t,u_t}$ as in [38, 39]. We further define the corresponding (expected) **excess risk** on task $t$ as the expected gap between the true risk of the learned hypothesis $W_t$ and the optimal true risk:

---

[2]It will be a delta function if $\mathcal{A}(u_t, S_t)$ is deterministic.

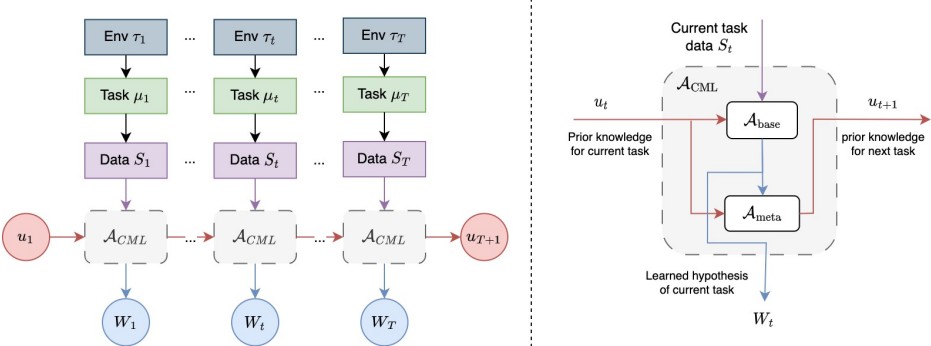

Figure 1: Illustration of Continual Meta-Learning (CML) process. At each time $t$, the CML algorithm $\mathcal{A}_{\text{CML}}$ (composed of the meta learner and the base learner) takes the current task data $S_t$ and the meta-parameter (prior knowledge) $u_t$ learned from previous tasks as input. Then, it outputs the learned hypothesis $W_t$ of the current task and the updated meta-parameter $u_{t+1}$ for the next task.

$$R_{\text{excess}}(\mathcal{A}, u_t) \overset{\text{def}}{=} \mathbb{E}_{S_t}\mathbb{E}_{W_t \sim P_{W_t|S_t,u_t}} \left[ \mathcal{L}_{\mu_t}(W_t) - \mathcal{L}_{\mu_t}(w_t^*) \right], w_t^* = \arg\min_{w \in \mathcal{W}} \mathcal{L}_{\mu_t}(w).$$

The excess risk of $\mu_t$ is thus explicitly determined by the meta-parameter $u_t$ and the base learner, where $u_t$ is provided by experts or random guesses in single-task learning. Although reflecting the true model performance, the excess risk is intractable due to the unknown distribution $\mu_t$.

The following theorem gives a **unified form of excess risk upper bound** that applies to common base learners such as Stochastic Gradient Descent (SGD), Stochastic Gradient Langevin Dynamics (SGLD), Regularized Loss Minimization (RLM), and Gibbs algorithm. In contrast to the upper bounds that contain the empirical risk, it avoids the task data access for the meta-learner and can thus be used to design new CML algorithms. See detailed proof in Appendix B.5.

**Theorem 3.1.** *For any $t \in [T]$, assume that $\mathcal{L}_{\mu_t}(\cdot)$ has $\alpha$-quadratic growth (see Definition A.2), then whenever the base learner is SGD, SGLD, RLM, or the Gibbs algorithm, there exists $f_t(\cdot)$ that gives*

$$R_{excess}(\mathcal{A}, u_t) \leq f_t(u_t) = \kappa_t\left(a\beta_t + \frac{b\|\phi_t - w_t\|^2 + \epsilon_t + \epsilon_0}{\beta_t} + \Delta_t\right), \kappa_t, \epsilon_t, \beta_t, \Delta_t \in \mathbb{R}^+, a, b, \epsilon_0 > 0.$$
(1)

*The meta-parameter $u_t = (\beta_t, \phi_t)$ decomposes into an initialization or bias $\phi_t$ and a learning rate $\beta_t$ (if $\mathcal{A} \in \{SGD, SGLD\}$) or a regularization coefficient (if $\mathcal{A} \in \{RLM, Gibbs\}$). Moreover, $w_t$ denotes the (single) output for the (possibly randomized) base learner, and $\epsilon_0 \overset{\text{def}}{=} 2\alpha\|w_t - \mathbb{E}W_t\|^2$ characterizes the randomness of the output. Finally, $a$ and $b$ are constants, and $\epsilon_t, \kappa_t, \Delta_t$ are functions of the task sample size $m_t$ that characterize the base learner (see Appendix B and D).*

*Remark* 3.2. (a) The above bound could be *estimated from observation*, so we can use it to design meta-learning algorithms to control the excess risk. (b) Besides learning the initialization/bias, the corresponding learning rate/regularization coefficient should be further considered to control $R_{\text{excess}}$. (c) $f_t$ is convex w.r.t $u_t$ (proved in Appendix B.6). (d) This upper bound gives an **explicit** form of *task-level* trade-off that depends on the meta-parameter, where a large $\beta_t$ represents conserving less model prior (larger forgetting) and learning more from data to obtain the model $w_t$. The similarity $\|\phi_t - w_t\|$ between the model prior and the new task can affect the choice of $\beta_t$ for a better generalization.

## 3.2 Meta learner

Since the tasks are sequentially encountered without the i.i.d. assumption and the excess risk upper bound (cost function) $f_t$ is convex, we can naturally consider the meta-learning process as a repeated game, following the Online Convex Optimization (OCO) regime [40]. At each time $t$, the meta-learner first selects $u_t \in \mathcal{U}$ with prior knowledge learned from previous tasks, then the base learner outputs $w_t$ given $u_t$ and dataset $S_t$. After that, the cost function value $f_t$ is revealed. The goal of the meta learner is to select a sequence of meta-parameters $u_{1:T}$ so as to minimize the regret over rounds.

**Static task environment** In static environments, the task distributions $\mu_t \sim \tau, \forall t \in [T]$ are assumed to be sampled from a fixed environment $\tau$. The corresponding static regret is defined as the gap between the total cost of $u_{1:T}$ and that of an optimal meta-parameter in hindsight $u_T^*$:

$$R_T^{\text{static}}(u_{1:T}) \stackrel{\text{def}}{=} \sum_{t=1}^T f_t(u_t) - \sum_{t=1}^T f_t(u_T^*), \; u_T^* \stackrel{\text{def}}{=} \arg\min_{u \in \mathcal{U}} \sum_{t=1}^T f_t(u).$$

The hindsight $u_T^*$ converges to the true minimum $u^*$ w.r.t $\tau$ as $T \to \infty$. So the *task-level* trade-off can be well-balanced by designing algorithms with sub-linear regret to approach $u^*$.

**Shifting task environment** In a more general CML setting where the task environment $\tau_t$ can change at each time $t$, using a meta-learner designed to obtain sub-linear static regret will suffer a shifting comparator. As a result, the optimal $u_T^*$ may not converge to a fixed point, and $\sum_{t=1}^T f_t(u_T^*)$ can be vacuous when $T \to +\infty$, which induces the *meta-level* forgetting. See a formal justification in Appendix A.2.2. To this end, we assume the horizon $T$ of the sequential tasks can be divided into $N$ slots, and the $n$-th slot contains $M_n$ tasks. We further assume that the environment does not change with $t$ within each slot. The $N$ optimal priors, in hindsight, are noted as $u_{1:N}^*$. Consequently, we define a more versatile dynamic regret as:

$$R_T^{\text{dynamic}}(u_{1:N}^*) \stackrel{\text{def}}{=} \sum_{n=1}^N \sum_{k=1}^{M_n} \left[ f_{n,k}(u_{n,k}) - f_{n,k}(u_n^*) \right], \; u_n^* \stackrel{\text{def}}{=} \arg\min_{u \in \mathcal{U}} \frac{1}{M_n} \sum_{k=1}^{M_n} f_{n,k}(u).$$

Given $\sum_{n=1}^N M_n = T$, if $N \to T$, the definition is equivalent to the conventional dynamic regret. If $N \to 1$, it recovers the static regret. To balance the *meta-level* trade-off, we need to find algorithms with sub-linear dynamic regret to sequentially approach the optimums in $N$ slots (environments).

### 3.3 Continual Meta-Learning objective

In practice, we aim to train the task-specific model with few-shot data and hope that the model will generalize on unseen data. Hence, we formulate the CML objective as selecting $u_t$ at each time $t$ to ensure a small Average Excess Risk (AER) upper bound for the task sequence $\{\mu_t\}_{t=1}^T$, which is defined as follows given the base learner $\mathcal{A}$:

$$\text{AER}_{\mathcal{A}}^T \stackrel{\text{def}}{=} \frac{1}{T} \sum_{t=1}^T R_{\text{excess}}(\mathcal{A}, u_t) \leq \frac{1}{T} \sum_{t=1}^T f_t(u_t) = \frac{1}{T} R_T^{\text{dynamic}}(u_{1:N}^*) + \frac{1}{T} \sum_{n=1}^N \sum_{k=1}^{M_n} f_{n,k}(u_n^*).$$

We adopt the dynamic regret defined in shifting environments since it can recover the static setting w.l.o.g. Considering the two terms in the above upper bound of AER, the CML objective is two-fold: (a) design an appropriate online meta-learner to minimize the dynamic regret. (b) choose the optimal split of the $T$ tasks to $N$ stationary slots. Based on this objective, we propose a novel continual meta-learning framework and provide a corresponding theoretical analysis in the following sections.

## 4 Balancing bi-level learning-forgetting trade-off

### 4.1 An illustrative example

The excess risk upper bound provides the intuition that we can quickly learn a new task similar to our prior knowledge by relying on this prior. In this case, the initialization/bias $\phi_t$ (model prior) is close to the base learner output $w_t$ (*i.e.*, a small distance $\|\phi_t - w_t\|^2$), and the excess risk bound will be small. So we will elaborate on the learning-forgetting trade-off on an example of a shifting environment visualized in Fig. 2, based on the following definitions of *task similarities* characterized by the *distances* in the model parameter space [30].

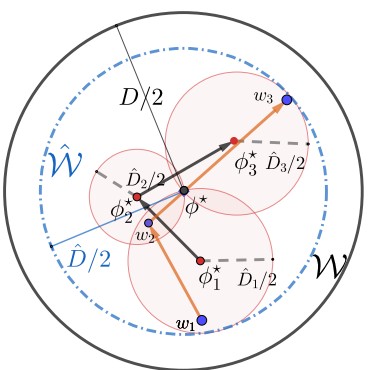

**Definition 4.1.** Let us denote $\hat{\mathcal{W}}$ as the set of all the learned model parameters: $\hat{\mathcal{W}} = \{w_t = \mathcal{A}(u_t, S_t)\}_{t=1}^T \cup \{\mathbf{0}\}$. Let $\hat{D}$ be the *diameter* of $\hat{\mathcal{W}}$ w.r.t norm $\|\cdot\|$: $\forall w, v \in \hat{\mathcal{W}}, \|w - v\| \leq \hat{D}$. In shifting task environments, we denote $\hat{D}_n$ the corresponding diameter for $n$-th slot. Assume $\mathcal{W}$ is a convex set containing all the possible model parameters. Then let $D$ be the diameter of $\mathcal{W}$: $\forall w, v \in \mathcal{W}, \|w - v\| \leq D$.

Figure 2: Illustration of parameter spaces in a shifting environment, where $\mathcal{W}, \hat{\mathcal{W}}$ and $D, \hat{D}, \hat{D}_n$ are defined in Def. 4.1. $\phi_n^\star$ is the $n$-th slot hindsight.

The task environment in Fig. 2 changes within three slots (i.e., three circles with diameters $\hat{D}_{1:3}$ in light red). The dark red point is denoted as $\phi_n^\star$ – the optimal initialization in hindsight for each slot.

Purple points $w_{1:3}$ are the learned model parameters in different slots. In static environments, CML can access additional few-shot data to address the *task-level forgetting*. Assume the models in $n$-th slot can recover performance (without forgetting) of similar tasks within a distance $\hat{D}_n/2$. Then, the optimal $\phi_n^*$ learned by CML can cover all the tasks in the slot with diameter $\hat{D}_n$. For instance, the two tasks in the first slot, shown in Fig. 2, have a large distance $\|w_1 - w_2\| > \hat{D}_1/2$. Directly adapting from one to another suffers catastrophic forgetting. However, keeping the optimal prior $\phi_1^*$ can address the forgetting, where $\|\phi_1^* - w_1\| < \hat{D}_1/2$ and $\|\phi_1^* - w_2\| < \hat{D}_1/2$.

However, this does not work in *shifting* environments. If we view three slots as one static environment (with large variance), the optimal overall model prior $\phi^*$ will be close to the origin, and the diameter is $\hat{D}$, which is much larger than any one of the slot $\hat{D}_n$. A larger diameter (*e.g.*, $\|\phi^* - w_3\| \gg \|\phi_3^* - w_3\|$) implies the need for more samples for each task to avoid forgetting, which is often not satisfied. A theoretical interpretation is provided in the discussion following Theorem 6.1. To address this, in the next session, we propose an algorithm that dynamically adapts the meta-parameter when environment change occurs so as to rapidly reconstruct meta-knowledge in each static slot. Moreover, it is not as catastrophic as directly adapting model parameters since the forgetting is w.r.t meta-knowledge, which is constrained in a core set of $\phi_{1:T}^*$, which has a diameter much smaller than $\hat{\mathcal{D}}$.

## 4.2 Dynamic algorithm

As discussed in the last section, it would be problematic in a shifting environment if the meta-learner adopts the same updating strategy as in the static setting. Specifically, if we aim to optimize the static regret, the learning rate of meta-parameter decays with $\mathcal{O}(1/t)$ for Follow The Leader (FTL) [41] and $\mathcal{O}(1/\sqrt{t})$ for Online Gradient Descent (OGD). Clearly, such strategies hardly learn from new tasks in a shifting environment. On the other side, a constant learning rate of the meta-parameter is often adopted for optimizing the regular dynamic regret [40]. This strategy could perform poorly when the environment only occasionally changes with a huge step, whereas the theoretical optimal learning rate becomes large. A large learning rate causes the meta learner to easily forget the previous tasks, making convergence difficult inside each slot. We offer a theoretical comparison with this in the discussion of Theorem 6.2.

To balance the learning and forgetting trade-off of meta-knowledge, we proposed the Dynamic Continual Meta-Learning (DCML) algorithm (see Algo. 1) that dynamically adjusts the learning rate of the meta-parameter. When a changing point is detected, the meta-learning

---

**Algorithm 1** Dynamic Continual Meta-Learning (DCML)

**Input:** Convex set $\mathcal{W}, \phi_1 = \mathbf{0} \in \mathcal{W}, T, \beta_1 > 0$, initial learning rate $\{\gamma_0, \rho\}$.
**if** $\exists$ a pre-trained model $\phi_0 \in \mathcal{W}$ **then**
  $\phi_1 = \phi_0$
**end if**
$n = 0, k = 0$
**for** $t = 1$ **to** $T$ **do**
  Sample task distribution: $\mu_t \sim \tau_t$;
  Sample dataset $S_t \sim \mu_t^{m_t}$;
  Learn base parameter $w_t = \mathcal{A}(u_t, S_t)$,
  *e.g.*, $\mathcal{A} = K$-step SGD or RLM
    $w_t = \phi_t - \sum_{i=1}^K \beta_t \nabla_{\phi_t^{i-1}} \mathcal{L}_{S_t}(\phi_t^{i-1}), \phi_t^0 = \phi_t,$
    $\phi_t^i = \phi_t^{i-1} - \beta_t \nabla_{\phi_t^{i-1}} \mathcal{L}_{S_t}(\phi_t^{i-1});$
    $w_t = \operatorname{argmin}_{w \in \mathcal{W}} \mathcal{L}_{S_t}(w) + \frac{1}{\beta_t} \|w - \phi_t\|^2;$
  Estimate excess risk upper bound $f_t(u_t)$ in Eq. (1);
  **if** $t == 1$ **or** environment change detected: $\tau_t \neq \tau_{t-1}$ **then**
    $M_n = k, n = n + 1, k = 1, \gamma_t = \rho;$
  **else**
    $k = k + 1, \gamma_t = \gamma_0/\sqrt{k};$
  **end if**
  Update meta-parameter:
  $\nabla f_t(u_t) = (\kappa_t(a - \frac{b\|\phi_t - w_t\|^2 + \epsilon_t + \epsilon_0}{\beta_t^2}), \frac{2b\kappa_t(\phi_t - w_t)}{\beta_t});$
  $u_{t+1} = \Pi_{\mathcal{U}}(u_t - \gamma_t \nabla f_t(u_t)), \textit{i.e.}:$
    $\phi_{t+1} = (1 - \frac{2b\kappa_t \gamma_t}{\beta_t})\phi_t + \frac{2b\kappa_t \gamma_t}{\beta_t} w_t ;$
    $\beta_{t+1} = \beta_t - \gamma_t(a\kappa_t - \frac{\kappa_t(b\|\phi_t - w_t\|^2 + \epsilon_t + \epsilon_0)}{\beta_t^2});$
**end for**

---

rate $\gamma_t$ adapts to a large hopping learning rate $\rho$, then relaunches the decay, in $\mathcal{O}(1/\sqrt{k})$, from $\gamma_0$ in each slot. Besides, the meta-parameter itself is also adaptively updated. According to $\phi_{t+1} = (1 - \frac{2b\kappa_t \gamma_t}{\beta_t})\phi_t + \frac{2b\kappa_t \gamma_t}{\beta_t} w_t$ in Algo. 1, adjusting $\beta_t$ and $\gamma_t$ represent the *explicit* control on task-level and meta-level trade-off, respectively. A corresponding discussion is provided in Appendix G.1. For a better understanding of DCML, we present a detailed introduction to it using SGD as the base learner in Appendix. G.3

Finally, the changing point can be detected with a broad class of Out-Of-Distribution (OOD) detection methods, *e.g.*, Bayes Online Changing-point Detection (BOCD) [42], loss threshold [13], or setting a fixed-length sliding window. Specifically, when the window size equals 1, a constant learning rate is set to the meta learner, which recovers optimizing the typical dynamic regret as in [40].

# 5 Main theorem

We present a general theorem for the proposed DCML framework, which could be deployed for various base learners with the same excess risk upper bound form as Eq. (1).

**Theorem 5.1.** *Consider both **static** and **shifting** environments. If the excess risk's upper bound of the base learner $\mathcal{A}(u_t, S_t)$ can be formulated as a unified form in Eq.(1), then, the **AER** of **DCML** (in Algo. 1) is upper bounded by:*

$$AER_{\mathcal{A}}^T \leq \underbrace{\frac{2}{T} \sum_{n=1}^{N} \sqrt{a(bV_n^2 + \epsilon_n + \epsilon_0)}\kappa_n + \frac{\Delta_n}{2}}_{\text{optimal trade-off in hindsight}} + \underbrace{\frac{3}{2T} \sum_{n=1}^{N} \tilde{D}_n G_n \sqrt{M_n - 1}}_{\text{average regret over slots}} + \underbrace{\frac{\tilde{D}_{max}}{T} \sqrt{2P^* \sum_{n=1}^{N} G_n^2}}_{\text{regret w.r.t environment shift}}$$

*Here, $\kappa_n = \sum_{k=1}^{M_n} \kappa_{n,k}$, $\Delta_n = \sum_{k=1}^{M_n} \kappa_{n,k}\Delta_{n,k}$, where the subscript $n,k$ indicates $k$-th task in $n$-th slot. The optimal meta-parameter in hindsight of $n$-th slot is $u_n^* = (\beta_n^*, \phi_n^*)$ and the path length of $N$ slots is $P^* = \sum_{n=1}^{N-1} \|u_n^* - u_{n+1}^*\| + 1$. $\phi_n^* = \sum_{k=1}^{M_n} \frac{\kappa_{n,k}}{\kappa_n} w_{n,k}$ is the weighted average of the base learner outputs within the slot. $\beta_n^* = \sqrt{(bV_n^2 + \epsilon_n + \epsilon_0)/a}$, where $V_n^2 = \sum_{k=1}^{M_n} \frac{\kappa_{n,k}}{\kappa_n} \|\phi_n^* - w_{n,k}\|_2^2$ is the variance in $n$-th slot and $\epsilon_n = \sum_{k=1}^{M_n} \frac{\kappa_{n,k}}{\kappa_n} \epsilon_{n,k}$. Moreover, $G_n$ is the maximal gradient norm of the cost function, $\tilde{D}_n$ is the diameter of meta-parameters in $n$-th slot, and $\tilde{D}_{max} = \max\{\tilde{D}_n\}_{n=1}^{N}$.*

The above theorem can be applied to static and shifting environments, where the proof is provided in Appendix D.2. For comparison, we also derive a general theorem from static regret in Appendix D.1.

**Important factors for bi-level trade-off** (a) The first term in Theorem 5.1 indicates the optimal trade-off can be achieved with the $N$ optimal meta-parameters, which is affected by the task similarities within each environment (slot variance $V_n^2$) and the environment changing points. (b) The second term is the average static regret over slots, which is related to the task similarities via the slot diameter $\tilde{D}_n$ and represents how well the algorithm can balance the task-level trade-off. (c) The last term is the regret w.r.t the environmental shift, reflecting how well the algorithm can address the meta-level trade-off. It is affected by the path length $P^*$, determined by the environment similarities and the non-stationarity (how frequently the environment changes).

*Remark* 5.2. The above theorem has considered different sample numbers for each task. We can see the best initialization (bias) in hindsight ($\phi_n^*$) is related to the weighted average of the task models $w_{n,k}$ in $n$-th slot through $\kappa_{n,k}$. E.g., for SGD, $\kappa_{n,k} = \sqrt{1/K\alpha + 2/m_{n,k}\alpha}L$ implies that the task trained on more samples will have a smaller weight. Since it's easier to recover model performance for the tasks with more samples, less information on these tasks is conserved in the meta-knowledge.

# 6 Results on specific base learners

Theorem 5.1 provides general theoretical guarantees without specifying the base leaner. In this section, we discuss the theorem in depth with two typical base learners: SGD within the meta-initialization setting and Gibbs algorithm for the meta-regularization.

## 6.1 Gibbs algorithm

**Theorem 6.1.** *Let $\mathcal{W} \subset \mathbb{R}^d$. Assume that the loss $\ell(\cdot, z) \in [0,1]$ is L-Lipschitz $\forall z \in \mathcal{Z}$ and that $\mathcal{L}_{\mu_t}(\cdot)$ has $\alpha$-quadratic-growth for all $t \in [T]$. Let $w_t = \mathcal{A}_{Gibbs}(\beta_t, \phi_t, S_t)$ be the output of the Gibbs algorithm (Definition B.2) on $S_t$, where $\phi_t$ is the mean of a prior Gaussian $\mathcal{N}(\phi_t, \sigma_t^2 \mathbf{1}_d)$ with $\sigma_t = m_t^{-1/4} d^{-1/4} L^{-1/2}$ and $\beta_t$ is the inverse temperature. Consider **DCML** (Algo. 1) and further assume that each slot has equal length $M$ and each task uses the sample number $m$. Then we have*

$$AER_{Gibbs}^T \in \mathcal{O}\left(\left(1 + \frac{1}{N} \sum_{n=1}^{N} V_n + \frac{\sqrt{MN} + \sqrt{P^*}}{M\sqrt{N}}\right) \frac{1}{m^{\frac{1}{4}}}\right).$$

The detailed proof is provided in Appendix E.1 and E.3. We also proved in Appendix E.2 that the static AER bound is in $\mathcal{O}\left(\left((V+1) + 1/\sqrt{T}\right) m^{-1/4}\right)$. As we claimed before, when $N = 1$ (which means the environment is static), the dynamic AER bound recovers the static AER bound.

**Benefits of DCML** Arbitrarily setting the prior mean $\phi_t$, the excess risk for single-task learning with Gibbs algorithm has an upper bound in $\mathcal{O}\left((D+1)m^{-1/4}\right)$, where $D$ is the diameter of the parameter space. While the static AER becomes $\mathcal{O}((V+1)m^{-1/4})$ with rate $\mathcal{O}(1/\sqrt{T})$. Since $V^2$ is the variance of all the outputs of the base learner, we have $V \leq \hat{D} \leq D$, which illustrates the benefits of the proposed DCML algorithm. Moreover, in **shifting environments**, the bound is further improved in Theorem 6.1, where the complexity factor is decreased by $\frac{1}{N}\sum_{n=1}^{N} V_n \leq V$ with rate $\mathcal{O}(1/\sqrt{M})$. This means that by considering environment change, DCML can achieve the same AER with a smaller $M$ (fewer tasks), *i.e.*, faster-constructing meta-knowledge in new environments.

## 6.2 Stochastic Gradient Descent (SGD)

**Theorem 6.2.** *Let $\mathcal{W} \subset \mathbb{R}^d$. Consider that $\ell(\cdot, z)$ is convex, $\beta$-smooth, L-Lipschitz $\forall z \in \mathcal{Z}$. Assume that $\mathcal{L}_{\mu_t}(\cdot)$ has $\alpha$-quadratic-growth for all $t \in [T]$. Let SGD be the base learner for each task where it outputs $w_t = \mathcal{A}_{SGD}(\eta_t, \phi_t, S_t)$ with learning rate $\eta_t$ and initialization $\phi_t$. Consider **DCML** (Algo. 1) for the SGD cost function. Further, assume each slot has equal length $M$, and each task uses the sample number $m$ and the same number of updating steps $K$. Then we have*

$$AER_{SGD}^T \in \mathcal{O}\left(\left(\frac{1}{N}\sum_{n=1}^{N} V_n + \frac{\sqrt{MN} + \sqrt{P^*}}{M\sqrt{N}}\right)\sqrt{\frac{1}{K} + \frac{1}{m}}\right).$$

The detailed proof is provided in Appendix E.4 and E.6. The CML setting uses *offline batch-training* for each task, so the related rate of the base learner is $\mathcal{O}(\sqrt{1/K + 1/m})$, which is determined by both the step number and the sample size. Let $\bar{V} = \frac{1}{N}\sum_{n=1}^{N} V_n$. CML focuses on fast learning new tasks and quickly recovers the model performance on past tasks with few-shot examples ($m$ is small). Hence, the optimal trade-off is dominated by the average deviation over slots $\bar{V}$ and the path length $P^*$, where $P^*$ reflects the environment similarities and the non-stationarity.

**Static environment** If the environment is static, *i.e.*, $N = 1$, we have $P^* = 1, M = T$. The bound becomes in $\mathcal{O}\left((V + \frac{1}{\sqrt{T}})\sqrt{\frac{1}{K} + \frac{1}{m}}\right)$, which recovers the static AER bound in Appendix E.5.

**Shifting environment** When the environment occasionally changes with a large step, as discussed in Sec 4.2, $N$ is small, and $M, P^*$ is large. The proposed bound has an improved rate $\mathcal{O}(1/\sqrt{M})$ on $P^*$ compared to $\mathcal{O}(\bar{V} + \frac{1}{\sqrt{M}} + \sqrt{\frac{P^*}{NM}})$ – an equivalent form of the task average regret bound in [31]. A more detailed comparison w.r.t [31] has been discussed in the proof for both static and dynamic AER. Similar to Theorem 6.1, the dynamic AER is improved w.r.t. the static AER with the improvement on complexity factor by $\bar{V} \leq V$. In addition, if the environments *differ a lot and frequently change*, $N, P^*$ is large, and $M$ is small. It's impossible to obtain a small AER, as demonstrated in the experiment on the Synbols dataset in Tab. 1.

## 7 Experiments

We first conduct analytic experiments on a synthetic dataset to offer an in-depth understanding of the theoretical counterpart. Then, we test the proposed algorithm on a recent CML benchmark – OSAKA [13], which empirically considers shifting environments on a large scale. The experimental results validate the proposed theory and illustrate the superiority of DCML in shifting environments.

### 7.1 Moving 2D Gaussian

We introduce a simple synthetic dataset, where the data is generated by the following strategy. Let the environment be a 2D Gaussian distribution with a moving mean. Initially, $\tau_0 = \mathcal{N}(\tilde{\phi}_0^*, \mathbb{I}_2)$ with $\tilde{\phi}_0^* = (-6, 6)^T$. Then at time point $t$, $\tau_t = \mathcal{N}(\tilde{\phi}_t^*, \mathbb{I}_2)$ is updated by: $\tilde{\phi}_t^* = \tilde{\phi}_{t-1}^* + (1, 1)$ with probability $p$ and $\tilde{\phi}_t^* = \tilde{\phi}_{t-1}^*$ with probability $1 - p$. Then we have $N$ changing points $\{t_n\}_{n=1}^{N}$. At each time step, we first sample $w_t^* \sim \tau_t$ from the current environment $\tau_t$ as the mean of the task distribution $\mu_t = \mathcal{N}(w_t^*, 0.1\mathbb{I}_2)$, which is also a 2D Gaussian. For task $t$, we sample $m$ examples $S_t^{tr} \sim \mu_t^m$ as train data, $m'$ samples $S_t^{te} \sim \mu_t^{m'}$ as test data. Then the corresponding empirical risk is

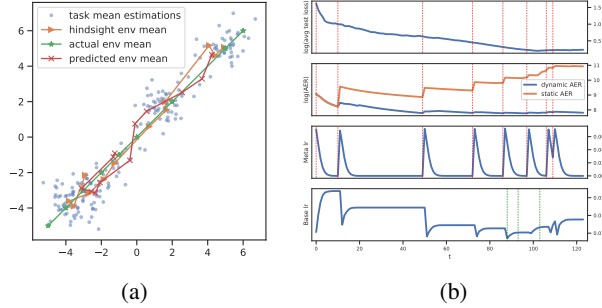

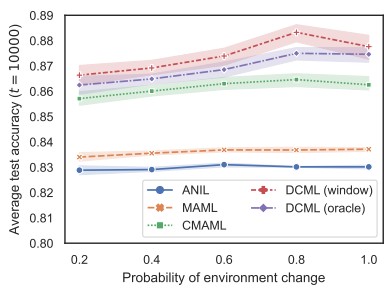

(a)            (b)

Figure 3: Moving 2D Gaussian mean estimation: (a) tracking example, (b) visualization of test loss, static and dynamic AER, learning rates of base and meta learners.

Figure 4: Average test acc at final step $t = 10000$ w.r.t environment change probability $p$ on the OMF dataset.

$\mathcal{L}_{S_t^{tr}}(w) = \frac{1}{m} \sum_{i=1}^{m} \|w - x_i\|^2$. SGD uses $\phi_t$ as initialization and a learning rate $\eta_t$ to optimize this objective, and outputs $w_t = \mathcal{A}_{\text{SGD}}(\eta_t, \phi_t, S_t^{tr})$. Since $w_t$ is obtained from limited data, it can be far away from $w_t^*$. If the environment shifts to the $n$-th slot at $t_n$, then the predicted environment mean $\hat{\phi}_n^* = \phi_{t_n}$. We denote $\tilde{\phi}_n^* = \tilde{\phi}_{t_n}^*$ the mean of the actual environment, and the hindsight environment mean of $n$-th slot as $\phi_n^* = \frac{1}{t_{n+1}-t_n} \sum_{t=t_n}^{t_{n+1}-1} w_t$.

In Fig. 3 (a), we visualize an example of tracking the environment shifts, where the task mean estimation is $w_t$, the output of SGD. The other three terms are as described above. We further visualize the test loss, the dynamic, and the static AER bounds in Fig. 3 (b). The red dotted lines represent the actual environment-changing points. We observe that the static AER is much larger than the dynamic AER, consistent with the result in Theorem 6.2. In addition, the evolving trend of dynamic AER is the same as the average test loss, which does not hold for static AER. We also find that the learning rate of the initialization $\phi_t$ (meta lr in Fig.3 (b)) is scheduled by the changing points, and the base learning rate is adjusted automatically by the algorithm. Additional experimental settings and results are provided in Appendix F.

### 7.2 OSAKA benchmark

**Experimental set-up** OSAKA considers a task-agnostic setting of unknown task boundaries and simultaneously switches the task and environment with low probabilities. We slightly modify the setting to introduce more non-stationarity by always switching to a new task and *changing the environment with probability $p$* at each timestep. We pre-train a meta-model as initialization from one environment. The new environment is selected with probability $0.5$ as the pre-trained one and $0.25$ for each of the two unseen environments, consisting of OOD tasks. We further test DCML on two representative datasets in OSAKA. For *Synbols* [43] dataset, the model is pre-trained to classify characters from an alphabet on randomized backgrounds. During CML time, the model is exposed to the pre-trained environment, a new alphabet for character classification, and a heterogeneous environment that conducts font classification, which induces *significant concept shift*. For *Omniglot-MNIST-FashionMNIST*(OMF), we pre-train the meta-model on the first 1000 classes of Omniglot [44]. Then, it is exposed to the full Omniglot dataset and two unseen datasets – MNIST [45] and FashionMNIST [46] at CML time. More details and baselines are deferred to Appendix F.1.1.

We tested 5-level of non-stationarity with $p = \{0.2, 0.4, 0.6, 0.8, 1.0\}$ for OMF and for Synbols we set $p = \{0.2, 0.4, 1.0\}$. The average test accuracy of all encountered tasks is used as the performance metric, which reflects the AER. In addition, we denote DCML (oracle) for the algorithm as offering correct environment-changing points and DCML (window) as using a fixed-length sliding window.

**Results** The average test accuracies at the last timestep for all environments and every single environment are reported in Tab.1 for OMF with $p = 0.4$ and Synbols with $p = 0.2$. Additional experimental results can be found in Appendix F.2.2. Clearly, DCML constantly outperforms all the baselines with improvements of $1 \sim 2\%$ points on OMF data and $0.7\%$ points on Synbols data compared to the best baseline method. A comparison w.r.t powerful baselines is presented in Fig. 5 for OMF data, where the average test accuracy over rounds in three environments is visualized separately.

Table 1: Average test accuracy (%) on OMF & Synbols datasets of OSAKA benchmark

| Model | OMF, $p = 0.4$ | | | | Synbols, $p = 0.2$ | | | |
|---|---|---|---|---|---|---|---|---|
| | All Env. | Omniglot | MNIST | FMNIST | All Env. | Alpha. | New Alpha. | Font |
| Fine Tuning | 10.8± 1.5 | 11.2± 2.2 | 10.4± 0.8 | 10.5± 0.8 | 25.3± 0.7 | 25.4± 0.9 | 25.1± 0.4 | 25.2± 0.5 |
| MetaCOG [8] | 40.9± 0.9 | 54.4± 1.2 | 25.6± 0.9 | 28.9± 0.4 | 25.3± 0.8 | 25.7± 0.9 | 25.0± 0.8 | 25.1± 1.0 |
| MetaBGD [8] | 54.5± 7.1 | 70.5± 8.5 | 40.2± 5.5 | 36.8± 5.8 | 29.9± 6.9 | 31.8± 9.6 | 28.1± 4.8 | 27.5± 3.6 |
| ANIL [47] | 83.2± 0.3 | 99.1± 0.1 | 73.9± 0.1 | 60.2± 0.1 | 60.5± 0.4 | 77.6± 0.6 | 53.6± 0.2 | 32.7± 0.4 |
| MAML [4] | 83.8± 0.4 | **99.3± 0.1** | 75.4± 0.1 | 61.0± 0.1 | 76.7± 0.4 | **96.8± 0.1** | 73.1± 0.1 | 40.5± 0.7 |
| CMAML [13] | 85.9± 0.7 | 98.1± 0.3 | 83.8± 1.8 | 64.2± 0.9 | 61.9± 2.5 | 76.2± 2.1 | 56.8± 3.2 | 37.9± 2.4 |
| DCML(oracle) | 86.5± 0.7 | 97.4± 0.6 | 85.7± 2.6 | 65.9± 1.4 | 77.2± 0.5 | 96.0± 0.3 | 73.4± 0.1 | **42.4± 0.6** |
| DCML(window) | **86.9± 0.7** | 97.2± 0.7 | **86.8± 2.0** | **66.7± 1.7** | **77.4± 0.4** | 96.2± 0.2 | **73.8± 0.2** | 42.4± 0.7 |

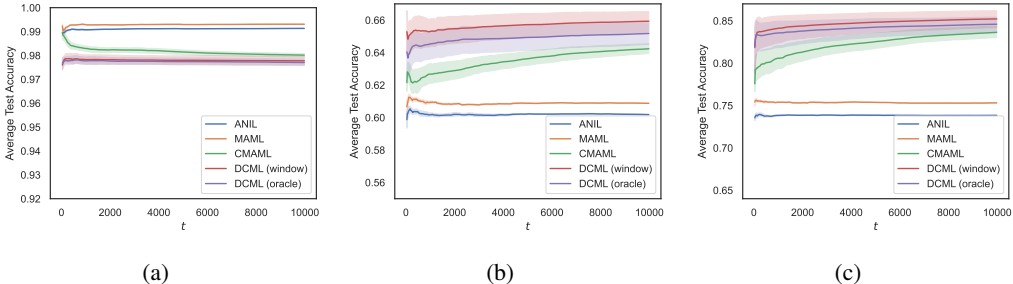

Figure 5: Average test accuracy on (a) Omniglot, the pre-trained environment, (b) FashionMNIST, and (c) MNIST, the two unseen environments where the environment shifts with probability $p = 0.2$.

**Analysis of learning and forgetting** In Fig. 5 for OMF data, MAML and ANIL do not suffer forgetting since they never update meta-parameters. The proposed algorithm forgets a bit more than CMAML but learns much faster in new environments, illustrating a better trade-off. The fine-grained adjustment for the meta-learning rate makes it possible to quickly reconstruct meta-knowledge w.r.t environment change and constantly learn in new environments. On the other hand, MetaBGD and MetaCOG perform poorly since using BGD hinders acquiring new knowledge. All the methods perform worse on Synbols data in Tab. 1, where each environment is more diverse than OMF, and the environments differ a lot, indicating a large $\hat{D}_n$ and $P^*$. Moreover, we observe a performance drop of CMAML on Synbols. Since CMAML uses empirical loss for modulation, it is prone to overfitting and can cause difficulty of convergence inside each slot when $\hat{D}_n$ is large, as discussed in Sec. 4.1.

**Analysis of stability w.r.t shifting probability** We conduct an ablation study on OMF w.r.t environment change probability in Fig. 4. The results demonstrate the stability of the proposed algorithm and the increase in performance w.r.t $p$. Since the environments have high similarities in OMF, the meta-model is better learned by seeing more tasks given the fixed time steps with a larger $p$.

**Analysis of environment change detection** Empirical results suggest that DCML (window) performs better than DCML (oracle). This is reasonable when slots of small length exist where adapting to new environments causes additional context-switch costs (justified in Appendix F.2.1). Given the changing probability $p$, the expected slot length is $1/p$, used as the sliding window size. In practice, even though $p$ is unknown, hyper-parameter searching for the one-dimensional window size is much easier. Because the similarity between any two environments can differ a lot, it's hard to find a single optimal hyper-parameter for change detection methods like BOCD or loss threshold.

## 8 Conclusion

This paper theoretically studies the stability and plasticity dilemma in CML. Based on the novel AER objective, it proposes a continual meta-learning framework in both static and shifting environments. The proposed DCML algorithm can quickly reconstruct meta-knowledge to alleviate forgetting and quickly adapt to new environments when change occurs. The corresponding theory provides tighter bounds and more flexible base learner selections. In addition, adaptively learning the meta-parameter can facilitate the training process in deep learning. Empirical evaluations on both synthetic and real datasets demonstrate the superiority of the proposed method.

## Acknowledgments and Disclosure of Funding

We appreciate constructive feedback from anonymous reviewers and meta-reviewers. This work is supported by the Natural Sciences and Engineering Research Council of Canada (NSERC) Discovery Grant, the Collaborative Research and Development Grant from SSQ Assurances and NSERC, and the China Scholarship Council.

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

# Appendix

## Table of Contents

# A Preliminaries

In the appendix, we assume that all sets are convex subsets of $\mathbb{R}^d$, and we use $\|\cdot\|$ to denote the Euclidean norm.

## A.1 Definitions

**Definition A.1** (Lipschitzness). A function $f : \mathcal{W} \to \mathbb{R}$ is $L$-Lipschitz w.r.t norm $\|\cdot\|$ if for all $w_1, w_2 \in \mathcal{W}$, $|f(w_1) - f(w_2)| \le L\|w_1 - w_2\|$

**Definition A.2** (Quadratic Growth). A function $f : \mathcal{W} \to \mathbb{R}$ has $\alpha$-quadratic-growth w.r.t $\|\cdot\|$ for $\alpha > 0$ if for any $w \in \mathcal{W}$, we have:

$$\frac{\alpha}{2}\|w - w^*\|^2 \le f(w) - f(w^*),$$

where $w^*$ denotes the global minimum point of $f$ which is closest to $w$.

**Definition A.3** (Convex). A function $f : \mathcal{W} \to \mathbb{R}$ is convex w.r.t norm $\|\cdot\|$ if $f$ is everywhere sub-differentiable and if $\forall x, y \in \mathcal{W}$,

$$f(y) \ge f(x) + \langle \nabla f(x), y - x \rangle,$$

where $\nabla f(x)$ is a subgradient of $f$ at $x$.

**Definition A.4** (Strongly Convex). A function $f : \mathcal{W} \to \mathbb{R}$ is $\alpha$-strongly convex w.r.t norm $\|\cdot\|$ if $f$ is everywhere sub-differentiable and if $\forall x, y \in \mathcal{W}$,

$$f(y) \ge f(x) + \langle \nabla f(x), y - x \rangle + \frac{\alpha}{2}\|y - x\|^2,$$

where $\nabla f(x)$ is a subgradient of $f$ at $x$.

**Definition A.5** (Smoothness). A function $f : \mathcal{W} \to \mathbb{R}$ is $\beta$-smooth w.r.t norm $\|\cdot\|$ if $f$ is everywhere sub-differentiable and if $\forall x, y \in \mathcal{W}$,

$$f(y) \le f(x) + \langle \nabla f(x), y - x \rangle + \frac{\beta}{2}\|y - x\|^2,$$

where $\nabla f(x)$ is a subgradient of $f$ at $x$.

**Definition A.6** (Bregman divergence). Let $R : \mathcal{U} \to \mathbb{R}$ be an everywhere sub-differentiable strongly convex regularization function. Then the Bregman divergence w.r.t function $R$ for any $x, y \in \mathcal{U}$ is defined as:

$$B_R(x\|y) \overset{\text{def}}{=} R(x) - R(y) - \langle \nabla R(y), (x - y) \rangle.$$

## A.2 Online Convex Optimization (OCO)

### A.2.1 Definitions of regrets

**Definition A.7** (Static regret). The static regret of an OCO algorithm on the action set $\mathcal{U}$ w.r.t a sequence of cost functions $\{f_t : \mathcal{U} \to \mathbb{R}\}_{t=1}^T$ is defined as:

$$R_T \overset{\text{def}}{=} \sum_{t=1}^T f_t(u_t) - \min_{u \in \mathcal{U}} \sum_{t=1}^T f_t(u)$$

**Definition A.8** (Traditional dynamic regret [40]). The dynamic regret of an OCO algorithm on the action set $\mathcal{U}$ w.r.t a sequence of cost functions $\{f_t : \mathcal{U} \to \mathbb{R}\}_{t=1}^T$ and the comparator sequence $\psi_{1:T}, \forall \psi_t \in \mathcal{U}$ is defined as:

$$R_T \overset{\text{def}}{=} \sum_{t=1}^T f_t(u_t) - \sum_{t=1}^T f_t(\psi_t)$$

### A.2.2 Static regret may be vacuous in shifting environments

*Proof.* For a shifting environment that has $u_T^* = \arg\min_u \frac{1}{T}\sum_{t=1}^T f_t(u)$ changing with $T$, we can prove that the static regret may be vacuous as $T$ goes to infinity.

Let us denote $F_T = \sum_{t=1}^T f_t(u_T^*)$, $F_{T-1} = \sum_{t=1}^{T-1} f_t(u_{T-1}^*)$, $F_0 = 0$.

Assume that the cost functions are non-negative, which is usually the case. Whenever we have $f_t(u_T^*) - f_t(u_{T-1}^*) \geq a > 0$, we can obtain

$$F_T - F_{T-1} = \sum_{t=1}^{T-1}(f_t(u_T^*) - f_t(u_{T-1}^*)) + f_T(u_T^*) \geq a(T-1) + f_T(u_T^*).$$

The above inequality holds for any $T \geq 1$, so we can rewrite it as $F_t - F_{t-1} \geq a(t-1) + f_t(u_t^*)$. By summing each side from $t = 1$ to $t = T$, we have $F_T \geq F_0 + \frac{aT(T-1)}{2} + \sum_{t=1}^T f_t(u_t^*) \geq \frac{aT(T-1)}{2}$. Consequently, we have $\frac{1}{T}F_T = \frac{1}{T}\sum_{t=1}^T f_t(u_T^*) \geq \frac{a(T-1)}{2}$, which means the static regret will be vacuous when $T \to \infty$. $\qquad\square$

### A.2.3 Online algorithms

**Definition A.9** (Follow The Leader (FTL)). Given a sequence of **strongly convex** cost functions $\{f_t : \mathcal{U} \to \mathbb{R}\}_{t \geq 1}$, **FTL** plays for $t > 1$:

$$u_t = \arg\min_{u \in \mathcal{U}} \sum_{i=1}^{t-1} f_i(u).$$

**Definition A.10** (Follow The Regularized Leader (FTRL)). Given a sequence of **convex** cost functions $\{f_t : \mathcal{U} \to \mathbb{R}\}_{t \geq 1}$, a **strongly convex** regularization function $R : \mathcal{U} \to \mathbb{R}$, and the starting point $u_1 = \arg\min_{u \in \mathcal{U}} R(u)$, **FTRL** plays for $t > 1$:

$$u_t = \arg\min_{u \in \mathcal{U}} \sum_{i=1}^{t-1} f_i(u) + R(u).$$

Khodak et al. [31] write FTRL as $u_t = \arg\min_{u \in \mathcal{U}} \sum_{i=1}^{t-1} f_i(u) + B_R(u\|u_1)$, while we follow Shalev-Shwartz et al. [41] and Hazan et al. [40].

**Definition A.11** (Online Mirror Descent (OMD)). Given a sequence of **convex** cost functions $\{f_t : \mathcal{U} \to \mathbb{R}\}_{t \geq 1}$, a **strongly convex** regularization function $R : \mathcal{U} \to \mathbb{R}$, and $u_1 = \nabla R^*(0)$, **OMD** plays for $t > 1$:

$$u_t = \nabla R^*(-\eta \sum_{i=1}^{t-1} \nabla f_i(u_i)),$$

where $R^*(\tilde{u}) = \max_u \langle u, \tilde{u} \rangle - R(u)$ is the Fenchel conjugate of $R(u)$.

There exists an equivalent linearized version of the above FTRL and lazy OMD through the linearization of the convex cost functions.

**Definition A.12** (Lazy OMD). Given a sequence of **convex** cost functions $\{f_t : \mathcal{U} \to \mathbb{R}\}_{t \geq 1}$, a **strongly convex** regularization function $R : \mathcal{U} \to \mathbb{R}$, and $\nabla R(\tilde{u}_1) = 0$, $u_1 = \arg\min_{u \in \mathcal{U}} B_R(u\|\tilde{u}_1)$, then **lazy OMD** and **FTRL** play for $t > 1$:

$$\tilde{u}_t = \arg\min_u \langle \eta \nabla f_{t-1}(u_{t-1}), u \rangle + B_R(u\|u_{t-1})$$
$$u_t = \arg\min_{u \in \mathcal{U}} B_R(u\|\tilde{u}_t).$$

The proof of the equivalence can be found in Hazan et al. [40].

# B  Missing Proofs in Section 3

**Definition B.1.** Consider any base learner $\mathcal{A}$ that takes a fixed prior information $u$ with a data set $S \sim \mu^m$ as input and outputs $W = \mathcal{A}(S, u) \sim P_{W|S,u}$. Define the expected generalization gap of algorithm $\mathcal{A}$ as

$$\text{gen}(\mu, \mathcal{A}) \stackrel{\text{def}}{=} \mathbb{E}_{\mathcal{A},S}[\mathcal{L}_\mu(\mathcal{A}(S,u)) - \mathcal{L}_S(\mathcal{A}(S,u))] = \mathbb{E}_{W,S}[\mathcal{L}_\mu(W) - \mathcal{L}_S(W)]$$

and the expected excess risk as

$$R_{\text{excess}}(\mathcal{A}, u) \stackrel{\text{def}}{=} \mathbb{E}_{\mathcal{A},S}[\mathcal{L}_\mu(\mathcal{A}(S,u)) - \mathcal{L}_\mu(w^*)] = \mathbb{E}_{W,S}[\mathcal{L}_\mu(W) - \mathcal{L}_\mu(w^*)],$$

where $w^* = \arg\min_{w \in \mathcal{W}} \mathcal{L}_\mu(w)$ is the hypothesis that achieves the minimum true risk.

## B.1  Excess risk bound for Gibbs algorithm

**Lemma B.2.** *Let $Q$ be an arbitrary distribution on $\mathcal{W}$ and let $\beta > 0$ be the inverse temperature that balances fitting and generalization. Then, we can jointly denote $u = (\beta, Q)$. Let $S \sim \mu^m$ be a sample of examples. The solution to the optimization problem*

$$P^*_{W|S,u} = \arg\inf_{P_{W|S}} \left\{ \mathbb{E}_{W,S}[L_S(W)] + \frac{1}{\beta}\mathbb{E}_S D_{KL}(P_{W|S}\|Q) \right\}$$

*is given by the **Gibbs algorithm** which satisfies*

$$dP^*_{W|S,u}(w) = \frac{e^{-\beta L_S(w)}dQ(w)}{\mathbb{E}_{W \sim Q}\left[e^{-\beta L_S(W)}\right]}.$$

See Proof of Theorem 5 in [48].

**Theorem B.3** (Excess risk bound for Gibbs algorithm of meta-parameter $u = (\beta, \phi)$)**.** *Suppose $\mathcal{W} = \mathbb{R}^d$ and assume that $\ell(\cdot, z) \in [0, 1]$ is L-Lipschitz for all $z \in \mathcal{Z}$. Let $W_g \sim P_{W_g|S,u}$ denote the output of the Gibbs algorithm [3] applied on data set $S$ with meta-parameter $u = (\beta, Q)$. Let $w^*$ be the hypothesis that achieves the minimum true risk among $\mathcal{W}$. The excess risk of $W_g$ satisfies*

$$R_{excess}(Gibbs, u) \stackrel{\text{def}}{=} \mathbb{E}[\mathcal{L}_\mu(W_g)] - \inf_{w \in \mathcal{W}} \mathcal{L}_\mu(w)$$

$$\leq \frac{\beta}{2m} + \inf_{\sigma > 0}\left(\sigma L\sqrt{d} + \frac{1}{\beta}D_{KL}(\mathcal{N}(w^*, \sigma^2 \mathbf{1}_d)\|Q)\right).$$

*Specifically, if we assume $Q$ is the Gaussian distribution $\mathcal{N}(\phi, \sigma^2 \mathbf{1}_d)$ with $\sigma = m^{-1/4}d^{-1/4}L^{-1/2}$ and denote $u = (\beta, \phi)$, then we have*

$$R_{excess}(Gibbs, u) \leq \frac{\beta}{2m} + m^{-1/4}L^{1/2}d^{1/4} + \frac{d^{1/2}m^{1/2}L}{2\beta}\|\phi - w^*\|^2.$$

*Proof.* The original proof can be found in Corollary 3 [48], we provide the proof below with more details and some tiny modifications. Consider an arbitrary data-free distribution related to the unknown minimum $w^*$ of the true risk, *i.e.*, $\mathcal{N}(w^*, \sigma^2 \mathbf{1}_d)$. Then for any $\sigma > 0$, we have:

$$\mathbb{E}_{W_g,S}[\mathcal{L}_\mu(W_g)] = \mathbb{E}_{W_g,S}[\mathcal{L}_S(W_g)] + \text{gen}(\mu, \text{Gibbs})$$

$$\leq \mathbb{E}_{W_g,S}[\mathcal{L}_S(W_g)] + \frac{\beta}{2m}$$

$$\leq \mathbb{E}_{W_g,S}[\mathcal{L}_S(W_g)] + \frac{1}{\beta}\mathbb{E}_S D_{KL}(P_{W_g|S,u}\|Q) + \frac{\beta}{2m}$$

$$\leq \int_{\mathcal{W}} \mathbb{E}_S \mathcal{L}_S(w)\mathcal{N}(w; w^*, \sigma^2 \mathbf{1}_d)dw + \frac{1}{\beta}D_{KL}(\mathcal{N}(w^*, \sigma^2 \mathbf{1}_d))\|Q) + \frac{\beta}{2m}$$

$$= \int_{\mathcal{W}} \mathcal{L}_\mu(w)\mathcal{N}(w; w^*, \sigma^2 \mathbf{1}_d)dw + \frac{1}{\beta}D_{KL}(\mathcal{N}(w^*, \sigma^2 \mathbf{1}_d))\|Q) + \frac{\beta}{2m}$$

---

[3]In some definitions, the output of the Gibbs algorithm is the optimal distribution itself. Here, we consider a sampling from this distribution.

The first step is obtained with previous work's results that Gibbs algorithm is $(\frac{2\beta}{m}, 0)$-differentially private [49]. The second inequality makes use of the non-negativity of KL divergence. The last inequality derives naturally from the fact that the output of the Gibbs algorithm follows the optimal distribution for the objective function in Lemma B.2.

Since $\ell(., z)$ is $L$-Lipschitz for all $z \in \mathcal{Z}$, we can obtain the following by combining the Jensen's inequality:

$$|\mathcal{L}_\mu(w) - \mathcal{L}_\mu(w^*)| \le \mathbb{E}_{Z \sim \mu}[|\ell(w, Z) - \ell(w^*, Z)|] \le L\|w - w^*\|, \forall w \in \mathcal{W}.$$

Then we have

$$\int_{\mathcal{W}} \mathcal{L}_\mu(w) \mathcal{N}(w; w^*, \sigma^2 \mathbf{1}_d) dw \le \int_{\mathcal{W}} (\mathcal{L}_\mu(w^*) + L\|w - w^*\|) \mathcal{N}(w; w^*, \sigma^2 \mathbf{1}_d) dw$$
$$\le \mathcal{L}_\mu(w^*) + L\sigma\sqrt{d}.$$

So we have for any $\sigma > 0$,

$$\mathbb{E}[\mathcal{L}_\mu(W_g)] - \inf_{w \in \mathcal{W}} \mathcal{L}_\mu(w) \le \frac{\beta}{2m} + \inf_{\sigma > 0} \left( \sigma L\sqrt{d} + \frac{1}{\beta} D_{KL}(\mathcal{N}(w^*, \sigma^2 \mathbf{1}_d) \| Q) \right)$$
$$= \frac{\beta}{2m} + \inf_{\sigma > 0} \left( \sigma L\sqrt{d} + \frac{1}{\beta} D_{KL}(\mathcal{N}(w^*, \sigma^2 \mathbf{1}_d) \| \mathcal{N}(\phi, \sigma^2 \mathbf{1}_d)) \right)$$
$$= \frac{\beta}{2m} + \inf_{\sigma > 0} \left( \sigma L\sqrt{d} + \frac{1}{2\beta\sigma^2} \|\phi - w^*\|^2 \right)$$
$$\le \frac{\beta}{2m} + m^{-1/4} L^{1/2} d^{1/4} + \frac{d^{1/2} m^{1/2} L}{2\beta} \|\phi - w^*\|^2$$

The last three lines are obtained with the assumption that $Q$ is the Gaussian distribution $\mathcal{N}(\phi, \sigma^2 \mathbf{1}_d)$ and setting $\sigma = m^{-1/4} d^{-1/4} L^{-1/2}$.

$\square$

## B.2 Excess risk bound for SGD

**Theorem B.4** (Excess Risk Bound for SGD of meta-parameter $u = (\eta, \phi)$)**.** *Suppose $\mathcal{W} = \mathbb{R}^d$, let $w^* \stackrel{def}{=} \arg\min_{w \in \mathcal{W}} \mathcal{L}_\mu(w)$ be the hypothesis that achieves the minimum population risk in $\mathcal{W}$. Consider the SGD algorithm with $K$ updates starting from $W_1 = \phi$: $W_{k+1} = W_k - \eta \nabla \ell(W_k, Z_k)$, where $Z_k$ is a sample randomly selected from data set $S$. Assume the loss function $\ell(\cdot, z)$ is convex, $\beta$-smooth and L-Lipschitz for all $z \in \mathcal{Z}$. Let the output of the algorithm be $W = \frac{1}{K}\sum_{k=1}^{K} W_k$. Then, the excess risk bound is given by:*

$$R_{excess}(SGD, u) \stackrel{def}{=} \mathbb{E}_{W,S}[\mathcal{L}_\mu(W)] - \inf_{w \in \mathcal{W}} \mathcal{L}_\mu(w) \le \frac{\|\phi - w^*\|^2}{2\eta K} + (\frac{L^2}{2} + \frac{L^2 K}{m})\eta \, .$$

*Proof.* The excess risk can be decomposed as

$$\mathbb{E}_{W,S}[\mathcal{L}_\mu(W) - \min_w \mathcal{L}_\mu(w)] = \mathbb{E}_{W,S}[\mathcal{L}_\mu(W) - \mathcal{L}_S(W)] + \mathbb{E}_{W,S}\mathcal{L}_S(W) - \min_w \mathcal{L}_\mu(w)$$
$$= \mathbb{E}_{W,S}[\mathcal{L}_\mu(W) - \mathcal{L}_S(W)] + [\mathbb{E}_{W,S}\mathcal{L}_S(W) - \mathbb{E}_S\mathcal{L}_S(w^*)]$$
$$= gen(\mu, SGD) + \mathbb{E}_{W,S}\mathcal{L}_S(W) - \mathbb{E}_S\mathcal{L}_S(w^*) \, .$$

The first term is the generalization gap defined in Definition B.1. Note that the second term is not the optimization error $\epsilon_{opt}^W \stackrel{def}{=} |\mathbb{E}_{W,S}[\mathcal{L}_S(W)] - \mathbb{E}_{W,S}[\mathcal{L}_S(w_S^*)]|$ used in [38], which is the expected gap between the SGD output $W$ and the ERM output $w_S^*$. The optimization error indicates how close the SGD output can approximate the possibly intractable exact ERM solution, which is an upper bound of the absolute value of the second term $|\mathbb{E}_{W,S}\mathcal{L}_S(W) - \mathbb{E}_S[\mathcal{L}_S(w^*)]| \le \epsilon_{opt}^W$ (can be derived with Lemma 5.1 in [50]).

Here, we directly make use of the convexity and obtain the gap compared to the true minimum. It's easy to bound $\mathbb{E}_{W,S}\mathcal{L}_S(W) - \mathbb{E}_S\mathcal{L}_S(w^*)$ using the method provided in Shalev-Shwartz and Ben-David [51] with some tiny modifications.

Since we have $W_{k+1} = W_k - \eta \nabla \ell(W_k, Z_k)$, $W_{1:K}$ is related to the randomness introduced by the sampling path $Z_{1:K}$ from $S$. Moreover, for the convex loss function, we have:

$$\mathbb{E}_{W,S}\mathcal{L}_S(W) - \mathbb{E}_S\mathcal{L}_S(w^*) = \mathbb{E}_S\left[\mathbb{E}_{W \sim P_{W|S,u}}\mathcal{L}_S(W) - \mathcal{L}_S(w^*)\right]$$
$$= \mathbb{E}_S\left[\mathbb{E}_{W_{1:K}}\mathcal{L}_S(\frac{1}{K}\sum_{k=1}^{K}W_k) - \mathcal{L}_S(w^*)\right]$$
$$\le \mathbb{E}_S\left[\mathbb{E}_{W_{1:K}}\frac{1}{K}\sum_{k=1}^{K}\mathcal{L}_S(W_k) - \mathcal{L}_S(w^*)\right]$$
$$\le \mathbb{E}_S\left[\mathbb{E}_{W_{1:K}}\frac{1}{K}\sum_{k=1}^{K}\langle W_k - w^*, \nabla\mathcal{L}_S(W_k)\rangle\right]$$
$$= \mathbb{E}_S\left[\mathbb{E}_{V_{1:K-1}}\frac{1}{K}\sum_{k=1}^{K}\langle W_k - w^*, \nabla\mathcal{L}_S(W_k)\rangle\right]$$
$$= \mathbb{E}_S\left[\frac{1}{K}\sum_{k=1}^{K}\mathbb{E}_{V_{1:k-1}}\langle W_k - w^*, \nabla\mathcal{L}_S(W_k)\rangle\right] \, ,$$

where $V_k = \nabla\ell(W_k, Z_k)$. The inequalities are obtained by using Jensen's inequality and the last two equalities are obtained with the fact that $W_{1:K}$ is determined by $V_{1:K-1}$.

Moreover, we have $\mathbb{E}[V_k|W_k] = \mathbb{E}[V_k|V_{1:k-1}] = \nabla\mathcal{L}_S(W_k)$. The first equality comes from the update rule, where $W_k$ is determined by the previous $k-1$ gradients and the meta parameter $u = (\eta, \phi)$. Note that $u$ is not a random variable and hence no randomness needs to be considered. The second equality holds because $Z_k$ is uniformly sampled from $S$, so $V_k$ is an unbiased estimator of $\nabla\mathcal{L}_S(W_k)$.

Hence,

$$\mathbb{E}_{V_{1:K}}\left(\frac{1}{K}\sum_{k=1}^{K}\langle W_k - w^*, V_k\rangle\right) = \frac{1}{K}\sum_{k=1}^{K}\mathbb{E}_{V_{1:K}}[\langle W_k - w^*, V_k\rangle]$$

$$= \frac{1}{K}\sum_{k=1}^{K}\mathbb{E}_{V_{1:k}}[\langle W_k - w^*, V_k\rangle]$$

$$= \frac{1}{K}\sum_{k=1}^{K}\mathbb{E}_{V_{1:k-1}}\mathbb{E}_{V_k}[\langle W_k - w^*, V_k\rangle|V_{1:k-1}]$$

$$= \frac{1}{K}\sum_{k=1}^{K}\mathbb{E}_{V_{1:k-1}}[\langle W_k - w^*, \mathbb{E}_{V_k}[V_k|V_{1:k-1}]\rangle]$$

$$= \frac{1}{K}\sum_{k=1}^{K}\mathbb{E}_{V_{1:k-1}}[\langle W_k - w^*, \nabla\mathcal{L}_S(W_k)\rangle]$$

By using Lemma14.1 in Shalev-Shwartz and Ben-David [51], we can obtain that

$$\mathbb{E}_{W,S}\mathcal{L}_S(W) - \mathbb{E}_S\mathcal{L}_S(w^*) \leq \mathbb{E}_S\left[\mathbb{E}_{V_{1:K}}\left(\frac{1}{K}\sum_{k=1}^{K}\langle W_k - w^*, V_k\rangle\right)\right]$$

$$\leq \frac{\|W_1 - w^*\|^2}{2\eta K} + \frac{\eta}{2K}\sum_{k=1}^{K}\mathbb{E}\|V_k\|^2.$$

Since the loss function is $L$-Lipschitz, we can further obtain:

$$\mathbb{E}_{W,S}\mathcal{L}_S(W) - \mathcal{L}_\mu(w^*) \leq \frac{\|W_1 - w^*\|^2}{2\eta K} + \frac{\eta}{2}L^2.$$

We have assumed that the loss function is convex and $\beta$-smooth. Therefore, following Theorem 4.7 in Hardt et al. [50], we have the following upper bound for the generalization gap of SGD if the step size is smaller than $\frac{2}{\beta}$:

$$|\text{gen}(\mu, SGD)| \leq \frac{L^2 K}{m}\eta.$$

So, we can conclude

$$R_{\text{excess}}(\text{SGD}, u) = \mathbb{E}_{W,S}[\mathcal{L}_\mu(W)] - \inf_{w\in\mathcal{W}}\mathcal{L}_\mu(w) \leq \frac{\|\phi - w^*\|^2}{2\eta K} + (\frac{L^2}{2} + \frac{L^2 K}{m})\eta.$$

$\square$

## B.3 Excess risk bound for SGLD

**Theorem B.5** (Excess risk bound for SGLD of meta-parameter $u = (\eta, \phi)$)**.** *Suppose $\mathcal{W} = \mathbb{R}^d$, let $w^* \stackrel{def}{=} \arg\min_{w \in \mathcal{W}} \mathcal{L}_\mu(w)$ be the hypothesis that achieves the minimum population risk in $\mathcal{W}$. For SGLD algorithm with $K$ adaptation starting from $W_1 = \phi$: $W_{k+1} = W_k - \eta\nabla\ell(W_k, Z_k) + \xi_k$, where $Z_k$ is a sample randomly selected from data set $S$, $\xi_k$ is the independently injected isotropic Gaussian noise $\mathcal{N}(0, \sigma^2 \mathbf{1}_d)$. Assume the loss function $\ell(\cdot, z)$ is convex and $L$-Lipschitz for all $z \in \mathcal{Z}$. And let the output of the algorithm be $W = \frac{1}{K}\sum_{k=1}^{K} W_k$. Then we have the following excess risk bound:*

$$R_{excess}(SGLD, u) \stackrel{def}{=} \mathbb{E}_{W,S}[\mathcal{L}_\mu(W)] - \inf_{w \in \mathcal{W}} \mathcal{L}_\mu(w) \le \frac{\|\phi - w^*\|^2}{2\eta K} + \frac{\eta}{2}L^2 + \frac{d\sigma^2}{2\eta} + \sqrt{\frac{K}{4m}}\frac{\eta L}{\sigma}.$$

*Proof.* Analogous to the proof of SGD, we can decompose the excess risk as

$$\mathbb{E}_{W,S}[\mathcal{L}_\mu(W) - \min_w \mathcal{L}_\mu(w)] = \mathbb{E}_{W,S}[\mathcal{L}_\mu(W) - \mathcal{L}_S(W)] + \mathbb{E}_{W,S}\mathcal{L}_S(W) - \min_w \mathcal{L}_\mu(w)$$
$$= \mathbb{E}_{W,S}[\mathcal{L}_\mu(W) - \mathcal{L}_S(W)] + \mathbb{E}_{W,S}\mathcal{L}_S(W) - \mathbb{E}_S[\mathcal{L}_S(w^*)]$$
$$= gen(\mu, SGLD) + \mathbb{E}_{W,S}\mathcal{L}_S(W) - \mathbb{E}_S\mathcal{L}_S(w^*).$$

According to Pensia et al. [38], we have $|gen(\mu, SGLD)| \le \sqrt{\frac{1}{4m}\frac{K\eta^2 L^2}{\sigma^2}}$.

In addition, we have the update for SGLD: $W_{k+1} = W_k - \eta\nabla\ell(W_k, Z_k) + \xi_k$, where $W_{1:K}$ is related to the randomness introduced by the sampling path $Z_{1:K}$ from $S$ and the injected noise $\xi_{1:K}$.

Denote $V_k = \nabla\ell(W_k, Z_k) + \frac{\xi_k}{\eta}$, then we have $\mathbb{E}[V_k|W_k] = \mathbb{E}[V_k|V_{1:k-1}] = \nabla\mathcal{L}_S(W_k)$.

Conduct similar proof as the above SGD algorithm and use Lemma 14.1 in [51], we can obtain that

$$\mathbb{E}_{W,S}\mathcal{L}_S(W) - \mathbb{E}_S\mathcal{L}_S(w^*) \le \mathbb{E}_S\left[\mathbb{E}_{V_{1:K}}\left(\frac{1}{K}\sum_{k=1}^{K}\langle W_k - w^*, V_k\rangle\right)\right]$$
$$\le \frac{\|W_1 - w^*\|^2}{2\eta K} + \frac{\eta}{2K}\sum_{k=1}^{K}\mathbb{E}\|V_k\|^2.$$

Since the loss function is $L$-Lipschitz, we have

$$\mathbb{E}_{W,S}\mathcal{L}_S(W) - \mathcal{L}_\mu(w^*) \le \frac{\|W_1 - w^*\|^2}{2\eta K} + \frac{\eta}{2}(L^2 + d\frac{\sigma^2}{\eta^2}).$$

Combining the aforementioned generalization gap bound of SGLD, we conclude the proof.

$\square$

## B.4 Excess risk bound for Regularized Loss Minimization (RLM)

**Theorem B.6** (Excess risk bound for RLM of meta-parameter $u = (\beta, \phi)$). *Suppose $\mathcal{W} = \mathbb{R}^d$, let $w^* \stackrel{def}{=} \arg\min_{w \in \mathcal{W}} \mathcal{L}_\mu(w)$ be the hypothesis that achieves the minimum population risk in $\mathcal{W}$. Let us denote the output of RLM with Tikhonov regularization as $W = \arg\min_{w \in \mathcal{W}} \mathcal{L}_S(w) + \frac{1}{\beta}\|w - \phi\|^2$. Suppose the loss function $\ell : \mathcal{W} \times \mathcal{Z} \to [0, 1]$ is convex and L-Lipschitz, so the excess risk of the above algorithm is bounded by:*

$$R_{excess}(RLM, u) \stackrel{def}{=} \mathbb{E}_{W,S}[\mathcal{L}_\mu(W)] - \inf_{w \in \mathcal{W}} \mathcal{L}_\mu(w) \leq \frac{\|\phi - w^*\|^2}{\beta} + \frac{2L^2\beta}{m} \,.$$

*Proof.* Here, since the algorithm is deterministic, we can consider $P_{W|S,u}$ as a delta distribution centered on $w_S^* = \mathcal{A}(u, S)$. Then, the bound becomes

$$\mathbb{E}_{W,S}[\mathcal{L}_\mu(W)] = \mathbb{E}_S[L_\mu(w_S^*)] = \mathbb{E}_S[L_\mu(\mathcal{A}(u, S)]$$

$$\leq \inf_{w \in \mathcal{W}} \mathcal{L}_\mu(w) + \frac{1}{\beta}\|\phi - w^*\|^2 + \frac{2L^2\beta}{m} \,,$$

which is equivalent to Corollary 13.8 in [51] with $\phi = \mathbf{0}, \beta = \frac{1}{\lambda}$.

To prove the above bound, we use the same decomposition of excess risk:

$$\mathbb{E}_{W,S}[\mathcal{L}_\mu(W) - \min_w \mathcal{L}_\mu(w)] = \mathbb{E}_S[L_\mu(w_S^*) - \mathcal{L}_S(w_S^*)] + \mathbb{E}_S\mathcal{L}_S(w_S^*) - \min_w \mathcal{L}_\mu(w)$$

$$= \mathbb{E}_S[L_\mu(w_S^*) - \mathcal{L}_S(w_S^*)] + \mathbb{E}_S\mathcal{L}_S(w_S^*) - \mathbb{E}_S[\mathcal{L}_S(w^*)]$$

$$= gen(\mu, \text{RLM}) + \mathbb{E}_S\mathcal{L}_S(w_S^*) - \mathbb{E}_S\mathcal{L}_S(w^*) \,.$$

From Corollary 13.6 in [51], the stability of RLM rule satisfies

$$gen(\mu, RLM) = \mathbb{E}_S[L_\mu(w_S^*) - \mathcal{L}_S(w_S^*)] \leq \frac{2L^2\beta}{m} \,,$$

which means the RLM rule using Tikhonov regularization described above is on average-replace-one-stable with rate $\frac{2L^2\beta}{m}$. From the definition of this algorithm, we have for any $w \in \mathcal{W}$,

$$\mathcal{L}_S(w_S^*) \leq \mathcal{L}_S(w_S^*) + \frac{1}{\beta}\|w_S^* - \phi\|^2 \leq \mathcal{L}_S(w) + \frac{1}{\beta}\|w - \phi\|^2 \,.$$

Take expectation w.r.t $S$, we can obtain $\mathbb{E}_S\mathcal{L}_S(w_S^*) \leq \mathcal{L}_\mu(w) + \frac{1}{\beta}\|w - \phi\|^2, \forall w \in \mathcal{W}$. Finally, combining the stability bound and letting $w = w^*$, we conclude the proof. $\square$

## B.5 Unified form of excess risk upper bound

**Lemma B.7.** *Assume that $\mathcal{L}_{\mu_t}(\cdot)$ has $\alpha$-quadratic growth (defined in Appendix A.2) $\forall t \in [T]$ and that for a given base learner $\mathcal{A}$, we have $R_{excess}(\mathcal{A}, u_t) \leq \frac{b_t + c_t \|\phi_t - w_t^*\|^2}{\beta_t} + d_t \beta_t + e_t$, with $b_t, e_t \geq 0, c_t, d_t > 0$. Then we have $R_{excess}(\mathcal{A}, u_t) \leq f_t(u_t)$, where $f_t(u_t)$ is given by eq. (1).*

*Proof.* Let $\bar{w}_t = \mathbb{E} W_t$ be the expected output, and $\xi_0 \stackrel{\text{def}}{=} \|w_t - \bar{w}_t\|^2$ characterizes the randomness of the algorithm output, which should be small enough.

Then, using Titu's lemma, Jensen's inequality, and the $\alpha$-quadratic growth assumption, we get:

$$
\begin{aligned}
\|\phi_t - w_t^*\|^2 &= \|\phi_t - \bar{w}_t + \bar{w}_t - w_t^*\|^2 \\
&\leq 2\|\phi_t - \bar{w}_t\|^2 + 2\|\bar{w}_t - w_t^*\|^2 \\
&= 2\|\phi_t - \bar{w}_t\|^2 + 2\|\mathbb{E} W_t - w_t^*\|^2 \\
&\leq 2\|\phi_t - \bar{w}_t\|^2 + 2\mathbb{E}\|W_t - w_t^*\|^2 \\
&\leq 2\|\phi_t - \bar{w}_t\|^2 + \frac{4\mathbb{E}[\mathcal{L}_{\mu_t}(W_t) - \mathcal{L}_{\mu_t}(w_t^*)]}{\alpha} \\
&\leq 2\|\phi_t - \bar{w}_t\|^2 + \frac{4(\frac{b_t + c_t \|\phi_t - w_t^*\|^2}{\beta_t} + d_t \beta_t + e_t)}{\alpha} \, .
\end{aligned}
$$

Rearrange and put the upper bound of $\|\phi_t - w_t^*\|^2$ into the excess risk bound, replace $\sqrt{c_t/d_t}\beta_t' = \alpha\beta_t - 4c_t$ with $\beta_t > 4c_t/\alpha$ and use again the Titu's lemma, we get

$$
R_{\text{excess}}(\mathcal{A}, u_t) \leq 4\alpha\sqrt{c_t d_t} \left( \frac{\|\phi_t - w_t\|^2 + \xi_0 + \frac{b_t}{4c_t} + \frac{4d_t c_t}{\alpha^2} + \frac{e_t}{\alpha}}{\beta_t'} + \frac{2\sqrt{d_t c_t}}{\alpha^2} + \frac{\beta_t'}{4\alpha^2} + \frac{e_t}{4\alpha\sqrt{d_t c_t}} \right)
$$

It's easy to relate $\kappa_t, \Delta_t, \epsilon_t, \epsilon_0$ to the terms in the above bound. Consequently, we have $\epsilon_0 = 2\alpha\xi_0 = 2\alpha\|w_t - \mathbb{E} W_t\|^2$, $\kappa_t = 2\sqrt{c_t d_t}$, $a = \frac{1}{2\alpha}$, $b = 2\alpha$, $\epsilon_t = \frac{\alpha b_t}{2c_t} + \frac{8d_t c_t}{\alpha} + 2e_t$, and $\Delta_t = \frac{4\sqrt{c_t d_t}}{\alpha} + \frac{e_t}{2\sqrt{c_t d_t}}$.

We prove Proposition E.1 and Proposition E.4 without using this Lemma and provide more details in the corresponding derivations. $\qquad\square$

**Theorem B.8.** *For any $t \in [T]$, assume that $\mathcal{L}_{\mu_t}(\cdot)$ has $\alpha$-quadratic growth [4] (see Definition A.2), then whenever the base learner is SGD, SGLD, RLM, or the Gibbs algorithm, there exists $f_t(\cdot)$ that gives*

$$
R_{excess}(\mathcal{A}, u_t) \leq f_t(u_t) = \kappa_t \left( a\beta_t + \frac{b\|\phi_t - w_t\|^2 + \epsilon_t + \epsilon_0}{\beta_t} + \Delta_t \right), \kappa_t, \epsilon_t, \beta_t, \Delta_t \in \mathbb{R}^+, a, b, \epsilon_0 > 0 \, .
$$

*The meta-parameter $u_t = (\beta_t, \phi_t)$ decomposes into an initialization or bias $\phi_t$ and a learning rate $\beta_t$ (if $\mathcal{A} \in \{SGD, SGLD\}$) or a regularization coefficient (if $\mathcal{A} \in \{RLM, Gibbs\}$). Moreover, $w_t$ denotes the (single) output for the (possibly randomized) base learner, and $\epsilon_0 \stackrel{\text{def}}{=} 2\alpha\|w_t - \mathbb{E} W_t\|^2$ characterizes the randomness of the output. Finally, $a$ and $b$ are constants, and $\epsilon_t, \kappa_t, \Delta_t$ are functions of the task sample size $m_t$ that characterize the base learner (see Appendix B and D).*

*Proof.* According to Theorem B.3, Theorem B.4, Theorem B.5 and Theorem B.6, we have

$$
\begin{aligned}
R_{\text{excess}}(\text{Gibbs}, (\eta_t, \phi_t)) &\leq \frac{\eta_t}{2m_t} + m_t^{-1/4} L^{1/2} d^{1/4} + \frac{d^{1/2} m_t^{1/2} L}{2\eta_t}\|\phi_t - w_t^*\|^2 \, , \\
R_{\text{excess}}(\text{SGD}, (\eta_t, \phi_t)) &\leq \frac{\|\phi_t - w_t^*\|^2}{2\eta_t K_t} + (\frac{L^2}{2} + \frac{L^2 K_t}{m_t})\eta_t \, , \\
R_{\text{excess}}(\text{SGLD}, (\eta_t, \phi_t)) &\leq \frac{\|\phi_t - w_t^*\|^2}{2\eta_t K_t} + \frac{\eta_t}{2}L^2 + \frac{d\sigma^2}{2\eta_t} + \sqrt{\frac{K_t}{4m_t}}\frac{\eta_t L}{\sigma} \, , \\
R_{\text{excess}}(\text{RLM}, (\eta_t, \phi_t)) &\leq \frac{\|\phi_t - w_t^*\|^2}{\eta_t} + \frac{2L^2\eta_t}{m_t} \, .
\end{aligned}
$$

---

[4]We assumed quadratic growth for $\ell(\cdot, z), \forall z \in \mathcal{Z}$ in the submitted version, which is stronger than the current assumption and is unnecessary.

Combining Lemma B.7, we observe that $b_t = 0, c_t = \frac{d^{1/2}m_t^{1/2}L}{2}, d_t = \frac{1}{2m_t}, e_t = m_t^{-1/4}L^{1/2}d^{1/4}$ for Gibbs algorithm; $b_t = 0, c_t = \frac{1}{2K_t}, d_t = \frac{L^2}{2} + \frac{L^2 K_t}{m_t}, e_t = 0$ for SGD; $b_t = \frac{d\sigma^2}{2}, c_t = \frac{1}{2K_t}, d_t = \frac{L^2}{2} + \sqrt{\frac{K_t}{4m_t}}\frac{L}{\sigma}, e_t = 0$ for SGLD; and $b_t = 0, c_t = 1, d_t = \frac{2L^2}{m_t}, e_t = 0$ for RLM. Therefore, simply adding the transformation $\eta_t = \sqrt{c_t/d_t}\beta_t/\alpha + 4c_t/\alpha$ inside the algorithm [5], the algorithms mentioned above have a unified form of excess risk upper bound as defined in eq. (1). $\qquad\square$

### B.6 Convexity of the unified upper bound

**Lemma B.9.** $\forall t \in [T]$, *the cost function that has the following form:*

$$f_t(u_t) = f_t(\beta_t, \phi_t) = \kappa_t(a\beta_t + \frac{b\|\phi_t - w_t\|^2 + \epsilon_t + \epsilon_0}{\beta_t} + \Delta_t), \beta_t, \kappa_t, \epsilon_t, \Delta_t \in \mathbb{R}^+, a, b, \epsilon_0 > 0,$$

*is convex w.r.t* $u_t = (\beta_t, \phi_t)$, *where* $\phi_t, w_t \in \mathcal{W} \subseteq \mathbb{R}^d$.

*Proof.*

$$\partial_{\beta_t} f_t = \kappa_t(a - \frac{b\|\phi_t - w_t\|^2 + \epsilon_t + \epsilon_0}{\beta_t^2})$$

$$\partial_{\phi_t} f_t = \kappa_t \frac{2b(\phi_t - w_t)}{\beta_t}$$

The Hessian matrix of $f_t(\beta_t, \phi_t)$:

$$\nabla^2 f_t = \begin{pmatrix} 2\kappa_t \frac{b\|\phi_t - w_t\|^2 + \epsilon_t + \epsilon_0}{\beta_t^3} & \frac{-2b(\phi_t - w_t)}{\beta_t^2}\kappa_t \\ \frac{-2b(\phi_t - w_t)}{\beta_t^2}\kappa_t & \kappa_t \frac{2b}{\beta_t}\mathbf{1}_{d\times d} \end{pmatrix}$$

For any $h = (h_0, h_d) \in \mathbb{R} \times \mathbb{R}^d$, and $\beta_t > 0, \phi \in \mathbb{R}^d$ we have:

$$(\nabla^2 f(u_t)h, h) = \kappa_t \frac{2(\epsilon_t + \epsilon_0)h_0^2 + 2b\|h_0(\phi_t - w_t) - \beta_t h_d\|^2}{\beta_t^3} \geq 0$$

, so $f_t$ is convex w.r.t $u_t$. $\qquad\square$

---

[5]Such transformation is not unique, *e.g.*, $\beta_t$ can be scaled by $\sqrt{\alpha}$. It depends on the concrete implementation.

# C   Table of Notations

Table 2: Summary of major notations

| Notation | Description |
|---|---|
| $T$ | length of task sequence |
| $N$ | number of slots (environments) |
| $t$ | index of task at time $t$ |
| $n, k$ | $k$-th task insider $n$-th slot |
| $M_n, M$ | number of tasks inside slot |
| $m_t, m_{n,k}, m$ | number of examples for each task |
| $\tau_t$ | environment distribution |
| $\mu_t$ | task distribution |
| $\mathcal{Z} = \mathcal{X} \times \mathcal{Y}$ | example space |
| $\mathcal{A}$ | base leaner/learning algorithm |
| $\mathcal{W} \subset \mathbb{R}^d$ | hypothesis/model parameter space |
| $w_t, w_{n,k}$ | task-model, algorithm output |
| $\mathcal{U}$ | meta-parameter space |
| $u_t, u_{n,k}$ | meta-parameter |
| $u^* = (\beta^*, \phi^*)$ | optimal meta-parameter in hindsight |
| $\phi_t, \phi_{n,k}$ | model initialization/bias |
| $\beta_t, \beta_{n,k}$ | learning rate/regularization coefficient of $\mathcal{A}$ |
| $\gamma_t, \gamma_{n,k}$ | learning rate of meta-parameter |
| $\rho$ | hopping learning rate |
| $K_t, K_{n,k}$ | number of gradient steps for SGD |
| $D$ | diameter of $\mathcal{W}$ |
| $\hat{D}$ | diameter of all the algorithm outputs space $\hat{\mathcal{W}}$ |
| $\tilde{D}$ | diameter of all meta-parameters |
| $V$ | variance of all the algorithm outputs |
| $\tilde{D}_n, \hat{D}_n$ | diameter in $n$-th slot |
| $\tilde{D}_{\max} = \max\{\tilde{D}_n\}_{n=1}^N$ | maximum of slot diameters |
| $V_n$ | slot variance |
| $G_n$ | maximum of cost function gradient norm in $n$-th slot |
| $P^*$ | path length |
| $L$ | Lipschitz constant |
| $\alpha$ | quadratic-growth parameter |
| $\epsilon_0$ | randomness of algorithm output |
| $\epsilon_v$ | parameter for controlling range of $\beta_t$ |
| $\mathcal{B} = \left[\sqrt{\tilde{\epsilon}_0/a}, \sqrt{(b\hat{D}_n^2 + \epsilon + \tilde{\epsilon}_0)/a}\right]$ | range of $\beta_{n,k}$, where $\tilde{\epsilon}_0 = \epsilon_0 + \epsilon_v$ |
| $\gamma_{n,0} = \dfrac{\tilde{D}_n}{G_n \sqrt{k}}$ | theoretically optimal value |
| $\rho = \sqrt{\dfrac{2P^* \tilde{D}_{\max}^2}{\sum_{n=1}^N G_n^2}}$ | theoretically optimal hopping learning rate |

# D Missing Proofs in Section 5

## D.1 AER bound in static environments

---

**Algorithm 2** Meta-OGD (Static Environment)

---

**Require**: Convex set $\mathcal{W}, T, \phi_1 = \mathbf{0} \in \mathcal{W}, \beta_1 > 0$, initial learning rate $\gamma$;
**for** $t \leftarrow 1$ *to* $T$ **do**
  Sample task distribution: $\mu_t \sim \tau_t$;
  Sample dataset $S_t \sim \mu_t$;
  Select meta parameter $u_t = (\beta_t, \phi_t) \in \mathcal{U}$;
  Learn base parameter $w_t = \mathcal{A}(u_t, S_t)$,
  Estimate excess risk upper bound $f_t(u_t)$ in Eq. (1);
  Update learning rate of the meta-parameter: $\gamma_t = \gamma/\sqrt{t}$;
  Update the meta-parameter:
    $u_{t+1} = \Pi_{\mathcal{U}}(u_t - \gamma_t \nabla f_t(u_t)), \nabla f_t(u_t) = (a\kappa_t - \frac{b\kappa_t \|\phi_t - w_t\|^2 + \kappa_t \epsilon_t + \kappa_t \epsilon_0}{\beta_t^2}, \frac{2b\kappa_t(\phi_t - w_t)}{\beta_t})$;
    *i.e.*:
    $\phi_{t+1} = (1 - \frac{2b\kappa_t\gamma_t}{\beta_t})\phi_t + \frac{2b\kappa_t\gamma_t}{\beta_t}w_t$ ; $\beta_{t+1} = \beta_t - \gamma_t(a\kappa_t - \frac{\kappa_t(b\|\phi_t - w_t\|^2 + \epsilon_t + \epsilon_0)}{\beta_t^2})$;
**end**

---

**Theorem D.1** (AER bound in static environment). *Let us consider the **static** environment. If the excess risk's upper bound of the base learner $\mathcal{A}(\beta_t, \phi_t)$ can be formulated as the aforementioned unified cost function form in Lemma B.6. Then, running **Meta-OGD** ( Algo. 2) as the meta learner , the **AER** is bounded as:*

$$AER_{\mathcal{A}}^T \leq (2\sqrt{a(bV^2 + \epsilon + \epsilon_0)}) \sum_{t=1}^{T} \frac{\kappa_t}{T} + \sum_{t=1}^{T} \frac{\kappa_t \Delta_t}{T}$$

$$+ \frac{3\kappa_{max}}{2\tilde{\epsilon}_0}\sqrt{T}\left(\sqrt{(\hat{D}^2 + \frac{b\hat{D}^2 + \epsilon_{max}}{a})(4ab^2\tilde{\epsilon}_0\hat{D}^2 + a^2(\epsilon_{max} + b\hat{D}^2)^2)}\right),$$

*where $\hat{D}$ is the diameter of the base learner outputs, $\phi^* = \sum_{t=1}^{T} \frac{\kappa_t}{\sum_{t=1}^{T} \kappa_t} w_t$ is the optimal initialization (or bias) in hindsight, $V^2 = \sum_{t=1}^{T} \frac{\kappa_t}{\sum_{t=1}^{T} \kappa_t}\|\phi^* - w_t\|^2$ is the variance of the base learner outputs, and $\epsilon = \frac{\sum_{t=1}^{T} \kappa_t \epsilon_t}{\sum_{t=1}^{T} \kappa_t}, \kappa_{max} = \max\{\kappa_{1:T}\}, \epsilon_{max} = \max\{\epsilon_{1:T}\}, \tilde{\epsilon}_0 = \epsilon_0 + \epsilon_v. \epsilon_0$ is the across tasks maximum corresponding term in eq.(1), and $\epsilon_v \in \mathbb{R}^+$, If $\epsilon_v > bV^2 + \epsilon$, we replace $\epsilon_0$ in the bound with $\tilde{\epsilon}_0$. Let $\mathcal{B} \overset{def}{=} \left[\sqrt{\tilde{\epsilon}_0/a}, \sqrt{(b\hat{D}^2 + \epsilon + \tilde{\epsilon}_0)/a}\right]$, then, the meta-parameter set in Algo. 2 is $\mathcal{U} \overset{def}{=} \mathcal{B} \times \mathcal{W}$.*

*Proof.* At first, in a static environment, all the tasks share the same optimal prior information $u^* = \arg\min_{u \in \mathcal{U}} \sum_{t=1}^{T} f_t(u)$ (the comparator in static regret). As proved in Lemma B.6, the cost functions are convex, so the sum of $T$ convex functions $\sum_{t=1}^{T} f_t(u)$ is convex. Then we have the minimum attained at $u^* = (\beta^*, \phi^*)$. By solving $\min_u \sum_{t=1}^{T} f_t(u)$, we get $\phi^* = \sum_{t=1}^{T} \frac{\kappa_t}{\sum_{t=1}^{T} \kappa_t} w_t$ and $\beta^* = \sqrt{(bV^2 + \epsilon + \epsilon_0)/a}$, where $\epsilon = \frac{\sum_{t=1}^{T} \kappa_t \epsilon_t}{\sum_{t=1}^{T} \kappa_t}$ and $V^2 = \sum_{t=1}^{T} \frac{\kappa_t}{\sum_{t=1}^{T} \kappa_t}\|\phi^* - w_t\|^2$.

According to Algo. 2, $\phi_t$ is determined by the previous base learner outputs $w_{1:t-1}$ and the initialization $\phi_1 = \mathbf{0}$. As long as $\frac{2b\kappa_t\gamma_t}{\beta_t} \leq 1, \forall t$, we have $\phi_t$ is always constrained within the ball (of diameter $\hat{D}$) that contains $\hat{\mathcal{W}}$. So we have $0 \leq \|\phi_t - w_t\| \leq \hat{D}$ and $V^2 = \sum_{t=1}^{T} \frac{\kappa_t}{\sum_{t=1}^{T} \kappa_t}\|\phi^* - w_t\|^2 \leq \hat{D}^2$. Moreover, we constrain $\beta_t \in \left[\sqrt{\tilde{\epsilon}_0/a}, \sqrt{(b\hat{D}^2 + \epsilon + \tilde{\epsilon}_0)/a}\right]$, where $\tilde{\epsilon}_0 = \epsilon_0 + \epsilon_v$ is related to the randomness of the base learner output $\epsilon_0$ and a factor $\epsilon_v$ to control the range of $\beta_t$. We will see later that $\epsilon_v$ can affect the gradient norm of the meta-parameter.

To derive the static regret bound, we need to use the following result first:

$$\|u_{t+1} - u^*\|^2 = \|\Pi_{\mathcal{U}}(u_t - \gamma_t \nabla f_t(u_t)) - u^*\| \leq \|u_t - \gamma_t \nabla f_t - u^*\|^2$$
$$\leq \|u_t - u^*\|^2 + \gamma_t^2 \|\nabla f_t\|^2 - 2\gamma_t \langle \nabla f_t, (u_t - u^*) \rangle .$$

So we have $\langle \nabla f_t, (u_t - u^*) \rangle \leq \frac{\|u_t - u^*\|^2 - \|u_{t+1} - u^*\|^2}{2\gamma_t} + \frac{\gamma_t \|\nabla f_t\|^2}{2}$.

Moreover, with $\beta_t \in \left[\sqrt{\tilde{\epsilon}_0/a}, \sqrt{(b\hat{D}^2 + \epsilon + \tilde{\epsilon}_0)/a}\right]$, we have

$$|\partial_{\beta_t} f_t| = |\kappa_t(a - \frac{b\|\phi_t - w_t\|^2 + \epsilon_t + \epsilon_0}{\beta_t^2})| \leq \frac{a\kappa_t(b\hat{D}^2 + \epsilon_{\max})}{\tilde{\epsilon}_0},$$

$$|\partial_\phi f_t| = \frac{2b\kappa_t \|\phi_t - w_t\|}{\beta_t} \leq \frac{2b\kappa_t \hat{D}}{\sqrt{\tilde{\epsilon}_0/a}}.$$

Hence, $\|\nabla f_t\|^2 \leq \frac{a^2 \kappa_t^2 (b\hat{D}^2 + \epsilon_{\max})^2}{\tilde{\epsilon}_0^2} + \frac{4ab^2 \kappa_t^2 \hat{D}^2}{\tilde{\epsilon}_0}$.

In addition, $\|u_t - u^*\|^2 = \|\phi_t - \phi^*\|^2 + |\beta_t - \beta^*|^2 \leq \hat{D}^2 + \frac{b\hat{D}^2 + \epsilon_{\max}}{a}$.

So the static regret is bounded by:

$$R_T^{\text{static}}(u^*) = \sum_{t=1}^T f_t(u_t) - f_t(u^*) \leq \sum_{t=1}^T \langle \nabla f_t, (u_t - u^*) \rangle$$

$$\leq \sum_{t=1}^T \frac{\|u_t - u^*\|^2 - \|u_{t+1} - u^*\|^2}{2\gamma_t} + \sum_{t=1}^T \frac{\gamma_t \|\nabla f_t\|^2}{2}$$

$$\leq \sum_{t=1}^T \|u_t - u^*\|^2 (\frac{1}{2\gamma_t} - \frac{1}{2\gamma_{t-1}}) + \sum_{t=1}^T \frac{\gamma_t \|\nabla f_t\|^2}{2}$$

$$\leq (\hat{D}^2 + \frac{b\hat{D}^2 + \epsilon_{\max}}{a}) \frac{1}{2\gamma_T} + (\frac{4ab^2 \hat{D}^2}{\tilde{\epsilon}_0} + \frac{a^2(\epsilon_{\max} + b\hat{D}^2)^2}{\tilde{\epsilon}_0^2}) \sum_{t=1}^T \frac{\gamma_t \kappa_t^2}{2}$$

$$\leq \frac{3\kappa_{\max}}{2\tilde{\epsilon}_0} \sqrt{T} \left( \sqrt{(\hat{D}^2 + \frac{b\hat{D}^2 + \epsilon_{\max}}{a})(4ab^2 \tilde{\epsilon}_0 \hat{D}^2 + a^2(\epsilon_{\max} + b\hat{D}^2)^2)} \right) .$$

We have the learning rate schedule $\gamma_t = \frac{\gamma}{\sqrt{t}}$ and $1/\gamma_0 = 0$. The third inequality holds because of the aforementioned bounds of $\|u_t - u^*\|^2$ and $\|\nabla f_t\|$. The last step is obtained by setting $\gamma = \frac{\tilde{\epsilon}_0 \sqrt{\hat{D}^2 + \frac{b\hat{D}^2 + \epsilon_{\max}}{a}}}{\kappa_{\max} \sqrt{4ab^2 \tilde{\epsilon}_0 \hat{D}^2 + a^2(\epsilon_{\max} + b\hat{D}^2)^2}}$ and $\sum_{t=1}^T \frac{1}{\sqrt{t}} \leq 2\sqrt{T}$, where $\kappa_{\max} = \max\{\kappa_{1:T}\}$, $\epsilon_{\max} = \max\{\epsilon_{1:T}\}$.

According to the setting of $\tilde{\epsilon}_0$, we have two conditions.

1. If $\epsilon_v \leq bV^2 + \epsilon$, $\beta^* = \sqrt{(bV^2 + \epsilon + \epsilon_0)/a} \in \left[\sqrt{\tilde{\epsilon}_0/a}, \sqrt{(b\hat{D}^2 + \epsilon + \tilde{\epsilon}_0)/a}\right]$ can be attained. Then, for the optimal comparator in hindsight $u^* = (\beta^*, \phi^*)$, we have the accumulated cost:

$$\sum_{t=1}^T f_t(u^*) = \sum_{t=1}^T \kappa_t(a\beta^* + \frac{b\|\phi^* - w_t\|^2 + \epsilon_t + \epsilon_0}{\beta^*} + \Delta_t)$$

$$= (2\sqrt{a(bV^2 + \epsilon + \epsilon_0)}) \sum_{t=1}^T \kappa_t + \sum_{t=1}^T \kappa_t \Delta_t .$$

Finally, we obtain:

$$
\begin{aligned}
\mathrm{AER}_{\mathcal{A}}^T &\leq \frac{1}{T} \sum_{t=1}^{T} f_t(u_t) \\
&\leq \frac{1}{T} (\sum_{t=1}^{T} f_t(u^*) + R_T^{\mathrm{static}}(u^*)) \\
&= (2\sqrt{a(bV^2 + \epsilon + \epsilon_0)}) \sum_{t=1}^{T} \frac{\kappa_t}{T} + \sum_{t=1}^{T} \frac{\kappa_t \Delta_t}{T} \\
&\quad + \frac{3\kappa_{\max}}{2\tilde{\epsilon}_0 \sqrt{T}} \left( \sqrt{(\hat{D}^2 + \frac{b\hat{D}^2 + \epsilon_{\max}}{a})(4ab^2\tilde{\epsilon}_0\hat{D}^2 + a^2(\epsilon_{\max} + b\hat{D}^2)^2)} \right) .
\end{aligned}
$$

2. If $\epsilon_v > bV^2 + \epsilon$, $\beta^* = \sqrt{(bV^2 + \epsilon + \epsilon_0)/a} < \sqrt{\tilde{\epsilon}_0/a}$ cannot be attained. We can simply upper bound the cost function by replacing $\epsilon_0$ with $\tilde{\epsilon}_0$ and obtain a newly attainable optimal comparator in hindsight $\tilde{u}^* = (\tilde{\beta}^*, \phi^*)$, where $\tilde{\beta}^* = \sqrt{(bV^2 + \epsilon + \tilde{\epsilon}_0)/a} \in \left[ \sqrt{\tilde{\epsilon}_0/a}, \sqrt{(b\hat{D}^2 + \epsilon + \tilde{\epsilon}_0)/a} \right]$.
Then, we have the accumulated cost:

$$
\begin{aligned}
\sum_{t=1}^{T} f_t(\tilde{u}^*) &= \sum_{t=1}^{T} \kappa_t(a\tilde{\beta}^* + \frac{b\|\phi^* - w_t\|^2 + \epsilon_t + \tilde{\epsilon}_0}{\tilde{\beta}^*} + \Delta_t) \\
&= (2\sqrt{a(bV^2 + \epsilon + \tilde{\epsilon}_0)}) \sum_{t=1}^{T} \kappa_t + \sum_{t=1}^{T} \kappa_t \Delta_t .
\end{aligned}
$$

Finally, we obtain:

$$
\begin{aligned}
\mathrm{AER}_{\mathcal{A}}^T &\leq \frac{1}{T} \sum_{t=1}^{T} f_t(u_t) \\
&\leq \frac{1}{T} (\sum_{t=1}^{T} f_t(\tilde{u}^*) + R_T^{\mathrm{static}}(\tilde{u}^*)) \\
&= (2\sqrt{a(bV^2 + \epsilon + \tilde{\epsilon}_0)}) \sum_{t=1}^{T} \frac{\kappa_t}{T} + \sum_{t=1}^{T} \frac{\kappa_t \Delta_t}{T} \\
&\quad + \frac{3\kappa_{\max}}{2\tilde{\epsilon}_0 \sqrt{T}} \left( \sqrt{(\hat{D}^2 + \frac{b\hat{D}^2 + \epsilon_{\max}}{a})(4ab^2\tilde{\epsilon}_0\hat{D}^2 + a^2(\epsilon_{\max} + b\hat{D}^2)^2)} \right) .
\end{aligned}
$$

$\square$

## D.2 AER bound in possibly shifting environments (Proof of Theorem 5.1)

**Theorem D.2.** *Consider both **static** and **shifting** environments. If the excess risk's upper bound of the base learner $\mathcal{A}(u_t, S_t)$ can be formulated as a unified form in Eq.(1), then, the **AER** of **DCML** (in Algo. 1) is upper bounded by:*

$$AER_{\mathcal{A}}^T \leq \underbrace{\frac{2}{T}\sum_{n=1}^{N}(\sqrt{a(bV_n^2 + \epsilon_n + \epsilon_0)}\kappa_n + \frac{\Delta_n}{2})}_{\text{optimal trade-off in hindsight}} + \underbrace{\frac{3}{2T}\sum_{n=1}^{N}\tilde{D}_n G_n\sqrt{M_n - 1}}_{\text{average regret over slots}} + \underbrace{\frac{\tilde{D}_{max}}{T}\sqrt{2P^*\sum_{n=1}^{N}G_n^2}}_{\text{regret w.r.t environment shift}},$$

*Here, $\kappa_n = \sum_{k=1}^{M_n}\kappa_{n,k}$, $\Delta_n = \sum_{k=1}^{M_n}\kappa_{n,k}\Delta_{n,k}$, where the subscript $n,k$ indicates $k$-th task in $n$-th slot. The optimal meta-parameter in hindsight of $n$-th slot is $u_n^* = (\beta_n^*, \phi_n^*)$ and the path length of $N$ slots is $P^* = \sum_{n=1}^{N-1}\|u_n^* - u_{n+1}^*\| + 1$. $\phi_n^* = \sum_{k=1}^{M_n}\frac{\kappa_{n,k}}{\kappa_n}w_{n,k}$ is the weighted average of the base learner outputs within the slot. $\beta_n^* = \sqrt{(bV_n^2 + \epsilon_n + \epsilon_0)/a}$, where $V_n^2 = \sum_{k=1}^{M_n}\frac{\kappa_{n,k}}{\kappa_n}\|\phi_n^* - w_{n,k}\|_2^2$ is the variance in $n$-th slot and $\epsilon_n = \sum_{k=1}^{M_n}\frac{\kappa_{n,k}}{\kappa_n}\epsilon_{n,k}$. Moreover, $G_n$ is the maximal gradient norm of the cost function, $\tilde{D}_n$ is the diameter of meta-parameters in $n$-th slot, and $\tilde{D}_{max} = \max\{\tilde{D}_n\}_{n=1}^N$.*

*Proof.* A possibly shifting environment can be considered slot-wise static or static, so we need to derive the dynamic regret bound. We use the subscript $n, k$ to indicate the $k$-th task inside the $n$-th slot and the corresponding timestep is $t = \sum_{i=1}^{n-1}M_i + k$. To obtain such a bound, we first use the following result for the meta-parameter updates inside the $n$-th slot:

$$\|u_{n,k+1} - u_n^*\|^2 = \|\Pi_{\mathcal{U}}(u_{n,k} - \gamma_{n,k}\nabla f_{n,k}(u_{n,k})) - u_n^*\| \leq \|u_{n,k} - \gamma_{n,k}\nabla f_{n,k} - u_n^*\|^2$$
$$\leq \|u_{n,k} - u_n^*\|^2 + \gamma_{n,k}^2\|\nabla f_{n,k}\|^2 - 2\gamma_{n,k}\langle\nabla f_{n,k}, (u_{n,k} - u_n^*)\rangle.$$

So we have $\langle\nabla f_{n,k}, (u_{n,k} - u_n^*)\rangle \leq \frac{\|u_{n,k} - u_n^*\|^2 - \|u_{n,k+1} - u_n^*\|^2}{2\gamma_{n,k}} + \frac{\gamma_{n,k}\|\nabla f_{n,k}\|^2}{2}$.

Let $G_n$ be the upper bound of the gradient norm related to the $n$-th slot, *i.e.*, $\max_k\{\|\nabla f_{n,k}\|\} \leq G_n$. Applying the above result and the update rule in Algo. 1, we have the dynamic regret:

$$R_T^{\text{dynamic}}(u_{1:N}^*) = \sum_{n=1}^{N}\sum_{k=1}^{M_n}f_{n,k}(u_{n,k}) - f_{n,k}(u_n^*)$$
$$\leq \sum_{n=1}^{N}\left(\sum_{k=1}^{M_n-1}\nabla f_{n,k}^\top(u_{n,k} - u_n^*) + \nabla f_{n,M_n}^\top(u_{n,M_n} - u_n^*)\right)$$
$$\leq \sum_{n=1}^{N}\sum_{k=1}^{M_n-1}\left[\frac{\|u_{n,k} - u_n^*\|^2 - \|u_{n,k+1} - u_n^*\|^2}{2\gamma_{n,k}} + G_n^2\frac{\gamma_{n,k}}{2}\right]$$
$$+ \sum_{n=1}^{N}\left(\frac{\|u_{n,M_n} - u_n^*\|^2 - \|u_{n+1,1} - u_n^*\|^2}{2\rho} + G_n^2\frac{\rho}{2}\right).$$

The meta-parameter updates that transition between slots adopt the hopping learning rate $\rho$, which reflects the environment change.

Denote

$$T_1 = \sum_{n=1}^{N}\left(\sum_{k=1}^{M_n-1}\left[\frac{\|u_{n,k} - u_n^*\|^2 - \|u_{n,k+1} - u_n^*\|^2}{2\gamma_{n,k}} + G_n^2\frac{\gamma_{n,k}}{2}\right] + G_n^2\frac{\rho}{2}\right)$$

and

$$T_2 = \sum_{n=1}^{N}\frac{\|u_{n,M_n} - u_n^*\|^2 - \|u_{n+1,1} - u_{n+1}^*\|^2 + \|u_{n+1,1} - u_{n+1}^*\|^2 - \|u_{n+1,1} - u_n^*\|^2}{2\rho}.$$

First, let us tackle $T_2$,

$$T_2 = \sum_{n=1}^{N} \frac{\|u_{n,M_n} - u_n^*\|^2 - \|u_{n+1,1} - u_{n+1}^*\|^2}{2\rho} + \sum_{n=1}^{N} \frac{\|u_{n+1}^*\|^2 - \|u_n^*\|^2 + 2u_{n+1,1}^\top(u_n^* - u_{n+1}^*)}{2\rho}$$

$$= \sum_{n=1}^{N-1} \frac{\|u_{n,M_n} - u_n^*\|^2 - \|u_{n+1,1} - u_{n+1}^*\|^2}{2\rho}$$

$$+ \sum_{n=1}^{N-1} \frac{\|u_{n+1}^*\|^2 - \|u_n^*\|^2 + 2u_{n+1,1}^\top(u_n^* - u_{n+1}^*)}{2\rho}$$

$$+ \frac{\|u_{N,M_N} - u_N^*\|^2 - \|u_{N+1,1} - u_{N+1}^*\|^2 + \|u_{N+1}^*\|^2 - \|u_N^*\|^2 + 2u_{N+1,1}^\top(u_N^* - u_{N+1}^*)}{2\rho}$$

$$\leq \sum_{n=1}^{N-1} \frac{\|u_{n,M_n} - u_n^*\|^2 - \|u_{n+1,1} - u_{n+1}^*\|^2}{2\rho} + \frac{\|u_N^*\|^2 - \|u_1^*\|^2}{2\rho} + \frac{\tilde{D}_{\max}}{\rho} \sum_{n=1}^{N-1} \|u_n^* - u_{n+1}^*\|$$

$$+ \frac{\|u_{N,M_N} - u_N^*\|^2 - \|u_{N+1,1}\|^2 + 2u_{N+1,1}^\top u_N^* - \|u_N^*\|^2}{2\rho}$$

$$\leq \sum_{n=1}^{N-1} \frac{\|u_{n,M_n} - u_n^*\|^2 - \|u_{n+1,1} - u_{n+1}^*\|^2}{2\rho} + \frac{\|u_N^*\|^2 - \|u_1^*\|^2}{2\rho} + \frac{\tilde{D}_{\max}}{\rho} \sum_{n=1}^{N-1} \|u_n^* - u_{n+1}^*\|$$

$$+ \frac{\|u_{N,M_N} - u_N^*\|^2}{2\rho} \,.$$

The first inequality is obtained with

$$u_{n+1,1}^\top(u_n^* - u_{n+1}^*) \leq \|u_{n+1,1}\|\|u_n^* - u_{n+1}^*\| \leq \tilde{D}_{\max}\|u_n^* - u_{n+1}^*\|\,,$$

where $\tilde{D}_{\max} = \max\{\tilde{D}_n\}_{n=1}^{N}$ is the maximal slot-wise diameter of the set of meta-parameters. Since we have assumed that $\mathbf{0} \in \mathcal{W}$, it gives $\|u_{n+1,1}\| \leq \tilde{D}_{\max}, \forall n$. The last inequality of the above proof holds because $\frac{-\|u_{N+1,1}\|^2 + 2u_{N+1,1}^\top u_N^* - \|u_N^*\|^2}{2\rho} = \frac{-\|u_{N+1,1} - u_N^*\|^2}{2\rho} \leq 0$.

Let $\hat{D}_n$ be the diameter of the base learner outputs in $n$-th slot, and let $\hat{D}_{\max} = \max\{\hat{D}_n\}_{n=1}^{N}$, $\kappa_n^{\max} = \max\{\kappa_{n,k}\}_{k=1}^{M_n}$, $\epsilon_n^{\max} = \max\{\epsilon_{n,k}\}_{k=1}^{M_n}$, $\epsilon_{\max} = \max\{\epsilon_n^{\max}\}_{n=1}^{N}$.

We constrain $\beta_{n,k} \in \left[\sqrt{\tilde{\epsilon}_0/a}, \sqrt{(b\hat{D}_n^2 + \epsilon_{\max} + \tilde{\epsilon}_0)/a}\right]$, where $\tilde{\epsilon}_0 = \epsilon_0 + \epsilon_v$ is related to the randomness of the base learner $\epsilon_0$ and a factor $\epsilon_v$ to control the range of $\beta_t$. We will see later that $\epsilon_v$ can affect the upper bound of the gradient norm of the meta-parameter. We can further verify that $\frac{2b\gamma_{n,k}\kappa_{n,k}}{\beta_{n,k}} \leq 1$ can always hold with $\tilde{\epsilon}_0 < \frac{a^2(\epsilon_{\max} + b\hat{D}_n^2)}{4b^2}$. Hence, $\phi_{n,k}$ is always inside the smallest ball that contains $\hat{\mathcal{W}}_n$. Then, we can define $\tilde{D}_n^2 \stackrel{\text{def}}{=} \hat{D}_n^2 + \frac{b\hat{D}_n^2 + \epsilon_n^{\max} + \tilde{\epsilon}_0}{a}$. Therefore, we have

$$\tilde{D}_{\max}^2 = (1 + b/a)\hat{D}_{\max}^2 + (\epsilon_{\max} + \tilde{\epsilon}_0)/a\,.$$

Now we tackle $T_1$. Since we scale the learning rate of the meta-parameter for shifting environments, we need to set $\rho > \gamma_{n,k}$ for better tracking of the change. Then, we note that if we define $\frac{1}{\gamma_{n,0}} = 0$, we have

$$\sum_{k=1}^{M_n-1} \frac{\|u_{n,k+1} - u_n^*\|^2}{2\gamma_{n,k}} = \sum_{k=0}^{M_n-1} \frac{\|u_{n,k+1} - u_n^*\|^2}{2\gamma_{n,k}} = \sum_{k=0}^{M_n-2} \frac{\|u_{n,k+1} - u_n^*\|^2}{2\gamma_{n,k}} + \frac{\|u_{n,M_n} - u_n^*\|^2}{2\gamma_{n,M_n-1}}$$

$$\geq \sum_{k=0}^{M_n-2} \frac{\|u_{n,k+1} - u_n^*\|^2}{2\gamma_{n,k}} + \frac{\|u_{n,M_n} - u_n^*\|^2}{2\rho}$$

$$= \sum_{k=1}^{M_n-1} \frac{\|u_{n,k} - u_n^*\|^2}{2\gamma_{n,k-1}} + \frac{\|u_{n,M_n} - u_n^*\|^2}{2\rho}\,.$$

In addition, we have $\|u_{n,k} - u_n^*\|^2 = \|\phi_{n,k} - \hat{\phi}_n^*\|^2 + |\beta_{n,k} - \beta_n^*|^2 \leq \hat{D}_n^2 + \frac{b\hat{D}_n^2 + \epsilon_n^{\max}}{a} \leq \tilde{D}_n^2$.

So, we can bound the dynamic regret as

$$R_T^{\text{dynamic}}(u_{1:N}^*) \leq T_1 + T_2$$

$$\leq \sum_{n=1}^{N} \left( \sum_{k=1}^{M_n-1} \left[ \frac{\|u_{n,k} - u_n^*\|^2}{2\gamma_{n,k}} - \frac{\|u_{n,k} - u_n^*\|^2}{2\gamma_{n,k-1}} + G_n^2 \frac{\gamma_{n,k}}{2} \right] - \frac{\|u_{n,M_n} - u_n^*\|^2}{2\rho} + G_n^2 \frac{\rho}{2} \right)$$

$$+ \sum_{n=1}^{N-1} \frac{\|u_{n,M_n} - u_n^*\|^2 - \|u_{n+1,1} - u_{n+1}^*\|^2}{2\rho}$$

$$+ \frac{\|u_N^*\|^2 - \|u_1^*\|^2}{2\rho} + \frac{\tilde{D}_{\max}}{\rho} \sum_{n=1}^{N-1} \|u_n^* - u_{n+1}^*\| + \frac{\|u_{N,M_N} - u_N^*\|^2}{2\rho}$$

$$= \sum_{n=1}^{N} \left( \sum_{k=1}^{M_n-1} \left[ \frac{\|u_{n,k} - u_n^*\|^2}{2\gamma_{n,k}} - \frac{\|u_{n,k} - u_n^*\|^2}{2\gamma_{n,k-1}} + G_n^2 \frac{\gamma_{n,k}}{2} \right] + G_n^2 \frac{\rho}{2} \right)$$

$$+ \sum_{n=1}^{N} \frac{\|u_{n,M_n} - u_n^*\|^2}{2\rho} + \sum_{n=1}^{N-1} \frac{-\|u_{n+1,1} - u_{n+1}^*\|^2}{2\rho} - \sum_{n=1}^{N} \frac{\|u_{n,M_n} - u_n^*\|^2}{2\rho}$$

$$+ \frac{\|u_N^*\|^2 - \|u_1^*\|^2}{2\rho} + \frac{\tilde{D}_{\max}}{\rho} \sum_{n=1}^{N-1} \|u_n^* - u_{n+1}^*\|$$

$$\leq \sum_{n=1}^{N} \left( \sum_{k=1}^{M_n-1} \tilde{D}_n^2 \left[ \frac{1}{2\gamma_{n,k}} - \frac{1}{2\gamma_{n,k-1}} \right] + G_n^2 \frac{\gamma_{n,k}}{2} + G_n^2 \frac{\rho}{2} \right)$$

$$+ \frac{\tilde{D}_{\max}^2}{\rho} + \frac{\tilde{D}_{\max}^2}{\rho} \sum_{n=1}^{N-1} \|u_n^* - u_{n+1}^*\|.$$

The last inequality is derived with $\|u_N^*\| \leq \tilde{D}_{\max}$ and ignoring the negative terms.

Denote the path length of the reference sequence as $P^* = \sum_{n=1}^{N-1} \|u_n^* - u_{n+1}^*\| + 1$. By setting $\gamma_{n,k} = \frac{\tilde{D}_n}{G_n \sqrt{k}}$ (satisfies $1/\gamma_{n,0} = 0$), $\rho = \sqrt{\frac{2P^* \tilde{D}_{\max}^2}{\sum_{n=1}^{N} G_n^2}}$, and using $\sum_{k=1}^{M_n-1} \frac{1}{\sqrt{k}} \leq 2\sqrt{M_n - 1}$, we have

$$R_T^{\text{dynamic}}(u_{1:N}^*) \leq \sum_{n=1}^{N} \left( \frac{3\tilde{D}_n G_n \sqrt{M_n - 1}}{2} + \frac{G_n^2}{2} \rho \right) + \frac{\tilde{D}_{\max}^2 P^*}{\rho}$$

$$\leq \sum_{n=1}^{N} \frac{3\tilde{D}_n G_n \sqrt{M_n - 1}}{2} + \tilde{D}_{\max} \sqrt{2P^* \sum_{n=1}^{N} G_n^2}.$$

For each slot $n$, the optimal prior in hindsight is $u_n^* = (\phi_n^*, \beta_n^*)$. By solving $\min_u \sum_{k=1}^{M_n} f_{n,k}(u)$, we can get:

$$\phi_n^* = \sum_{k=1}^{M_n} \frac{\kappa_{n,k}}{\sum_{k=1}^{M_n} \kappa_{n,k}} w_{n,k}, \beta_n^* = \sqrt{(bV_n^2 + \epsilon_n + \epsilon_0)/a},$$

where $V_n^2 = \sum_{k=1}^{M_n} \frac{\kappa_{n,k}}{\sum_{k=1}^{M_n} \kappa_{n,k}} \|\phi_n^* - w_{n,k}\|^2$ and $\epsilon_n = \frac{\sum_{k=1}^{M_n} \kappa_{n,k} \epsilon_{n,k}}{\sum_{k=1}^{M_n} \kappa_{n,k}}$.

Thus, the accumulated cost for the slot-wise optimal priors in hindsight:

$$\sum_{n=1}^{N} \sum_{k=1}^{M_n} f_{n,k}(u_n^*) = \sum_{n=1}^{N} \sum_{k=1}^{M_n} \kappa_{n,k} \left( a\beta_n^* + \frac{b\|\phi_n^* - w_{n,k}\|^2 + \epsilon_{n,k} + \epsilon_0}{\beta_n^*} + \Delta_{n,k} \right)$$

$$= \sum_{n=1}^{N} (2\sqrt{a(bV_n^2 + \epsilon_n + \epsilon_0)}) \sum_{k=1}^{M_n} \kappa_{n,k} + \sum_{k=1}^{M_n} \kappa_{n,k} \Delta_{n,k}$$

Finally, we obtain the bound in the main theorem:

$$
\begin{aligned}
\mathrm{AER}_{\mathcal{A}}^{T} &\leq \frac{1}{T}\sum_{t=1}^{T} f_t(u_t) \\
&\leq \frac{1}{T}\left(\sum_{n=1}^{N}\sum_{k=1}^{M_n} f_{n,k}(u_n^*) + R_T^{\mathrm{dynamic}}(u_{1:N}^*)\right) \\
&= \frac{1}{T}\sum_{n=1}^{N}(2\sqrt{a(bV_n^2+\epsilon_n+\epsilon_0)})\sum_{k=1}^{M_n}\kappa_{n,k} + \frac{1}{T}\sum_{n=1}^{N}\sum_{k=1}^{M_n}\kappa_{n,k}\Delta_{n,k} \\
&\quad + \frac{1}{T}\sum_{n=1}^{N}\frac{3\tilde{D}_n G_n\sqrt{M_n-1}}{2} + \frac{1}{T}\tilde{D}_{\max}\sqrt{2P^*\sum_{n=1}^{N}G_n^2}\,.
\end{aligned}
$$

Moreover, by limiting the range of $\beta_{n,k}\in\mathcal{B}\overset{\mathrm{def}}{=}\left[\sqrt{\tilde{\epsilon}_0/a},\sqrt{(b\hat{D}_n^2+\epsilon_{\max}+\tilde{\epsilon}_0)/a}\,\right]$, we have

$$
|\partial_{\beta_{n,k}} f_{n,k}| = |\kappa_{n,k}(a - \frac{b\|\phi_{n,k}-w_{n,k}\|^2+\epsilon_{n,k}+\epsilon_0}{\beta_{n,k}^2})| \leq \frac{a\kappa_n^{\max}(b\hat{D}_n^2+\epsilon_n^{\max})}{\tilde{\epsilon}_0}\,,
$$

$$
|\partial_{\phi_{n,k}} f_{n,k}| = \frac{2b\kappa_{n,k}\|\phi_{n,k}-w_{n,k}\|}{\beta_{n,k}} \leq \frac{2b\kappa_n^{\max}\hat{D}_n}{\sqrt{\tilde{\epsilon}_0/a}}\,.
$$

Therefore, $\|\nabla f_{n,k}\|^2 \leq \frac{a^2(\kappa_n^{\max})^2(b\hat{D}_n^2+\epsilon_n^{\max})^2}{\tilde{\epsilon}_0^2} + \frac{4ab^2(\kappa_n^{\max})^2\hat{D}_n^2}{\tilde{\epsilon}_0}, \forall k\in[M_n]$.

Moreover, we let $G_n^2 = \frac{a^2(\kappa_n^{\max})^2(b\hat{D}_n^2+\epsilon_n^{\max})^2}{\tilde{\epsilon}_0^2} + \frac{4ab^2(\kappa_n^{\max})^2\hat{D}_n^2}{\tilde{\epsilon}_0}$, replace $\tilde{D}_n$ with the tighter $\sqrt{\hat{D}_n^2 + \frac{b\hat{D}_n^2+\epsilon_n^{\max}}{a}}$ and use $\tilde{D}_{\max}^2 = (1+b/a)\hat{D}_{\max}^2 + (\epsilon_{\max}+\tilde{\epsilon}_0)/a$.

Analogous to the proof of static AER, if $\epsilon_v \leq bV_n^2+\epsilon_n$, $\beta_n^* = \sqrt{(bV_n^2+\epsilon_n+\epsilon_0)/a}\in\mathcal{B}$ can be attained. Then, the above AER can be alternately bounded as:

$$
\begin{aligned}
\mathrm{AER}_{\mathcal{A}}^{T} &\leq \frac{1}{T}\sum_{n=1}^{N}(2\sqrt{a(bV_n^2+\epsilon_n+\epsilon_0)})\sum_{k=1}^{M_n}\kappa_{n,k} + \frac{1}{T}\sum_{n=1}^{N}\sum_{k=1}^{M_n}\kappa_{n,k}\Delta_{n,k} \\
&\quad + \frac{3}{2\tilde{\epsilon}_0 T}\sum_{n=1}^{N}\kappa_n^{\max}\left(\sqrt{(\hat{D}_n^2+\frac{b\hat{D}_n^2+\epsilon_n^{\max}}{a})(M_n-1)}\sqrt{a^2(b\hat{D}_n^2+\epsilon_n^{\max})^2+4ab^2\tilde{\epsilon}_0\hat{D}_n^2}\right) \\
&\quad + \frac{1}{T}\sqrt{\frac{(a+b)\hat{D}_{\max}^2+\epsilon_{\max}+\tilde{\epsilon}_0}{a}}\sqrt{2P^*\left[\sum_{n=1}^{N}\frac{(a^2(b\hat{D}_n^2+\epsilon_n^{\max})^2+4ab^2\tilde{\epsilon}_0\hat{D}_n^2)(\kappa_n^{\max})^2}{\tilde{\epsilon}_0^2}\right]}\,.
\end{aligned}
$$

Otherwise, if $\epsilon_v > bV_n^2+\epsilon_n$, $\beta_n^* = \sqrt{(bV_n^2+\epsilon_n+\epsilon_0)/a} < \sqrt{\tilde{\epsilon}_0/a}$ cannot be attained by optimizing within $\mathcal{B}$. We can simply upper bound the cost function by replacing $\epsilon_0$ with $\tilde{\epsilon}_0$ and obtain newly attainable optimal comparators in hindsight $\tilde{u}_n^* = (\tilde{\beta}_n^*,\phi_n^*)$, where $\tilde{\beta}_n^* = \sqrt{(bV_n^2+\epsilon_n+\tilde{\epsilon}_0)/a}\in\left[\sqrt{\tilde{\epsilon}_0/a},\sqrt{(b\hat{D}^2+\epsilon_{\max}+\tilde{\epsilon}_0)/a}\,\right]$. Finally, we can obtain a bound for this case by replacing $\epsilon_0$ in the first line of the above dynamic AER bound with $\tilde{\epsilon}_0$. $\qquad\square$

# E Missing Proofs in Section 6

## E.1 Cost function for Gibbs base learner

**Proposition E.1.** *Suppose $\mathcal{W} \subset \mathbb{R}^d$, let $w_t^*$ be the hypothesis that achieves the minimum population risk among $\mathcal{W}$. Suppose $\ell \in [0, 1]$, and $\ell(\cdot, z)$ is $L$-Lipschitz $\forall z \in \mathcal{Z}$. Assume that $\mathcal{L}_{\mu_t}(\cdot)$ has $\alpha$-quadratic-growth for all $t \in [T]$. Let $w_t = \mathcal{A}(\phi_t, \beta_t)$ denote the output of the Gibbs algorithm applied on dataset $S_t$, and $Q_t$ be the prior distribution and $\beta_t$ the inverse temperature of $t$-th task. Assume $Q_t$ is a gaussian distribution $\mathcal{N}(\phi_t, \sigma_t^2 \mathbf{1}_d)$ with $\sigma_t = m_t^{-1/4} d^{-1/4} L^{-1/2}$. To meet the general form of the cost function, we replace $\beta_t$ with $\beta_t'$ by $\alpha\beta_t - 2\sqrt{dm_t}L = m_t^{\frac{3}{4}} d^{\frac{1}{4}} \sqrt{L}\beta_t'$. Hence, the upper bound of the excess risk of the Gibbs algorithm is given by*

$$f_t(\beta_t', \phi_t) = \left( \frac{d^{\frac{1}{4}}\sqrt{L}}{m_t^{\frac{1}{4}}} \right) \left( \frac{\beta_t'}{2\alpha} + 1 + \frac{2d^{\frac{1}{4}}\sqrt{L}}{\alpha m_t^{\frac{1}{4}}} + \frac{2\alpha\|\phi_t - w_t\|^2 + \epsilon_0 + \frac{2}{\alpha}\sqrt{\frac{d}{m_t}}L + 2\frac{\sqrt{L}d^{\frac{1}{4}}}{m_t^{\frac{1}{4}}}}{\beta_t'} \right).$$

*Proof.* From Theorem B.3, we have:

$$R_{\text{excess}}(\text{Gibbs}, u_t) \leq \frac{\beta_t}{2m_t} + m_t^{-1/4}L^{1/2}d^{1/4} + \frac{d^{1/2}m_t^{1/2}L}{2\beta_t}\|\phi_t - w_t^*\|^2.$$

Let $\bar{w}_t = \mathbb{E}W_t$ be the expected output of the Gibbs algorithm, and let $\epsilon_0 = 2\alpha\|w_t - \bar{w}_t\|^2$ be the randomness of the algorithm, which should be a small. Then, using Titu's lemma and the $\alpha$-quadratic growth assumption on the loss function, we get

$$\begin{aligned}
\|\phi_t - w_t^*\|^2 &= \|\phi_t - \bar{w}_t + \bar{w}_t - w_t^*\|^2 \\
&\leq 2\|\phi_t - \bar{w}_t\|^2 + 2\|\bar{w}_t - w_t^*\|^2 \\
&\leq 2\|\phi_t - \bar{w}_t\|^2 + 2\|\mathbb{E}W_t - w_t^*\|^2 \\
&\leq 2\|\phi_t - \bar{w}_t\|^2 + 2\mathbb{E}\|W_t - w_t^*\|^2 \\
&\leq 2\|\phi_t - \bar{w}_t\|^2 + \frac{4\mathbb{E}[\mathcal{L}_{\mu_t}(W_t) - \mathcal{L}_{\mu_t}(w_t^*)]}{\alpha} \\
&\leq 2\|\phi_t - \bar{w}_t\|^2 + \frac{4(\frac{\beta_t}{2m_t} + m_t^{-1/4}L^{1/2}d^{1/4} + \frac{d^{1/2}m_t^{1/2}L}{2\beta_t}\|\phi_t - w_t^*\|^2)}{\alpha}.
\end{aligned}$$

Then, we have $\|\phi_t - w_t^*\|^2 \leq \dfrac{2\alpha\|\phi_t - \bar{w}_t\|^2 + \frac{2\beta_t}{m_t} + 4m_t^{-1/4}L^{1/2}d^{1/4}}{\alpha - \frac{2d^{1/2}m_t^{1/2}L}{\beta_t}}$.

Consequently, we can obtain:

$$R_{\text{excess}}(\text{Gibbs}, u_t) \leq \frac{\beta_t}{2m_t} + m_t^{-1/4}L^{1/2}d^{1/4} + \frac{d^{1/2}m_t^{1/2}L(\alpha\|\phi_t - \bar{w}_t\|^2 + \frac{\beta_t}{m_t} + 2m_t^{-1/4}L^{1/2}d^{1/4})}{\alpha\beta_t - 2d^{1/2}m_t^{1/2}L}.$$

Replace $\alpha\beta_t - 2d^{1/2}m_t^{1/2}L = m_t^{3/4}d^{1/4}L^{1/2}\beta_t'$, we have

$$R_{\text{excess}}(\text{Gibbs}, u_t) \leq \frac{(m_t^{3/4} d^{1/4} L^{1/2} \beta_t' + 2 d^{1/2} m_t^{1/2} L)/\alpha}{2 m_t} + m_t^{-1/4} L^{1/2} d^{1/4}$$

$$+ \frac{\alpha \|\phi_t - \bar{w}_t\|^2 + \frac{(m_t^{3/4} d^{1/4} L^{1/2} \beta_t' + 2 d^{1/2} m_t^{1/2} L)/\alpha}{m_t} + 2 m_t^{-1/4} L^{1/2} d^{1/4}}{m_t^{1/4} d^{-1/4} L^{-1/2} \beta_t'}$$

$$= \frac{1}{2\alpha} m_t^{-1/4} d^{1/4} L^{1/2} \beta_t' + \frac{1}{\alpha} m_t^{-1/2} d^{1/2} L + m_t^{-1/4} L^{1/2} d^{1/4}$$

$$+ \frac{\alpha \|\phi_t - \bar{w}_t\|^2 + \frac{1}{\alpha} m_t^{-1/4} d^{1/4} L^{1/2} \beta_t' + \frac{2}{\alpha} m_t^{-1/2} d^{1/2} L + 2 m_t^{-1/4} L^{1/2} d^{1/4}}{m_t^{1/4} d^{-1/4} L^{-1/2} \beta_t'}$$

$$= \frac{1}{2\alpha} m_t^{-1/4} d^{1/4} L^{1/2} \beta_t' + m_t^{-1/4} L^{1/2} d^{1/4} + \frac{2}{\alpha} m_t^{-1/2} d^{1/2} L$$

$$+ \frac{\alpha \|\phi_t - \bar{w}_t\|^2 + \frac{2}{\alpha} m_t^{-1/2} d^{1/2} L + 2 m_t^{-1/4} L^{1/2} d^{1/4}}{m_t^{1/4} d^{-1/4} L^{-1/2} \beta_t'}$$

$$= \frac{d^{\frac{1}{4}} \sqrt{L}}{m_t^{\frac{1}{4}}} \left( \frac{\beta_t'}{2\alpha} + 1 + \frac{2}{\alpha} m_t^{-1/4} d^{1/4} L^{1/2} \right.$$

$$\left. + \frac{\alpha \|\phi_t - \bar{w}_t\|^2 + \frac{2}{\alpha} m_t^{-1/2} d^{1/2} L + 2 m_t^{-1/4} L^{1/2} d^{1/4}}{\beta_t'} \right)$$

$$\leq \frac{d^{\frac{1}{4}} \sqrt{L}}{m_t^{\frac{1}{4}}} \left( \frac{\beta_t'}{2\alpha} + 1 + \frac{2 d^{\frac{1}{4}} \sqrt{L}}{\alpha m_t^{\frac{1}{4}}} + \frac{\alpha \|\phi_t - \bar{w}_t\|^2 + \frac{2}{\alpha} \sqrt{\frac{d}{m_t}} L + 2 \frac{\sqrt{L} d^{\frac{1}{4}}}{m_t^{\frac{1}{4}}}}{\beta_t'} \right)$$

$$\leq \frac{d^{\frac{1}{4}} \sqrt{L}}{m_t^{\frac{1}{4}}} \left( \frac{\beta_t'}{2\alpha} + 1 + \frac{2 d^{\frac{1}{4}} \sqrt{L}}{\alpha m_t^{\frac{1}{4}}} + \frac{2\alpha \|\phi_t - w_t\|^2 + \epsilon_0 + \frac{2}{\alpha} \sqrt{\frac{d}{m_t}} L + 2 \frac{\sqrt{L} d^{\frac{1}{4}}}{m_t^{\frac{1}{4}}}}{\beta_t'} \right)$$

$$= f_t(\beta_t', \phi_t).$$

$\square$

## E.2 Static AER of Gibbs

**Theorem E.2.** *Consider the Gibbs algorithm as described in Proposition E.1 to be the base learner. Apply the **Meta-OGD** (Algo. 2) on the cost function $f_t(\beta'_t, \phi_t)$. Further, assume that each task uses the same sample number $m$. Then*

$$AER^T_{Gibbs} \in \mathcal{O}\left(\left((V+1) + \frac{1}{\sqrt{T}}\right)\frac{1}{m^{\frac{1}{4}}}\right).$$

*Proof.* From Lemma B.6, we know that $f_t$ is convex w.r.t $u$, so we can directly apply **OGD**. Following Proposition E.1, we have

$$f_t(\beta'_t, \phi_t) = \left(\frac{d^{\frac{1}{4}}\sqrt{L}}{m_t^{\frac{1}{4}}}\right)\left(\frac{\beta'_t}{2\alpha} + 1 + \frac{2d^{\frac{1}{4}}\sqrt{L}}{\alpha m_t^{\frac{1}{4}}} + \frac{2\alpha\|\phi_t - w_t\|^2 + \epsilon_0 + \frac{2}{\alpha}\sqrt{\frac{d}{m_t}}L + 2\frac{\sqrt{L}d^{\frac{1}{4}}}{m_t^{\frac{1}{4}}}}{\beta'_t}\right).$$

To apply Theorem D.1, we first find $a = \frac{1}{2\alpha}$, $b = 2\alpha$, $\epsilon_t = \frac{2}{\alpha}\sqrt{\frac{d}{m_t}}L + 2\frac{\sqrt{L}d^{\frac{1}{4}}}{m_t^{\frac{1}{4}}}$, $\Delta_t = 1 + \frac{2d^{\frac{1}{4}}\sqrt{L}}{\alpha m_t^{\frac{1}{4}}}$, $\epsilon_0 = 2\alpha\|w_t - \bar{w}_t\|^2$, and $\kappa_t = \frac{d^{\frac{1}{4}}\sqrt{L}}{m_t^{\frac{1}{4}}}$. Since we have assumed $m_t = m, \forall t \in [T]$, then we have $\kappa = \kappa_t = \kappa_{\max} = \frac{d^{1/4}\sqrt{L}}{m^{1/4}}$, $\epsilon = \epsilon_t = \epsilon_{\max} = \frac{2}{\alpha}\sqrt{\frac{d}{m}}L + 2\frac{\sqrt{L}d^{\frac{1}{4}}}{m^{\frac{1}{4}}}$.

Taking the above values into Theorem D.1, we have:

$$AER^T_{Gibbs} \leq (\sqrt{2(2\alpha V^2 + \epsilon + \epsilon_0)/\alpha} + 1)\sum_{t=1}^{T}\frac{d^{1/4}\sqrt{L}}{Tm^{1/4}} + \sum_{t=1}^{T}\frac{2\sqrt{d}L}{T\alpha\sqrt{m}}$$

$$+ \frac{3d^{1/4}\sqrt{L}}{2\tilde{\epsilon}_0\sqrt{T}m^{1/4}}\sqrt{((1+4\alpha^2)\hat{D}^2 + 4\sqrt{d/m}L + \frac{4\alpha\sqrt{L}d^{1/4}}{m^{1/4}})}$$

$$\times \sqrt{8\alpha\tilde{\epsilon}_0\hat{D}^2 + (\frac{1}{\alpha^2}\sqrt{\frac{d}{m}}L + \frac{\sqrt{L}d^{\frac{1}{4}}}{\alpha m^{\frac{1}{4}}} + \hat{D}^2)^2}.$$

Since we have assumed the bounded gradient norm for the cost function, *i.e.*, $L'$-Lipschitz, then we can obtain:

$$\hat{D}^2 \leq \frac{\tilde{\epsilon}_0 L' - a\kappa\epsilon}{ab\kappa},$$

which implies $\tilde{\epsilon}_0 \geq a\kappa\epsilon/L'$, where $a\kappa\epsilon/L' \in \mathcal{O}(1/\sqrt{m})$. So it suffices to ensure $\tilde{\epsilon}_0 > \Omega(1/\sqrt{m})$.

For the Gibbs algorithm, by the law of large numbers, we have $\epsilon_0 = 2\alpha\|w_t - \mathbb{E}_{S_t,\mathcal{A}}W_t\|^2 \to 2\alpha\|w_t - \mathbb{E}_{\mathcal{A}}W_t\|^2$ as $m \to \infty$. In addition, since the variance of Gaussian is $\sigma = m^{-1/4}d^{-1/4}L^{-1/2}$, we have $\epsilon_0 \in o_m(1)$.

So if we set $\epsilon_v = bV^2 < bV^2 + \epsilon$ with constant $V$, then $\tilde{\epsilon}_0 = \epsilon_0 + \epsilon_v$ satisfies the above condition. Consequently, we have

$$AER^T_{Gibbs} \in \mathcal{O}\left(\left((V+1) + \frac{1}{\sqrt{T}}\right)\frac{1}{m^{\frac{1}{4}}}\right).$$

$\square$

### E.3 Proof of Theorem 6.1 (Dynamic AER of Gibbs)

**Theorem E.3.** *Let $\mathcal{W} \subset \mathbb{R}^d$. Assume that $\ell(\cdot, z) \in [0, 1]$ is $L$-Lipschitz $\forall z \in \mathcal{Z}$. Assume that $\mathcal{L}_{\mu_t}(\cdot)$ has $\alpha$-quadratic-growth for all $t \in [T]$. Let $w_t = \mathcal{A}_{Gibbs}(\beta_t, \phi_t, S_t)$ be the output of the Gibbs algorithm (Definition B.2) on $S_t$, where $\phi_t$ is the mean of a prior Gaussian $\mathcal{N}(\phi_t, \sigma_t^2 \mathbf{1}_d)$ with $\sigma_t = m_t^{-1/4} d^{-1/4} L^{-1/2}$ and $\beta_t$ is the inverse temperature. Consider Algo. 1 (**DCML**) and further assume that each slot has equal length $M$ and each task uses the sample number $m$. Then*

$$AER_{Gibbs}^T \in \mathcal{O}\left( \left( (1 + \frac{1}{N} \sum_{n=1}^{N} V_n + \frac{1}{\sqrt{M}}) + \frac{\sqrt{P^*}}{M\sqrt{N}} \right) \frac{1}{m^{\frac{1}{4}}} \right).$$

*Proof.* From Proposition E.1, we first obtain $a = \frac{1}{2\alpha}$, $b = 2\alpha$, $\epsilon_{n,k} = \frac{2}{\alpha}\sqrt{\frac{d}{m_{n,k}}} L + 2\frac{\sqrt{L}d^{\frac{1}{4}}}{m_{n,k}^{\frac{1}{4}}}$,

$\Delta_{n,k} = 1 + \frac{2d^{\frac{1}{4}}\sqrt{L}}{\alpha m_{n,k}^{\frac{1}{4}}}$, and $\kappa_{n,k} = \frac{d^{\frac{1}{4}}\sqrt{L}}{m_{n,k}^{\frac{1}{4}}}$. Moreover, $\kappa_n^{\max} = \frac{d^{1/4}\sqrt{L}}{m_n^{*\,1/4}}$, $\epsilon_n^{\max} = \frac{2}{\alpha}\sqrt{\frac{d}{m_n^*}} L + 2\frac{\sqrt{L}d^{\frac{1}{4}}}{m_n^{*\,\frac{1}{4}}}$,

where $m_n^* = \min\{m_{n,k}\}_{k=1}^{M_n}$.

For simplicity, let us assume that each environment slot has the same number of tasks, *i.e.*, $M_n = M$, $T = NM$, and each task has the same number of samples, *i.e.*, $m_{n,k} = m$. So we have $\kappa_{n,k} = \kappa_n^{\max} = \kappa = d^{1/4}L^{1/2}m^{-1/4}$ and $\epsilon_{n,k} = \epsilon_n^{\max} = \epsilon = \frac{2}{\alpha}\sqrt{d/m}L + 2\sqrt{L}d^{1/4}m^{-1/4}$.

Taking these values into Theorem 5.1, we can obtain:

$$
\begin{aligned}
AER_{Gibbs}^T \leq & \frac{1}{N} \sum_{n=1}^{N} (\sqrt{2(2\alpha V_n^2 + \epsilon_n + \epsilon_0)/\alpha}) \frac{d^{\frac{1}{4}}\sqrt{L}}{m^{\frac{1}{4}}} + \frac{d^{\frac{1}{4}}\sqrt{L}}{m^{\frac{1}{4}}} + \frac{2\sqrt{d}L}{\alpha\sqrt{m}} \\
& + \frac{3}{2\tilde{\epsilon}_0 NM} \frac{d^{\frac{1}{4}}\sqrt{L}}{m^{\frac{1}{4}}} \sqrt{M-1} \times \\
& \sum_{n=1}^{N} \sqrt{(1 + 4\alpha^2)\hat{D}_n^2 + 4\sqrt{\frac{d}{m}}L + \frac{4\alpha\sqrt{L}d^{\frac{1}{4}}}{m^{\frac{1}{4}}}} \sqrt{(\hat{D}_n^2 + \sqrt{\frac{d}{m}}L\frac{1}{\alpha^2} + \frac{\sqrt{L}d^{\frac{1}{4}}}{\alpha m^{\frac{1}{4}}})^2 + 8\alpha\tilde{\epsilon}_0 \hat{D}_n^2} \\
& + \frac{1}{NM} \sqrt{\tilde{\epsilon}_0(1 + 4\alpha^2)\hat{D}_{\max}^2 + 4\sqrt{\frac{d}{m}}L + \frac{4\alpha\sqrt{L}d^{1/4}}{m^{1/4}}} \times \frac{d^{1/4}\sqrt{L}}{m^{\frac{1}{4}}} \\
& \times \sqrt{2P^* \sum_{n=1}^{N} \frac{(\hat{D}_n^2 + \frac{1}{\alpha^2}\sqrt{d/m}L + \frac{\sqrt{L}d^{\frac{1}{4}}}{\alpha m^{\frac{1}{4}}})^2}{\tilde{\epsilon}_0^2} + \frac{8\alpha\hat{D}_n^2}{\tilde{\epsilon}_0}}.
\end{aligned}
$$

Since we have assumed the bounded gradient norm for the cost function, *i.e.*, $L'$-Lipschitz, then we can obtain:

$$\hat{D}_n^2 \leq \frac{\tilde{\epsilon}_0 L' - a\kappa\epsilon}{ab\kappa},$$

which implies $\tilde{\epsilon}_0 \geq a\kappa\epsilon/L'$, where $a\kappa\epsilon/L' \in \mathcal{O}(1/\sqrt{m})$. So it suffices to ensure $\tilde{\epsilon}_0 > \Omega(1/\sqrt{m})$.

For the Gibbs algorithm, by the law of large numbers, we have $\epsilon_0 = 2\alpha\|w_t - \mathbb{E}_{S_t,\mathcal{A}} W_t\|^2 \to 2\alpha\|w_t - \mathbb{E}_\mathcal{A} W_t\|^2$ as $m \to \infty$. In addition, since the variance of Gaussian is $\sigma = m^{-1/4}d^{-1/4}L^{-1/2}$, we have $\epsilon_0 \in o_m(1)$. So if we set $\epsilon_v = bV_n^2 < bV_n^2 + \epsilon$ with constant $V_n$, $\tilde{\epsilon}_0 = \epsilon_0 + \epsilon_v$ satisfies the above condition. Consequently, we have

$$AER_{Gibbs}^T \in \mathcal{O}\left( \left( (1 + \frac{1}{N} \sum_{n=1}^{N} V_n + \frac{1}{\sqrt{M}}) + \frac{\sqrt{P^*}}{M\sqrt{N}} \right) \frac{1}{m^{\frac{1}{4}}} \right).$$

$\square$

### E.4 Cost function for SGD base leaner

**Proposition E.4.** *Consider $\mathcal{W} \subset \mathbb{R}^d$. Let $w_t^*$ be the hypothesis that achieves the minimum population risk among $\mathcal{W}$. Suppose $\ell(\cdot, z)$ is convex, $\beta$-smooth, and $L$-Lipschitz for all $z \in \mathcal{Z}$. Assume that $\mathcal{L}_{\mu_t}(\cdot)$ has $\alpha$-quadratic-growth $\forall t \in [T]$. Let $w_t$ denote the output of the SGD algorithm applied on dataset $S_t$ and $\phi_t$ be the initialization of $t$-th task, $\eta_t$ is the learning rate, and $K_t$ is the gradient updates number. To meet the unified form in Eq. (1), let $\eta_t' = (K_t \alpha \eta_t - 2)\kappa_t$. Then, the upper bound of the excess risk is given by*

$$f(\eta_t', \phi_t) = \kappa_t \left( \frac{2\alpha\|\phi_t - w_t\|^2 + \epsilon_0 + 2(L^2 + 2L^2\frac{K_t}{m_t})/(K_t\alpha)}{\eta_t'} + \frac{1}{2}\eta_t' + 2\kappa_t \right),$$

*where $\kappa_t = \sqrt{(1/(K_t\alpha) + 2/(m_t\alpha)}L$.*

From Theorem .B.4, we have the following excess risk bound for SGD:

*Proof.*

$$R_{\text{excess}}(\text{SGD}, u_t) \leq \frac{\|\phi_t - w_t^*\|^2}{2\eta_t K_t} + (\frac{L^2}{2} + \frac{L^2 K_t}{m_t})\eta_t.$$

Let $\bar{w}_t = \mathbb{E}W_t$ be the expected output of the SGD algorithm of $t$-th task, and $w_t$ is the actual output. Let $\epsilon_0 = 2\alpha\|w_t - \bar{w}_t\|^2$ be the randomness of the algorithm, which should be small. With the quadratic growth property, we have

$$
\begin{aligned}
\|\phi_t - w_t^*\|^2 &= \|\phi_t - \bar{w}_t + \bar{w}_t - w_t^*\|^2 \\
&\leq 2\|\phi_t - \bar{w}_t\|^2 + 2\|\bar{w}_t - w_t^*\|^2 \\
&\leq 2\|\phi_t - \bar{w}_t\|^2 + 2\mathbb{E}\|W_t - w_t^*\|^2 \\
&\leq 2\|\phi_t - \bar{w}_t\|^2 + \frac{4\mathbb{E}[\mathcal{L}_{\mu_t}(W_t) - \mathcal{L}_{\mu_t}(w_t^*)]}{\alpha} \\
&\leq 2\|\phi_t - \bar{w}_t\|^2 + \frac{4(\frac{\|\phi_t - w_t^*\|^2}{2\eta_t K_t} + (\frac{L^2}{2} + \frac{L^2 K_t}{m_t})\eta_t)}{\alpha}.
\end{aligned}
$$

Therefore, we can obtain

$$\|\phi_t - w_t^*\|^2 \leq \frac{2\alpha\|\phi_t - \bar{w}_t\|^2 + (2L^2 + 4L^2\frac{Kt}{m_t})\eta_t}{\alpha - \frac{2}{K_t\eta_t}},$$

$$R_{\text{excess}}(\text{SGD}, u_t) \leq \frac{2\alpha\|\phi_t - w_t\|^2 + \epsilon_0 + (L^2 + 2L^2\frac{K_t}{m_t})\eta_t}{K_t\alpha\eta_t - 2} + (\frac{L^2}{2} + \frac{L^2 K_t}{m_t})\eta_t.$$

Let

$$f(\eta_t, \phi_t) = \frac{2\alpha\|\phi_t - w_t\|^2 + \epsilon_0 + (L^2 + 2L^2\frac{K_t}{m_t})\eta_t}{K_t\alpha\eta_t - 2} + (\frac{L^2}{2} + \frac{L^2 K_t}{m_t})\eta_t,$$

and replace $\eta_t' = (K_t\alpha\eta_t - 2)\kappa_t, \kappa_t = \sqrt{\frac{(L^2 + 2L^2\frac{K_t}{m_t})}{K_t\alpha}}$, we have

$$
\begin{aligned}
f(\eta_t', \phi_t) &= \frac{2\alpha\|\phi_t - w_t\|^2 + \epsilon_0 + (L^2 + 2L^2\frac{K_t}{m_t})(\eta_t'/\kappa_t + 2)/(K_t\alpha)}{\eta_t'/\kappa_t} \\
&\quad + (\frac{L^2}{2} + L^2\frac{K_t}{m_t})(\eta_t'/\kappa_t + 2)/(K_t\alpha) \\
&= \kappa_t \left( \frac{2\alpha\|\phi_t - w_t\|^2 + \epsilon_0 + 2(L^2 + 2L^2\frac{K_t}{m_t})/(K_t\alpha)}{\eta_t'} + \frac{1}{2}\eta_t' + 2\kappa_t \right)
\end{aligned}
$$

$\square$

## E.5 Static AER of SGD

**Theorem E.5.** *Consider the SGD algorithm as described in Proposition E.4 to be the base learner. Apply the **Meta-OGD**(Algo. 2) algorithm on the cost function $f_t(\eta'_t, \phi_t)$. Further, assume that for each task, we have the same sample number $m$ and that SGD performs the same number of gradient steps $K$. Then*

$$AER^T_{SGD} \in \mathcal{O}\left((V + \frac{1}{\sqrt{T}})\sqrt{\frac{1}{K} + \frac{1}{m}}\right).$$

*Proof.* According to Proposition E.4, we have $\kappa_t = \sqrt{\frac{(L^2 + 2L^2\frac{K_t}{m_t})}{K_t\alpha}}$, $a = 1/2$, $b = 2\alpha$ and $\epsilon_t = \frac{2(L^2 + 2L^2\frac{K_t}{m_t})}{K_t\alpha} = 2\kappa_t^2$, $\Delta_t = 2\kappa_t$, $\epsilon_0 = 2\alpha\|w_t - \bar{w}_t\|^2$.

For simplicity, we assume the SGD step and sample number are the same for different tasks. Hence, we have $K_t = K$, $m_t = m$, $\kappa_t = \kappa_{\max} = \sqrt{\frac{1}{K\alpha} + \frac{2}{m\alpha}}L = \kappa$ and $\epsilon_t = \epsilon = \epsilon_{\max} = 2\kappa^2$.

Put these variable into the bound of Theorem D.1, we have:

$$AER^T_{SGD} \le (\sqrt{2(2\alpha V^2 + (\frac{2}{K\alpha} + \frac{4}{m\alpha})L^2 + \epsilon_0)})\sqrt{\frac{1}{K\alpha} + \frac{2}{m\alpha}}L + 2(\frac{1}{K\alpha} + \frac{2}{m\alpha})L^2$$

$$+ \frac{3}{2\tilde{\epsilon}_0\sqrt{T}}\sqrt{\frac{1}{K\alpha} + \frac{2}{m\alpha}}L\times$$

$$\left(\sqrt{\left((1 + 4\alpha)\hat{D}^2 + 4(\frac{1}{K\alpha} + \frac{2}{m\alpha})L^2\right)\left(8\alpha^2\tilde{\epsilon}_0\hat{D}^2 + \left((\frac{1}{K\alpha} + \frac{2}{m\alpha})L^2 + \alpha\hat{D}^2\right)^2\right)}\right).$$

Since we have assumed the bounded gradient norm for the cost function, *i.e.*, $L'$-Lipschitz, then we can obtain:

$$\hat{D}^2 \le \frac{\tilde{\epsilon}_0 L' - a\kappa\epsilon}{ab\kappa},$$

which implies $\tilde{\epsilon}_0 \ge a\kappa\epsilon/L'$, where $a\kappa\epsilon/L' \in \mathcal{O}(m^{-3/2} + K^{-3/2})$. So it suffices to ensure $\tilde{\epsilon}_0 \in \Omega(m^{-3/2} + K^{-3/2})$.

For SGD, by the law of large numbers, we have $\epsilon_0 = 2\alpha\|w_t - \mathbb{E}_{S_t, \mathcal{A}}W_t\|^2 \to 2\alpha\|w_t - \mathbb{E}_{\mathcal{A}}W_t\|^2$ as $m \to \infty$. In addition, $2\alpha\|w_t - \mathbb{E}_{\mathcal{A}}W_t\|^2 \to 0$ as $K \to \infty$, so we have $\epsilon_0 \in o_m(1) + o_K(1)$.

Then, if we set $\epsilon_v = bV^2 < bV^2 + \epsilon$, s.t. $\tilde{\epsilon}_0 = \epsilon_0 + \epsilon_v$ satisfies the above condition, we have

$$AER^T_{SGD} \le \mathcal{O}\left((V + o_m(1) + o_K(1) + \frac{\frac{1}{o_m(1) + o_K(1) + V^2}}{\sqrt{T}})\sqrt{\frac{1}{K} + \frac{1}{m}}\right).$$

Considering a constant variance, it gives the following bound that has the same complexity factor and rate as Theorem 3.2 in [31] (Khodak et al. [31] used OGD as the base learner, where each step only takes one sample, *i.e.*, $K = m$).

$$AER^T_{SGD} \in \mathcal{O}\left((V + \frac{1}{\sqrt{T}})\sqrt{\frac{1}{K} + \frac{1}{m}}\right).$$

If we have $V \to 0$, we can set $\epsilon_v = bV^2 + \epsilon + 1/T^{1/4} > bV^2 + \epsilon$, s.t. $\tilde{\epsilon}_0 = \epsilon_0 + \epsilon_v$ satisfies the above condition. In this case, we have:

$$AER^T_{SGD} \le \mathcal{O}\left(\left(V + o_m(1) + o_K(1) + \frac{1}{\sqrt{T}} + \frac{\frac{1}{o_m(1) + o_K(1) + V^2 + 1/T^{1/4}}}{\sqrt{T}}\right)\sqrt{\frac{1}{K} + \frac{1}{m}}\right)$$

$$= \mathcal{O}\left(\left(\frac{1}{\sqrt[4]{T}} + o_m(1) + o_K(1)\right)\sqrt{\frac{1}{K} + \frac{1}{m}}\right), \text{ when } V \to 0,$$

which has an additional term $o_m(1) + o_K(1)$ compared to Theorem 3.2 in [31]. This is because the variance $V^2$ (task similarity) in our paper is calculated via the average SGD iterates. A similar term will be introduced using the same output for estimating $V$ in [31]. □

## E.6 Proof of Theorem 6.2 (Dynamic AER of SGD)

**Theorem E.6.** *Let $\mathcal{W} \subset \mathbb{R}^d$. Consider that $\ell(\cdot, z)$ is convex, $\beta$-smooth, $L$-Lipschitz for all $z \in \mathcal{Z}$. Assume that $\mathcal{L}_{\mu_t}(\cdot)$ has $\alpha$-quadratic-growth for all $t \in [T]$. Let SGD be the base learner for each task where it outputs $w_t = \mathcal{A}_{SGD}(\eta_t, \phi_t, S_t)$ with learning rate $\eta_t$ and initialization $\phi_t$. Consider the Algo. 1 (**DCML**) on the SGD cost function. Further, assume each slot has equal length $M$, each task has the sample number $m$, and that SGD performs the same number of updating steps $K$. Then*

$$AER_{SGD}^T \in \mathcal{O}\left(\left(\frac{1}{N}\sum_{n=1}^{N} V_n + \frac{1}{\sqrt{M}} + \frac{\sqrt{P^*}}{M\sqrt{N}}\right)\sqrt{\frac{1}{K} + \frac{1}{m}}\right).$$

*Proof.* For simplicity, we assume the SGD step and sample number are the same for different tasks. Hence, $K_{n,k} = K$ and $m_{n,k} = m$. Based on Proposition E.4, we have $a = 1/2$, $b = 2\alpha$. Then, we get $\kappa_{n,k} = \sqrt{\frac{1}{K\alpha} + \frac{2}{m\alpha}} L = \kappa$ $\epsilon_{n,k} = \epsilon = 2\kappa^2$, and $\Delta_{n,k} = 2\kappa_{n,k}$.

Put these variable into the bound of Theorem 5.1, we can obtain:

$$AER_{SGD}^T \leq \left(\frac{1}{N}\sum_{n=1}^{N}\sqrt{2(2\alpha V_n^2 + 2(\frac{1}{K\alpha} + \frac{2}{m\alpha})L^2 + \epsilon_0)}\right)\sqrt{\frac{1}{K\alpha} + \frac{2}{m\alpha}}L + 2(\frac{1}{K\alpha} + \frac{2}{m\alpha})L^2$$

$$+ \frac{3}{2\tilde{\epsilon}_0 M}\sqrt{\frac{1}{K\alpha} + \frac{2}{m\alpha}}L\sqrt{M-1}\times$$

$$\frac{1}{N}\sum_{n=1}^{N}\left(\sqrt{(1+4\alpha)\hat{D}_n^2 + (\frac{4}{K\alpha} + \frac{8}{m\alpha})L^2}\sqrt{(\alpha\hat{D}_n^2 + (\frac{1}{K\alpha} + \frac{2}{m\alpha})L^2)^2 + 8\alpha^2\tilde{\epsilon}_0\hat{D}_n^2}\right)$$

$$+ \frac{1}{M}\sqrt{(1+4\alpha)\hat{D}_{\max}^2 + 4L^2(\frac{1}{K\alpha} + \frac{2}{m\alpha}) + 2\tilde{\epsilon}_0} \times \sqrt{\frac{1}{K\alpha} + \frac{2}{m\alpha}}L$$

$$\times \sqrt{\frac{2P^*}{N}\frac{1}{N}\sum_{n=1}^{N}\left[\frac{(\alpha\hat{D}_n^2 + (\frac{1}{K\alpha} + \frac{2}{m\alpha})L^2)^2}{\tilde{\epsilon}_0^2} + \frac{8\alpha^2\hat{D}_n^2}{\tilde{\epsilon}_0}\right]}.$$

Since we have assumed the bounded gradient norm for the cost function, *i.e.*, $L'$-Lipschitz, then we can obtain:

$$\hat{D}_n^2 \leq \frac{\tilde{\epsilon}_0 L' - a\kappa\epsilon}{ab\kappa},$$

which implies $\tilde{\epsilon}_0 \geq a\kappa\epsilon/L'$, where $a\kappa\epsilon/L' \in \mathcal{O}(m^{-3/2} + K^{-3/2})$. So it's sufficient to ensure $\tilde{\epsilon}_0 \in \Omega(m^{-3/2} + K^{-3/2})$.

For SGD, by the law of large numbers, we have $\epsilon_0 = 2\alpha\|w_t - \mathbb{E}_{S_t,\mathcal{A}}W_t\|^2 \to 2\alpha\|w_t - \mathbb{E}_{\mathcal{A}}W_t\|^2$ as $m \to \infty$. In addition, $2\alpha\|w_t - \mathbb{E}_{\mathcal{A}}W_t\|^2 \to 0$ as $K \to \infty$, so we have $\epsilon_0 \in o_m(1) + o_K(1)$.

Then, if we set $\epsilon_v = bV_n^2 < bV_n^2 + \epsilon$ s.t. $\tilde{\epsilon}_0 = \epsilon_0 + \epsilon_v$ satisfies the above condition, we have

$$AER_{SGD}^T \leq \mathcal{O}\left(\left(\frac{1}{N}\sum_{n=1}^{N} V_n + o_m(1) + o_K(1) + \frac{\frac{1}{\epsilon_0+V^2}}{\sqrt{M}} + \frac{\frac{1}{\epsilon_0+V^2}\sqrt{\frac{P^*}{N}}}{M}\right)\sqrt{\frac{1}{K} + \frac{1}{m}}\right).$$

The above bound has additional $o_m(1) + o_T(1)$ compared to Theorems in [31]. Since the variance $V_n^2$ (task similarity) in this paper is calculated via the average SGD iterates. A similar term will be introduced using the same algorithm outputs for estimating $V$ in [31]. Moreover, our bound is in terms of average task excess risk, while [31] upper bounds the average task regret for online base learners using a different meta-learning algorithm.

Considering a constant variance, we have

$$AER_{SGD}^T \in \mathcal{O}\left(\left(\frac{1}{N}\sum_{n=1}^{N} V_n + \frac{1}{\sqrt{M}} + \frac{\sqrt{P^*}}{M\sqrt{N}}\right)\sqrt{\frac{1}{K} + \frac{1}{m}}\right).$$

When we have a large $P^*$ with a small $N$ (the environment occasionally changes with a large shift), the above bound has an improved rate of $\mathcal{O}(1/\sqrt{M})$ or $\sqrt{P^*}$ on the path length $P^*$ (reflecting

environment similarities and shifts) compared to $\mathcal{O}\left(\frac{1}{N}\sum_{n=1}^{N} V_n + \frac{1}{\sqrt{M}} + \min\{\sqrt{\frac{P^*}{MN}}, \frac{P^*}{NM}\}\right)-$ the equivalent form of Theorem 3.3 [31] in this case.

If we have $V_n \to 0$, we can set $\epsilon_v = bV_n^2 + \epsilon + 1/M^{1/4} > bV_n^2 + \epsilon$, s.t. $\tilde{\epsilon}_0 = \epsilon_0 + \epsilon_v$ satisfies the aforementioned condition. In this case, we have:

$$\mathcal{O}\left(\left(\frac{1}{N}\sum_{n=1}^{N} V_n + o_m(1) + o_K(1) + \frac{1 + \frac{1}{\epsilon_0+V^2+1/M^{1/4}}}{\sqrt{M}} + \frac{\frac{1}{\epsilon_0+V^2+1/M^{1/4}}\sqrt{\frac{P^*}{N}}}{M}\right)\sqrt{\frac{1}{K}+\frac{1}{m}}\right)$$

$$= \mathcal{O}\left(\left(\frac{1}{\sqrt[4]{M}}\left(1 + \sqrt{\frac{P^*}{NM}}\right) + o_m(1) + o_K(1)\right)\sqrt{\frac{1}{K}+\frac{1}{m}}\right), \text{ when } V_n \to 0.$$

The above bound has an improvement of rate $\mathcal{O}(1/\sqrt[4]{M})$ on $P^*$ as $V_n \to 0$ compared to the equivalent term $\mathcal{O}\left(\frac{1}{\sqrt[4]{M}} + \sqrt{\frac{P^*}{NM}} + o_m(1) + o_K(1)\right)$ of Theorem 3.3 in [31]. □

# F   Experiments

In this section, we provide experimental settings, some additional experimental results, and experimental details. The code is available in `https://github.com/livreQ/DynamicCML`.

## F.1   Experimental settings

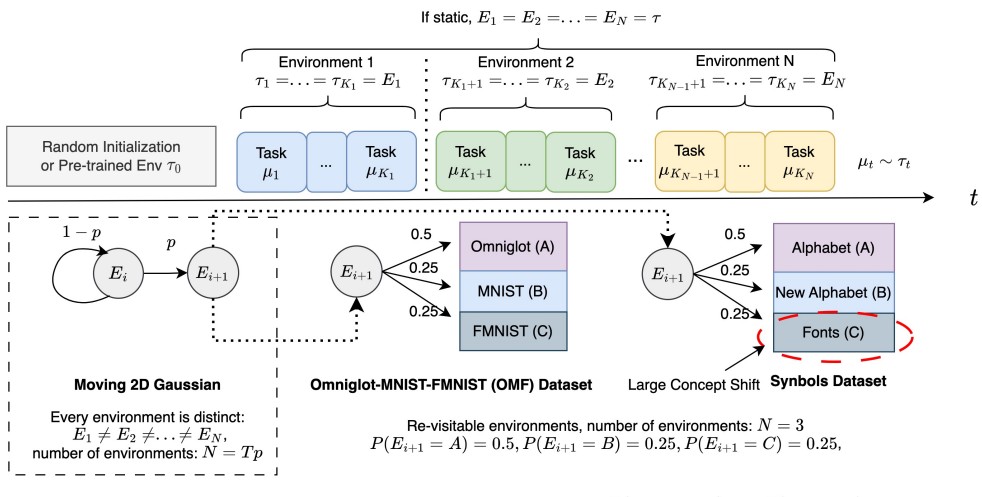

Figure 6: Summary of experimental settings

### F.1.1   Detailed setting in OSAKA

**Task data generating process**   Following OSAKA, we use a pre-trained meta-model as initialization. The datasets in OSAKA contain three environments. One is used for pre-training, and the other two are considered unseen, consisting of OOD tasks compared to the pre-trained environment. We run 10 CML episodes for each method, where each episode has a length of $T = 10000$ time steps.

At each time step, we choose to change to another environment with probability $p$. We select the new environment with a probability of $0.5$ for the pre-trained environment and $0.25$ for one of the other two. Then, a task is sampled from the selected environment. The above generation process is slightly different from the original OSAKA, which mainly considers a task-agnostic setting (*i.e.*, unknown task boundary). They used a single probability for switching the task and environment simultaneously. In contrast, we focus on shifting environments under the task-aware setting, where the task switches with probability $1$, and the environment changes with the process described above. A visualization of the data-generating process is presented in Fig. 6.

**Datasets**   We test DCML on two representative datasets in OSAKA.

- The *Synbols* [43] dataset uses characters from different alphabets on randomized backgrounds as the pre-training environment. A new alphabet and font classification tasks are considered unseen environments. This is a large and complex dataset including a heterogeneous environment (font) with *extremely large concept shift* w.r.t other environments. We conduct a $4$-way $5$-shot classification for this dataset.

- For *Omniglot-MNIST-FashionMNIST*(OMF), we pre-train the meta-model on the first $1000$ classes of Omniglot [44]. Then, at CML time, the models are updated on a stream of tasks generated by the process discussed in the previous section exposed to the full Omniglot dataset and two OOD datasets – MNIST [45] and FashionMNIST [46]. In this study, we conduct $10$-way $5$-shot classification as the benchmark does.

Table 3: Comparison of computational complexities with baselines

| Methods | Computational complexity |
|---------|--------------------------|
| Fine-tuning | $\mathcal{O}(T * K)$ |
| MetaCOG | $\mathcal{O}(T * (K + M))$ |
| MetaBGD | $\mathcal{O}(T * (K + M))$ |
| ANIL | $\mathcal{O}(T * K)$ |
| MAML | $\mathcal{O}(T * K)$ |
| CMAML | $\mathcal{O}(T * (K + 1))$ |
| DCML | $\mathcal{O}(T * (K + 1))$ |

**Baselines** We compared the proposed algorithm with different baselines. For a fair comparison, we further use the same meta-model pre-trained with MAML [4] for all the methods. During the continual learning phase, all the methods adapt with few-shot data and are then evaluated on separate test data for each task.

- *ANIL* is a variation of *MAML* that only adapts the network's head w.r.t new tasks. Following OSAKA, MAML, and ANIL *do not update* the meta-model during the CL phase. so they will not suffer from forgetting but *lack the plasticity* to learn from tasks in new environments. They are compared to show the problem with static representations in shifting environments.

- *CMAML* [13] includes an update modulation phase and a prolonged adaptation phase. The former uses a function $g_\lambda : \mathbb{R} \to (0, 1)$ to modulate the learning rate of the meta-model proportionally to the loss value. The latter updates the meta-model with buffered tasks when the environment changes or a task switch is detected. Otherwise, it keeps updating w.r.t the previous model. When $\lambda = 0$, CMAML is equivalent to MAML, which never updates the meta-model. When $\lambda \to \infty$, CMAML updates the meta-model with a constant learning rate.

- *MetaBGD and MetaCOG* [8] perform CML based on MAML and Bayesian Gradient Descent (BGD), where Meta-COG introduced a per-parameter mask.

- *Fine-tuning* uses the meta-model as initialization and consistently updates this model w.r.t the tasks encountered, which can be considered a lower bound in the setting to illustrate *catastrophic forgetting*.

*Remark* F.1. Two modifications w.r.t original OSAKA benchmark: (1) We consider a task-aware setting, where the task constantly shifts at each timestep. (2) The cumulative performance in the original OSAKA is evaluated on the model before updating to current task data, which does not use the support query splits. Instead, we follow the CML setting and evaluate all the baseline algorithms on the model after updating to the current task.

**Computational Complexity** The computational complexities of all the methods are provided in Tab. 3, where the run time is measured in terms of the number of gradient computations. All the methods are assumed to use a $K$-step SGD as the inner algorithm.

- Fine-tuning uses the meta-model as initialization and consistently updates this model w.r.t the tasks encountered. MAML and ANIL do not update the meta-model during the learning phase. Therefore, these methods need to compute $T * K$ gradients for the $T$ encountered tasks.

- MetaBGD and MetaCOG perform CML based on MAML and use Bayesian Gradient Descent (BGD) for meta-model adaptation, which requires $M$ Monte Carlo samplings for the meta-gradient computation. Hence, the computation complexity is $\mathcal{O}(T * (K + M))$.

- DCML and CMAML need one meta-gradient and $K$ inner gradients computations for each task, so the complexity is $\mathcal{O}(T * (K + 1))$.

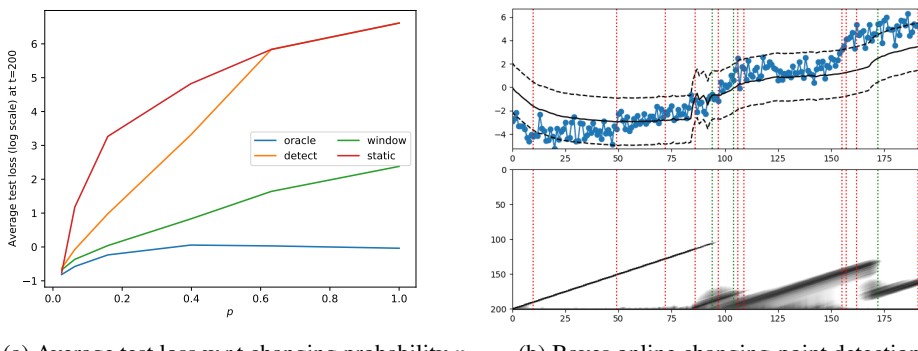

(a) Average test loss w.r.t changing probability $p$.      (b) Bayes online changing-point detection

Figure 7: Additional experiments for moving 2D Gaussian

## F.2 Additional results

### F.2.1 Moving 2D Gaussian

**Comparison of different environment shift detection methods** In Fig. 7 (a), we plot the average test loss at $t = 200$ w.r.t the environment changing probability $p$ in log scale, where we set the window size to 10 for "window". We can see that the "window" performs similarly to the "oracle" when the environment changes occur not too frequently. "Static" shows the worst result, then the "detect ." We simply (without carefully adjusting the hyper-parameters) adapted the Bayes Online Changing-point Detection (BOCD) to the algorithm output, which, in fact, highly depends on the performance of the base learner. Fig. 7 (b) illustrates the detection results of BOCD, where the green lines are the detection results and the red lines are the actual changing points. The upper figure shows one of the dimensions of the algorithm output $w_t$ with blue points. The lower figure illustrates the run-length posterior at each time step using a logarithmic color scale, where darker indicates a higher probability. We can see that with the current hyper-parameter setting, detection results are delayed compared to the actual changing points. BOCD is more likely to miss detecting some changing points rather than conducting false detection. The pseudo-code for BOCD is provided in Algo. 3.

**Not always a necessity for the exact detection of changing points** We observed a better performance of "window" than "oracle" for the OSAKA dataset in the main paper, which implies that the exact changing points detection may not be necessary.

This result is related to the algorithm, which does not necessarily hold for other methods that maintain multiple meta-models. By updating a *single meta-model online*, the context switch cost exists for reconstructing the meta-knowledge in each slot. Setting a fixed window size can ensure the meta-knowledge quality of each slot and yield good performance on average.

In addition, the phenomenon is related to the *overlap between the consecutive environment distributions* and the *environment shift probability $p$*. We empirically tested these factors, and the result is presented in Fig. 8. In Fig. 8, "oracle" is better than "window" when $p < 0.1$, and the gap becomes more evident when the distribution overlap is smaller.

*Remark* F.2. A larger overlap between distributions indicates that they are more similar, which can provide a better transfer performance (smaller test loss in Fig. 8 (a) than in (b)). In this case, the two distributions are close, so it's hard to determine precisely which distribution the encountered task belongs to. When two distributions are well-separated, it's easy to detect the shift correctly, but the transfer error is larger.

### F.2.2 OSAKA benchmark

We present in Tab. 6 additional results for the Synbols dataset with the environment changing probability $p = 0.4$ and $p = 1.0$. We can see DCML performs much better than baselines on new environments, especially the hardest one – Font.

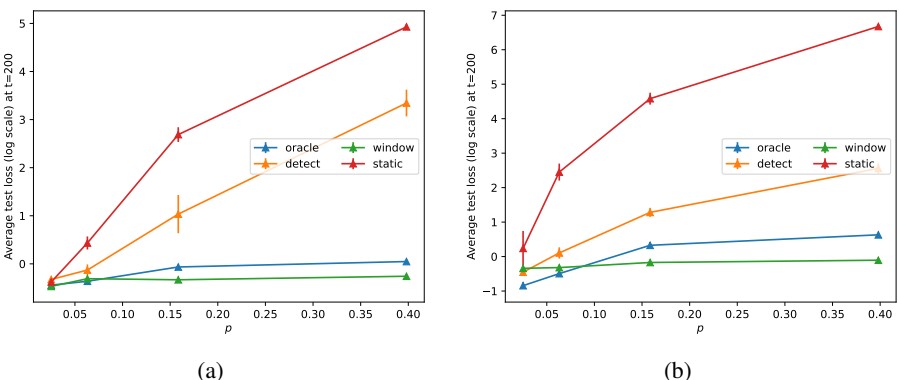

(a)                 (b)

Figure 8: Justification of "Oracle" and "Window" (window size $= 1/p$) methods on the Moving 2D Gaussian mean estimation: (a) Consecutive environment distributions with a **large overlap**, where $\tau_t = \mathcal{N}(\tilde{\phi}_t^*, I_2)$, and $\tilde{\phi}_t^* = \tilde{\phi}_{t-1}^* + (1, 1)$ with probability of $p$. (b) Consecutive environment distributions with a **small overlap**, where $\tau_t = \mathcal{N}(\tilde{\phi}_t^*, 0.1I_2)$, and $\tilde{\phi}_t^* = \tilde{\phi}_{t-1}^* + (2, 2)$ with probability of $p$.

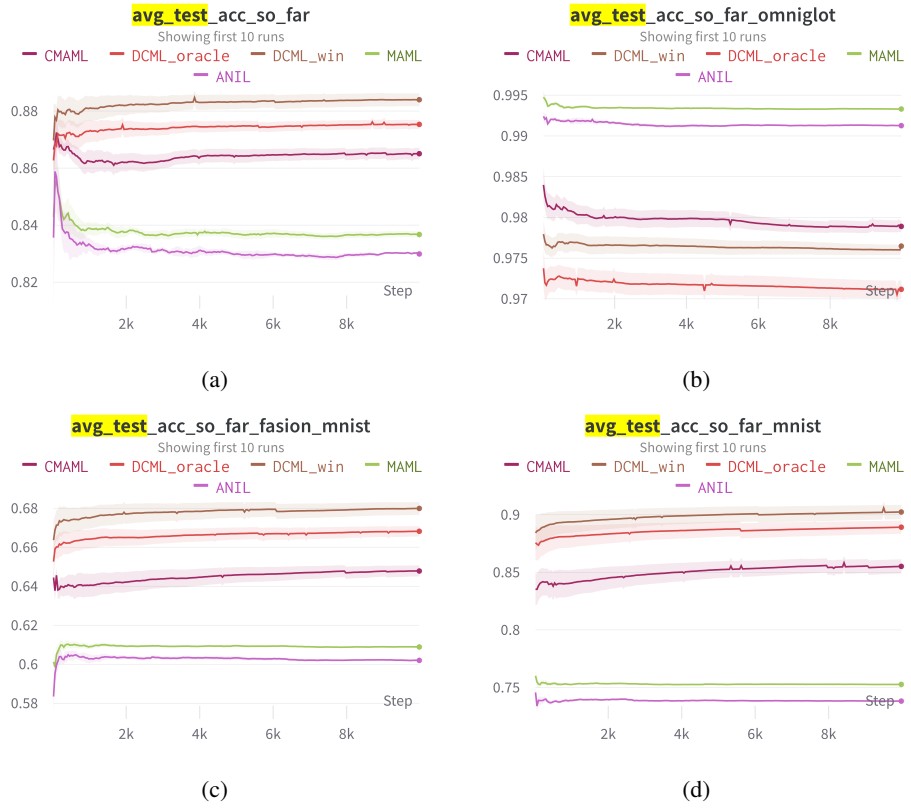

Figure 9: Average test accuracy on (a) all environments (b) Omniglot, the pre-trained environment (c) Fashion MNIST, the unseen environment (d) MNIST, the unseen environment when the environment changes with probability $p = 0.8$.

In addition, we provide the experimental records in wandb for the OMF dataset when the environment changes with probability $p = 0.8$. The average test accuracies w.r.t timesteps in the CL phase are plotted in Fig. 9.

Table 4: Average test accuracy (%) at $t = 10000$ on Synbols datasets of OSAKA benchmark

| | SYNBOLS, $p = 0.4$ | | | | SYNBOLS, $p = 1.0$ | | | |
|---|---|---|---|---|---|---|---|---|
| MODEL | ALL ENV. | ALPHA. | NEW ALPHA. | FONT | TOTAL | ALPHA. | NEW ALPHA. | FONT |
| FINE TUNING | 25.3± 0.8 | 25.4± 1.2 | 25.2± 0.5 | 25.1± 0.3 | 25.2± 0.7 | 25.4± 1.0 | 25.2± 0.5 | 25.1± 0.4 |
| METACOG [8] | 25.1± 0.9 | 25.1± 1.3 | 25.0± 0.7 | 25.0± 0.3 | 25.0± 0.8 | 25.7± 0.9 | 25.0± 0.8 | 25.0± 1.0 |
| METABGD [8] | 29.2± 7.5 | 30.9± 10.8 | 27.8± 5.0 | 27.0± 3.2 | 28.6± 6.7 | 30.2± 9.6 | 27.4± 4.6 | 26.6± 3.2 |
| ANIL [47] | 60.6± 0.3 | 77.5± 0.1 | 53.4± 0.2 | 32.6± 0.2 | 60.4± 0.1 | 77.6± 0.1 | 53.7± 0.1 | 32.8± 0.2 |
| MAML [4] | 77.0± 0.3 | **96.8± 0.1** | 72.9± 0.2 | 40.3± 0.3 | 76.8± 0.1 | **96.8± 0.2** | 73.0± 0.2 | 40.2± 0.4 |
| CMAML [13] | 64.9± 1.5 | 79.0± 3.8 | 60.2± 0.6 | 40.3± 1.4 | 63.1± 1.1 | 77.0± 3.1 | 58.6± 0.8 | 40.1± 1.6 |
| DCML(ORACLE) | 77.1± 0.6 | 95.8± 0.2 | 73.2± 0.3 | **42.3± 0.4** | 77.1± 0.1 | 95.7± 0.5 | 73.1± 0.4 | 43.4± 1.2 |
| DCML(WINDOW) | **77.2± 0.5** | 96.0± 0.3 | **73.4± 0.4** | 42.2± 0.8 | **77.2± 0.2** | 95.6± 0.6 | **73.2± 0.5** | **44.2± 1.6** |

## F.3 Experimental details

### F.3.1 Moving 2D Gaussian

**Model** As we described in the main paper, the hypothesis space of the moving 2D Gaussian is $\mathbb{R}^2$. And we adopt mean square loss $\|w - x\|^2$, which is 2-strongly convex, so it's 2-quadratic growth with $\alpha = 2$. Moreover, we limited the data to $[-10, 10] \times [-10, 10]$. Thus, for the given loss function, the Lipshitz constant is $L = 20\sqrt{2}$, and we set $L = 30$ in the code.

**Training Details** The hyper parameter settings and training details for Moving 2D Gaussian are presented in Table 5. We can see with the Algorithm 1 provided in the main paper. There is no need to set the hyper-parameters for the base learner. They are automatically adapted to the derived equations. Moreover, the initial learning rate of the meta learner is controlled by D_hat_guess. Then, it's also scheduled automatically with time and environment changing points.

Table 5: Moving 2D Gaussian experimental details

| Hyper-parameters | Fig. 3(b) | Fig. 7(a) |
|---|---|---|
| task horizon $T$ | | 200 |
| loss | | mean square loss |
| Lipschitz constant $L$ | | 30 |
| $\alpha$ | | 2 |
| hopping lr $\rho$ | | 0.8 |
| D_hat_guess | | 5 |
| sample numbers $m$ | 100 | 100 |
| inner epoch $K$ | 1 | 2 |
| changing prob $p$ | 0.05 | [0.025, 0.063, 0.158, 0.398, 0.631, 1] |

**Computing Resource** All experiments for Moving 2D Gaussian were tested on a Mac with an Intel Core i5 CPU and 8G memory.

### F.3.2 OSAKA benchmark

**Model for OMF dataset** We used a CNN network architecture as the basic model. The four modules are identical: A $3 \times 3$ 2d convolution layer with 64 filters, stride 1, and padding 1. The following are a batch normalization layer, a Relu layer, and a max-pooling layer of stride 2. The image data passing the aforementioned modules form a $64 \times 1 \times 1$ feature map, which was further taken into a fully connected layer. Then the fully connected layer outputs the logits for a 10-class classification. At last, the cross-entropy loss is calculated with the logits and the corresponding labels.

**Model for Synbols dataset** We used a CNN network architecture as the basic model. The four modules are identical: A $3 \times 3$ 2d convolution layer with 64 filters, stride 1, and padding 1. The following are a batch normalization layer, a Relu layer, and a max-pooling layer of stride 2. The image data passing the aforementioned modules form a $64 \times 2 \times 2$ feature map, which was further flattened and taken into a fully connected layer. Then, the fully connected layer outputs the logits for a 4-class classification. At last, the cross-entropy loss is calculated with the logits and the corresponding labels.

**Training Details** The hyper-parameter settings and training details for OSAKA data sets are outlined in Table 6.

**Computing Resource** The experiments for OSAKA were run on a server node with 6 CPUs and 1 GPU of 32GB memory.

Table 6: OSAKA experimental details

| Hyper-parameters | OMF | Synbols |
|---|---|---|
| task horizon $T$ | 10000 | |
| loss | cross entropy | |
| $\alpha$ | 4 | |
| D_hat_guess | 100 | |
| image size | 28*28 | 32*32 |
| image channel | 1 | 3 |
| n_shots | 5 | 5 |
| n_shots_test | 15 | 15 |
| n_ways | 10 | 4 |
| prob_statio | 0.0 | 0.0 |
| sample number $m =$ n_shots * n_ways | 50 | 20 |
| inner epoch $K$ | 8 | 16 |
| prob_env_switch $p$ | [0.2, 0.4, 0.6, 0.8, 1.0] | [0.2, 0.4, 1.0] |
| Lipschitz constant $L$ | 200 | 100 |
| hopping lr $\rho$ | 1.0 | 0.2 |
| eta_0 | 6.0 | 20 |
| epsilon_0_rate | 1.0 | 4.0 |

**Hyper-parameter Search**    Following OSAKA, the hyper-parameters were tuned by random search, and the same number of trials were allocated for each algorithm. For each trial, we sampled hyper-parameter combinations uniformly from the search space presented in Tab. 7 for baselines and in Tab. 8 for DCML.

According to the experimental setting of OSAKA, we do not need a validation dataset for searching the hyper-parameters. New CML episodes (task sequences) generated with different random seeds can be considered held-out data. The results reported in the paper are tested on 10 new CML episodes using the learned hyper-parameters.

Table 7: Hyper-parameter search space for baselines

| Hyper-parameters | Search space |
|---|---|
| meta-learning rate $\eta$ | 0.0001, 0.0005, 0.001, 0.005, 0.01 |
| base learning rate | 0.0005, 0.001, 0.005, 0.01, 0.05, 0.1, 0.5 |
| inner steps | 1, 2, 4, 8, 16 |
| first-order | True, False |
| modulation $\lambda$ | 0.25, 0.5, 0.75, 1.0, 1.25, 1.5, 2.0, 2.5, 3.0 |
| parameter variance $\sigma$ | 0.0001, 0.01, 0.1 |
| step-size for parameter mean $\beta$ | 0.5, 1.0, 10 |
| MC samples | 5 |
| threshold for task switch $\gamma$ | -2 |

### F.4    Bayes Online Changing-point Detection

We modify the implementation of BOCD in [52] for our experiments. The detailed method is presented in [42]. Given a sequence of tasks, we use the base leaner outputs $w_{1:t}$ as observations of BOCD to be divided into non-overlapping product partitions. Each partition can be considered as sampled from the same static environment slot. BOCD model the detection of changing point as estimating the posterior distribution over the current run-length $r_t \in [0, 1, ..., t]$, *i.e.*, $P(r_t|w_{1:t}) = \frac{P(r_t, w_{1:t})}{P(w_{1:t})}$. Thus, $r_t = 0$ means to start a new partition.

We treat the hypothesis as factorized Gaussian and apply BOCD on each dimension of the algorithm output $w_t$. For Moving 2D Gaussian, we detect the whole $w_t$, since it's just 2D. Here, we provide the pseudo-code in Algorithm 3 with Gaussian Prior for each dimension applied in our experiments.

Table 8: Hyper-parameter search space for DCML

| Hyper-parameters | Search space |
|---|---|
| quadratic growth parameter $\alpha$ | 0.05, 0.5, 1, 2, 4, 6, 8, 10 |
| Lipschitz constant $L$ | 50, 100, 150, 200, 250, 300, 350, 400 |
| number of inner steps $K$ | 1, 2, 4, 8, 16 |
| slot diameter guess $\hat{D}$ | 50, 100, 200, 400, 600 |
| hopping learning rate $\rho$ | 0.2, 0.4, 0.6, 0.8, 1.0 |
| initialization $\beta_1$ | 2, 4, 6, 8, 10, 12, 14, 16, 18, 20 |
| initialization of $\epsilon_0$ | 1, 2, 4, 6, 8, 10 |

---

**Algorithm 3** Pseudo code for detecting environment change

---

**Require**: $P(r_0 = 0) = 1$, prior mean $\mu_1^0 = 0$, prior precision $\delta_1^0 = 1/0.1$, changing probability $p$, data precision $\delta_w = 1$

**for** $t \leftarrow 1$ *to* $T$ **do**

    observe new algorithm output $w_t$

    evaluate the predictive probability $\pi_t^r = P(w_t | \mu_t^r, \delta_t^r)$

    calculate the growth probability $P(r_t = r_{t-1} + 1, w_{1:t}) = P(r_{t-1}, w_{1:t-1})\pi_t^r(1 - p)$

    calculate the changepoint probability $P(r_t = 0, w_{1:t}) = \sum_{r_{t-1}} P(r_{t-1}, w_{1:t-1})\pi_t^r p$

    calculate evidence $P(w_{1:t}) = \sum_{r_t} P(r_t, w_{1:t})$

    update run length distribution $P(r_t | w_{1:t}) = P(r_t, w_{1:t})/P(w_{1:t})$

    update statistics:

    $\delta_{t+1}^r = \delta_t^r, \delta_{t+1}^{r+1} = \delta_t^r + \delta_w$

    $\mu_{t+1}^r = \mu_t^r, \mu_{t+1}^{r+1} = 1/\delta_{t+1}^{r+1} * (\delta_t^r \mu_t^r + w_t \delta_w)$

    **if** $\arg\max_k P(r_t = k | w_{1:t}) < \arg\max_k P(r_{t-1} = k | w_{1:t-1})$ **then**

        changing point detected

    **end**

**end**

---

# G  Additional Discussion

## G.1  Learning-forgetting w.r.t meta-updates

In this section, we offer some intuitive interpretations for the adaptive updates of the meta-parameter of the base learner.

The updates of the meta-parameter ($\phi$ and $\beta$) in Algo. 1 and 2 can be separately expressed as:

$$\phi_{t+1} = (1 - \frac{2b\kappa_t\gamma_t}{\beta_t})\phi_t + \frac{2b\kappa_t\gamma_t}{\beta_t}w_t \text{ and } \beta_{t+1} = \beta_t - \gamma_t(a\kappa_t - \frac{\kappa_t(b\|\phi_t - w_t\|^2 + \epsilon_t + \epsilon_0)}{\beta_t^2}).$$

Considering the case that all the tasks have the same sample number. $\beta_t$ reflects the confidence of $\mathcal{A}$ on the given initialization for learning the task $t$. For example, $\beta_t$ is the learning rate (step size) for SGD. A large learning rate means that the learner relies more on the data, and a small one means believing the prior. From the update of $\beta$, we can see a large $\beta_{t+1}$ implies that the current task output is far from its initialization ($\|\phi_t - w_t\|$ is large), which indicates the current task $t$ may be an outlier, exceeding the range of meta-model for recovering its performance. Hence, the environment may have changed. So in the next task, the base learner learns more from the task data instead of keeping close to the initialization ($\beta_{t+1}$ is large).

The learning rate of the initialization (or bias) is determined by $\frac{2b\kappa_t\gamma_t}{\beta_t}$. As mentioned above, a large distance from the initialization indicates that the current task $t$ may be an outlier. So from $t+1$ step, the tasks are treated with low confidence to update the common initialization $\phi_{t+1}$ with $\frac{2b\kappa\gamma_t}{\beta_{t+1}}$ being small ($\beta_{t+1}$ is large). Then until the distance w.r.t the common initialization holds a small value for several rounds, $\beta$ also decays to a smaller value. The tasks start to contribute more to updating the initialization.

The above mechanism works well for static environments and can address forgetting in shifting environments. But it will lose tracking ability in the shifting environment, so we need slot-wise adjusting for $\gamma_t$ to obtain a better trade-off.

## G.2  Real-world examples for practicality and necessity of CML

A typical real-world application of CML is the online recommendation system, where we aim to predict the user's preference for various products. Different users' preferences are considered as different task distributions. The distribution of users can be regarded as a task environment. The recommendation system maintains a meta-model that predicts the recent preferences shared across users and adapts task-specific models that predict the user-specific preferences.

The user queries for displaying products arrive sequentially, and the system randomly distributes several (e.g., $m_t = 200$) products in response to the query. Denote the products and the corresponding preferences for $t$-th user as $(X_i, Y_i) \sim \mu_t, \forall i \in [m_t]$, which are i.i.d.. The labels are initially unknown for this setting, but a few labels can be acquired through the following interaction:

- The meta-model predicts the preferences over products and displays them according to the predicted preference order.
- Normally, the user will only view a small part ($m'_t = 50$) of the distributed products, and the preference can be determined with the clicks or view time over these products, which gives the few-shot labeled samples.
- Then, the task-specific model is adapted from the meta-model with these examples. We hope the model can generalize on the rest of the unseen examples with correct preference prediction, which can provide new recommendations.
- Finally, the task-specific model of each user is used for updating the meta-model. Since the shared preferences change with the season, fashion trends, and accidental events, the task environment is shifting.

## G.3  More details on DCML

**Relate parameters in eq. 1 to specific base learner**   Since the proposed framework is valid for different base learners, the parameters like $a$, $b$, and $\kappa$ in the unified form of excess risk upper bounds

(cost function) are related to the *specific algorithm*. Since all of the experiments adopt SGD, we give here a detailed description of it.

According to Proposition D.4, the $K$-step SGD with learning rate $\eta_t$ and initialization $\phi_t$ for the $t$-th task has the following cost function:

$$f(\beta_t, \phi_t) = \kappa_t \left( \frac{1}{2}\beta_t + \frac{2\alpha\|\phi_t - w_t\|^2 + \epsilon_0 + 2\kappa_t^2}{\beta_t} + 2\kappa_t \right),$$

where $a = \frac{1}{2}, b = 2\alpha, \kappa_t = L\sqrt{\frac{1}{K\alpha} + \frac{2}{m_t\alpha}}, \epsilon_t = 2\kappa_t^2, \Delta_t = 2\kappa_t$. The learning rate of the base learner $\eta_t = (\frac{\beta_t}{\kappa_t} + 2)/(K\alpha)$ is related to the sample number $m_t$ of each task via $\kappa_t$, which can reflect the generalization.

**Impact of the initial meta-parameter on the model's behavior**  The initial meta-parameter substantially affects the model's behavior. For instance, the aforementioned recommendation system suffers from a well-known *cold-start* problem. A well-pertained model initialization $\phi_1$ can help address the problem.

Given a good initialization $\phi_1$, $\beta_1$ also affects the performance substantially. According to the meta-parameter adaptation, if $\beta_1$ is too large, the algorithm tends to keep the prior knowledge and will not learn from the data. If $\beta_1$ is too small, the algorithm will ignore the initialization and learn from scratch.

**How does DCML adjust the meta-parameter**  The updates in Algo. 1 for the meta-parameter $u = (\beta_t, \phi_t)$ are:

$$\phi_{t+1} = (1 - \frac{2b\kappa_t\gamma_t}{\beta_t})\phi_t + \frac{2b\kappa_t\gamma_t}{\beta_t}w_t, \beta_{t+1} = \beta_t - \gamma_t(a\kappa_t - \frac{\kappa_t(b\|\phi_t - w_t\|^2 + \epsilon_t + \epsilon_0)}{\beta_t^2}).$$

We adjust the learning rate of the meta-parameter ($\gamma_t$) with the following strategy:

- For $k$-th task inside the $n$-th environment (slot), $\gamma_t = \gamma_0/\sqrt{k}$, where the initialization is determined by the theoretical optimal value $\gamma_0 = \frac{\epsilon_0}{\kappa_t}\sqrt{\frac{(1+b/a)\hat{D}^2 + \epsilon_t/\alpha}{4ab^2\hat{D}^2\epsilon_0 + a^2(b*\hat{D}^2 + \epsilon_t^2)^2}}$.

- When an environment change is detected, $\gamma_t$ is set to a large hopping rate $\gamma_t = \rho$, which is related to the path length.

**How to set hyper-parameters**  In the experiments, we tune the following hyper-parameters $\alpha, K, L, \hat{D}, \epsilon_0, \rho, \beta_1$ by random search on the defined space in Tab. 8. Specifically for the simple moving 2D Gaussian: since the loss function is 2-strongly convex, we have $\alpha = 2$.

## G.4  Limitations

The proposed theory works for non-convex loss when using the Gibbs algorithm. However, We do not offer analysis for SGD with non-convex loss, where it's hard to find an easily optimizable form for the excess risk upper bound. Excess risk analysis for gradient-based learning algorithms with non-convex loss is a promising but challenging direction for the deep learning community. We hope to work on this topic in the future.

In this paper, we do not provide theoretical guarantees for memory-based approaches, which can better address meta-level forgetting with additional memory cost. Future theoretical studies can be conducted to complete the framework.

## G.5  Broader Impacts

In general, this work is fundamental. It aims to understand the bi-level learning-forgetting trade-off in CML and improve the model performance in real-time intelligent systems with a better trade-off balance. The potential negative impacts depend on the specific application.

**Potential negative impacts**   Since the main objective is to obtain the optimal average excess risk over rounds, when each task relates to a person in some applications like personalized recommendation systems or medical diagnosis systems, it may cause some fairness issues for individuals, especially those who appear around the changing points.

**How to address?**   The aforementioned potential unfairness can be mitigated by adding calibration modules as in [53–55] or providing more training data for tasks around changing points.

# H   Additional Related Works

## H.1   Summary and comparison of related settings

Table 9: Comparison of different settings adapted from [13], where $S$ and $Q$ represent the train and test data from the same distribution, respectively. Train data $S = \bar{S} \cup \tilde{S}$ is further divided into the support and query sets. $\mu_t$ is a sub-task in CL with i.i.d. data, $CLP_{1:M}$ represent whole CL problems of infinite non-i.i.d. data stream and $p(\mathcal{T})$ is the distribution over them.

| Settings | Task Datasets | Tasks & Environments | Task Parameter (base) | Meta Updates | Evaluation |
|---|---|---|---|---|---|
| SL | $S, Q \sim \mu$ | — | $w = \mathcal{A}(S)$ | — | $\mathcal{L}(w, Q)$ |
| CL | $S_{1:T}, Q_{1:T} \sim \mu_{1:T}$ | $\mu_{1:T} \sim \tau$ (static ) $\mu_{1:T} \sim \tau_{1:T}$ (shifting) | $w = \mathcal{A}_{CL}(S_{1:T})$ | — | $\sum_{t=1}^{T} \mathcal{L}(w, Q_t)$ |
| ML | $S_{1:M} \sim \mu_{1:M}$ (train tasks), $S_{M:N} \sim \mu_{M:N}$ (test tasks) | $\mu_{1:N} \sim \tau$ | $w_i = \mathcal{A}(u, S_i),$ $\forall i \in [M], M < N$ | $\nabla_u \sum_{i=1}^{M} \mathcal{L}(\mathcal{A}(u, \bar{S}_i), \tilde{S}_i)$ | $\sum_{i=M}^{N} \mathcal{L}(\mathcal{A}(u, \bar{S}_i), \tilde{S}_i)$ |
| CML | $S_{1:T}, Q_{1:T} \sim \mu_{1:T}$ | $\mu_{1:T} \sim \tau$ (static) $\mu_{1:T} \sim \tau_{1:T}$ (shifting) | $w_t = \mathcal{A}(u, \bar{S}_t)$ | $\nabla f_t(u_t)$ or $\nabla_{u_t} \mathcal{L}(\mathcal{A}(u_t, \bar{S}_t), \tilde{S}_t)$ | $\sum_t \mathcal{L}(\mathcal{A}(u_t, \bar{S}_t), Q_t)$ |
| MCL | $\bar{S}_{i,1:T}, Q_{i,1:T} \sim CLP_i$ $\forall i \in [M]$ | $CLP_{1:M} \sim p(\mathcal{T})$ | $w_i = \mathcal{A}_{CL}(u, \bar{S}_{i,1:T})$ | $\nabla_u \sum_t \mathcal{L}(\mathcal{A}_{CL}(u, \bar{S}_{i,1:T}), \tilde{S}_{i,t})$ | $\sum_{i=1}^{M} \sum_t \mathcal{L}(\mathcal{A}_{CL}(u, \bar{S}_{i,1:T}), Q_{i,t})$ |

We present different settings related to the Continual Meta-Learning (CML) problem studied in this paper in Tab. 9. The detailed discussions in each setting can be found in the following sections.

Supervised Learning (SL) methods learn as a single task from the train data and evaluate on independent test data. The data points are assumed, i.i.d. sampled from the same distribution. Continual Learning (CL) aims to learn from a sequence of non-i.i.d. tasks, where the data points inside each task are i.i.d. Meta-Learning (ML) has many settings. Here, we present the statistical version. All the tasks are assumed to be i.i.d. sampled from the same task environment. The evaluation is conducted on test tasks to measure the generalization performance of the algorithm on new tasks.

In Meta-Continual Learning (MCL), a continual learning prediction problem (CLP) is defined as a non-i.i.d. data stream, which is a task sampled from the stationary distribution $p(\mathcal{T})$. Random subsequences of dependent data points of length $k$ are sampled from each CLP task for training. The major difference between CML and MCL settings is that the non-stationarity of CML comes from the sequentially encountered non-i.i.d. task distributions. In contrast, the non-stationarity of MCL comes from the non-i.i.d. data points within a task.

## H.2   Continual Learning

Except for the taxonomy w.r.t different approaches mentioned in the main paper, CL can also be divided into the following three scenarios w.r.t the different properties of the task distributions [56, 57]. *Class Incremental Learning (CIL)* [58] involves learning new classes over time without having access to all classes at once during training. The label space is growing over time, where $\mathcal{Y}_t \subset \mathcal{Y}_{t+1}, \forall t$. In the final step, the predictor is aimed to be capable of classifying all the seen classes. *Domain Incremental Learning (DIL)* [59] defines on the same label space $\mathcal{Y}$ that has a fixed associated semantic meaning, and the marginal distributions on input space $\mu_t(\mathcal{X})$ varies over time. *Task Incremental Learning (TIL)* [5] can define on different label spaces $\mathcal{Y}_t \neq \mathcal{Y}_{t+1}, \forall t$ or the same label space $\mathcal{Y}$ associated to different semantic meanings over tasks. In conventional definitions, CIL and DIL are under a weak task-agnostic setting (the task identities or task boundaries are unknown during testing but known during training), while TIL is under the task-aware setting with known task identities. However, with increasing research diversities, TIL has also been studied in task-agnostic settings recently. In this paper, we study CML within a similar TIL scenario of CL.

**Key Differences between CL and CML**   The two settings share the common objective of addressing the stability-plasticity dilemma, which is the typical objective for learning from non-stationary data. Their key differences are:

Continual Learning:

- Standard CL methods sequentially adapt a task model and aim for a final model that performs well on all the tasks encountered. (Some memory-based approaches like dynamic architecture grow a subnet for each task.)
- The current task model is adapted from the previous task model.
- Standard CL reports the final performance of the agent on all the tasks at the end of its lifetime.
- Standard CL approaches mainly focus on minimizing catastrophic forgetting without considering quick generalization for tasks with few-shot data, e.g., the replay-based and regularization-based methods.

Continual Meta-Learning:

- CML methods sequentially adapt a meta-model and aim to recover the performance on previous tasks by adapting the meta-model with few additional samples. Therefore, it's more suitable for learning few-shot tasks. (Some memory-based approaches grow a subnet for each environment.)
- The current task model is adapted from the meta-model.
- CML reports the cumulative performance of the agent throughout its lifetime.
- CML focuses on quick generalization for new tasks and fast recovering performance of previous tasks with few-shot data.

### H.3   Meta-Continual Learning

Javed and White [60] and Gupta et al. [61] study a different Meta-Continual Learning (MCL) setting. A continual learning prediction problem (CLP) is defined as a non-i.i.d. data stream, which can be considered a CLP task $\mathcal{T}$. The CLP tasks are sampled from a stationary distribution $p(\mathcal{T})$, where for each CLP task, random subsequences of dependent data points are sampled from the task for training. [60] conducts offline meta-representation learning w.r.t two objectives MAML-REP and OML. MAML-REP uses batch data for inner adaptation. OML is calculated with a single data point for each inner adaptation, which can reflect the degree of forgetting in CLP tasks. Gupta et al. [61] proposes La-MAML, which adopts a replay buffer and conducts meta-initialization with optimization on the OML objective in an online manner. In addition, La-MAML adopts a modulation of per-parameter learning rates in the meta-learning updates. The optimization of learning rates in La-MAML is w.r.t empirical loss, while our learning rates optimization is based on the excess risk upper bound. So, our approach suffers less from over-fitting.

### H.4   Continual Meta-Learning

**Static environments**   Harrison et al. [32] studies continual meta-learning (CML) under task agnostic setting, where the task boundaries are unknown. It proposed an algorithmic framework MOCA that incorporates different meta-learning methods with Bayesian Online Changing-point Detection (BOCD) to identify unknown task boundaries during the meta-learning process. The task environment is still assumed static. On the contrary, we focus on the bi-level trade-off in shifting task environments with known task boundaries, where we detect the environmental distribution shift, not the task distribution shift.

**Shifting environments**   To address the environment shift, Jerfel et al. [14] proposes a non-parametric Dirichlet process mixture of hierarchical Bayesian models that allows adaptively adding new task clusters and selecting over a set of learned meta-initialization parameters for the newly encountered task. A similar approach is also used to detect the task shift in a task-agnostic setting. For both task-agnostic and task-aware settings, [14] has tested the proposed algorithm on datasets with two times environment shifts. The proposed method requires additional $\mathcal{O}(K)$ memory for

storing the initialization parameters for the detected $K$ clusters. For each task, it needs to update the task-specific model and the meta-model for all the $K$ initializations. $K$ can grow with the time.

Zhang et al. [15] use a Dynamic Gaussian Mixture Model (DGMM) to model the distribution of the meta-parameters. Different from [14], which applies point estimation during inference, [15] derived a structured variational inference method that can reduce overfitting. The experiment is conducted with three times environment shifts of four datasets. [15] needs $\mathcal{O}(K)$ extra memory for storing the meta-parameters.

Wang et al. [16] studies a slightly different setting where each environment has a super long task sequence. In this case, the cartographic forgetting for meta-knowledge is more obvious. To address this, they use a memory buffer to store a small number of training tasks from previous environments and a shared representation for different task environments and grow the subnets when a new environment is detected on latent space using BOCD. For deciding which cluster the task belongs to, they store the average embedding for each environment for calculating the distances. Hence, its memory complexity is $\mathcal{O}(K + M)$.

### H.5   Meta-Learning

**Statistical Learning to Learn (LTL)**   LTL [62] has been intensively studied with various theoretic tools. The early theoretical framework was proposed by Baxter [24], where they first defined the notion of the task environment and derived the uniform convergence bounds. And the tasks are i.i.d sampled from the environment. Denevi et al. [63, 28] provided excess risk for ridge regression and discussed the train-validation split impact. Amit and Meir [25] applied PAC Bayes theory and provided generalization bounds for stochastic neural networks. They also derived a joint training algorithm that simultaneously updates the meta-parameter and task parameters, which cannot be extended to the sequential learning scenario. Recently, Chen et al. [26] derived an information-theoretic generalization bound for the MAML-like algorithm, which provides non-vacuous bounds for deep few-shot learning and can be applied to sequential task learning.

**Statistical Lifelong Learning**   The difference between Lifelong learning [64] and LTL is that the tasks are observed sequentially, while LTL has all the tasks in hand. Pentina and Lampert [65] first applied PAC Bayes theory to Lifelong Learning, where they also assumed that all the tasks are, i.i.d. sampled from the task environment. In a subsequent work [66], they relaxed the i.i.d. assumption with two scenarios. The first is that the tasks are dependent, but a dependency graph is known so that they can prove a statistic bound. In the second, they also consider a shifting environment change. However, they assume that the base learner output of the current task only depends on the current and the previous task data, and the expected performance of the base learner does not change with time. Under these assumptions, they obtain statistical guarantees for the gradually changing environment.

**Online LTL**   The online LTL methods have all the tasks in hand, but at each round $t$, they sample a task from the $N$ tasks in hand. Cavallanti et al. [67] works on this setting for online multi-task learning. Multi-task learning is often treated as a simplified version of LTL in previous works [68].

**Online Meta-Learning**   Two main settings to meta-learn sequential tasks in an online manner are referred to as *online-within-online* (OWO) [28–31] and *batch-within-online* (BWO) [9, 27]. The former applies online algorithms for both base and meta learners. The latter treats task-level learning differently as a statistical batch setting. The BWO setting is close to the CML studied in this paper, while the theoretical work is rare, and none of the previous BWO methods considers shifting task environments. Khodak et al. [31] first considered shifting environments in the OWO setting, where the base learner is the gradient-based online learner. In contrast, we consider more choices of batch base learners and a fine-grained algorithm w.r.t environment change. *Even though their bounds are not w.r.t AER and the results are not directly comparable*, we make an intuitive comparison, which suggests the proposed algorithm in this paper has an improved rate for gradient-based base learners in slot-wise stationary environments over their related results. Moreover, we conducted a rigorous analysis of the bi-level trade-off, which is missing in the related works mentioned above. Finally, although not clearly defined in previous meta-learning literature, we note Meta Continual Learning (MCL) [60, 61] can be named in a similar way as *Online-Within-Batch* (OWB), where the tasks are processed in one batch, but the data within each task are processed online.

**Other related works**  Model-Agnostic Meta-Learning (MAML) [4] that uses the higher-order gradients for meta-updates has gained tremendous success in practice. Therefore, a lot of work emerges to improve MAML. Finn et al. [69], Grant et al. [70], Yoon et al. [71] combine MAML with Bayesian methods. Rajeswaran et al. [72] improves MAML with implicit gradient calculation for the meta-updates. Since meta-learning considers learning and generalization for both seen and unseen tasks, it has not only been applied to CL for addressing catastrophic forgetting but has also been applied to Test-Time Domain Adaptation (DA) for quick generalization. For more related works in Test-Time DA or DA, please refer to [73–75].

