# OpenReview forum: "On the Stability-Plasticity Dilemma in Continual Meta-Learning: Theory and Algorithm"
_NeurIPS.cc/2023/Conference — NeurIPS 2023 poster_

### Official Review · Reviewer_HjRj · 2023-06-27

**Soundness:** 4 excellent
**Presentation:** 4 excellent
**Contribution:** 3 good
**Rating:** 7
**Confidence:** 2

**Summary:**

This paper aims to address the balance between stability (preserving past knowledge) and plasticity (rapid learning from new experiences) in Continual Meta-Learning (CML). The authors propose a novel theoretical framework for CML in both static and shifting task environments. They discuss the bi-level (task- and meta-) trade-off in shifting environments and theoretically identify factors that affect stability and plasticity. They then propose a novel algorithm, Dynamic Continual Meta-Learning, that addresses the bi-level trade-off in shifting environments. DCML minimizes the expected regret by dynamically adjusting the meta-parameter and its learning rate when an environment change is detected.

**Strengths:**

1. The paper is theoretically robust, proposing a unified framework for understanding CML in both static and shifting task environments.
2. The paper is the first theoretical work studying the stability-plasticity dilemma in CML with bi-level shifts.
3. Based on the theoretical findings, the authors introduce a novel and practical algorithm which show improved estimated bounds and superior performance compared to baselines.

**Weaknesses:**

1. From a non-expert perspective, the practicality and necessity of the problem the paper addresses are unclear to me. While meta continual learning is understandable (non-stationarity comes from the non-i.i.d data points within a task), it's not clear why continual meta-learning (non-stationarity comes from the task changes for each time step) is a practical scenario. More real-world examples or explanations of why continual meta-learning is required would be beneficial.
2. The paper lacks a clear, detailed explanation of the robustness of the proposed algorithm. For instance, it's unclear how to set parameters such as a, b, and kappa for DCML. Additionally, the impact of the initial meta parameters (beta) on the model's behavior is not well-explained.

**Questions:**

1. Could the authors provide more details on how the algorithm dynamically adjusts the meta-parameter and its learning rate when an environment change is detected? Also, it would be helpful to see how beta actually moves during training.

---

> ### Author Rebuttal · Authors · 2023-08-08
>
>
> We sincerely thank the reviewer for appreciating the strength of our work and the constructive comments we can incorporate into the revision. Below are our responses to your comments.
>
> > Weakness 1. “Real-world examples for practicality and necessity of CML.”
>
> * A representative real-world application of CML is the online recommendation system, where we aim to predict each user's preference for various products.
>   - Different users' preferences are considered as different task distributions, and the distribution over users can be regarded as a task environment.
>   - The recommendation system maintains a meta-model that predicts the recent preferences shared across users and adapts task-specific models that predict the user-specific preferences.
>
> * The queries for displaying products of each user arrive sequentially, and the system randomly distributes several (e.g., $m_t = 200$) products in response to the query. Denote the products and the corresponding preferences of $t$-th user as $(X_i, Y_i) \sim \mu_t, \forall i \in [m_t]$, which are i.i.d.. The labels are initially unknown for this setting, but few labels can be acquired through the following interaction:
>   - The meta-model predicts the preferences over products and displays them according to the predicted preference order.
>   - Normally, the user will only views a small part ($m_t^\prime=50$) of the distributed products, and the preferences can be determined with the clicks or view time over these products, which yields the few-shot labeled samples.
>
> * Then, the task-specific model is adapted from the meta-model with these examples. And we hope that the model can ***generalize on the rest of unseen examples*** with correct preference prediction, which can provide new recommendations.
>
> * Finally, the task-specific model of each user is used for updating the meta-model. The task environment is shifting since the shared preferences change with the season, fashion trends, and accidental events.
>
> > Weakness 2. "detailed explanation of the robustness of the proposed algorithm..."
>
> * How to set parameters like $a,b,\kappa$
>   - Since the proposed framework is valid for different base learners, the parameters like $a$, $b$, and $\kappa$ in the unified form of excess risk upper bounds (cost function) are related to the *specific algorithm*.
>   - We have related these parameters for the Gibbs algorithm and SGD in Proposition D.1 and Proposition D.4, respectively. A more general derivation for all the algorithms is provided in Theorem 3.1.
>   - Since all of the experiments adopt SGD, we give here a detailed description for it.
>     - According to Proposition D.4, the $K$-step SGD with learning rate $\eta_t$ and initialization $\phi_t$ for the $t$-th task has the following cost function:
>         $f(\beta_t, \phi_t)=\kappa_t\left(\frac{1}{2}\beta_t + \frac{2\alpha\|\|\phi_t-w_t\|\|^2 + \epsilon_0 + 2\kappa_t^2}{\beta_t} + 2\kappa_t\right),$
>     where $a=\frac{1}{2}, b=2\alpha, \kappa_t=L\sqrt{\frac{1}{K\alpha}+\frac{2}{m_t\alpha}},\epsilon_t=2\kappa_t^2, \Delta_t=2\kappa_t$.
>     - The learning rate of the base learner $\eta_t=(\frac{\beta_t}{\kappa_t} + 2)/(K\alpha)$ is related to the sample number $m_t$ of each task via $\kappa_t$, which can reflect the generalization.
>
> * Impact of the initial meta-parameter on the model's behaviour.
>   - The initial meta-parameter affects substantially the model's behaviour. For instance, the aforementioned recommendation system suffers from a well-known **cold-start** problem. A well pertained model initialization $\phi_1$ can help addressing the problem.
>   - Given a good initialization $\phi_1$, $\beta_1$ also affects the performance substantially. According to the meta-parameter adaptation, if $\beta_1$ is *too large*, the algorithm tends to keep the prior knowledge and *will not learn* from the data. If $\beta_1$ is *too small*, the algorithm will ignore the initialization and *learn from scratch*.
>
> > Q1. "more details on how the algorithm dynamically adjusts the meta-parameter and its learning rate"
>
> * Adjust the meta-parameter:
>
>   - The meta-parameter adaptation $u_{t+1} = \Pi_{\mathcal{U}}(u_{t} - \gamma_t \nabla f_t(u_t))$ corresponds to:
>
>     - $\phi_{t+1} = (1 - \frac{2b\kappa_t\gamma_t}{\beta_t})\phi_t + \frac{2b\kappa_t\gamma_t}{\beta_t} w_t$  and $\beta_{t+1} = \beta_t- \gamma_t (a\kappa_t - \frac{\kappa_t(b\|\|\phi_t-w_t\|\|^2 + \epsilon_t + \epsilon_0)}{{\beta_t}^{2}})$.
> * Adjust the learning rate of the meta-parameter ($\gamma_t$):
>   - For $k$-th task inside the $n$-th environment (slot), $\gamma_t = \gamma_0/\sqrt{k}$, where the initialization is determined by the theoretical optimal value $\gamma_0=\frac{\epsilon_0}{\kappa_t}\sqrt{\frac{(1+b/a)\hat{D}^2 + \epsilon_t/\alpha}{4ab^2\hat{D}^2\epsilon_0+a^2(b*\hat{D}^2+\epsilon_t^2)^2}}$.
>   - When an environment change is detected, $\gamma_t$ is set to a large hopping rate $\gamma_t=\rho$, which is related to the path length.
>
>
> * How to set hyper-parameters:
>   - In the experiments, we tune the following hyper-parameters $\alpha, K, L, \hat{D}, \epsilon_0, \rho, \beta_1$ by random search on the defined space in Table 2 (***rebuttal file***). Specifically for the simple moving 2D Gaussian: since the loss function is 2-strongly convex, we have $\alpha=2$.
>
> * How $\beta$ actually moves?
>   - In Figure A.3 (***rebuttal file***), we plot the adaptation process to show how $\beta_t$ and $\gamma_t$ move during the continual meta-learning process.
>   - In the figure, meta_lr is the learning rate of the meta-initialization $\phi_t$, which equals $\frac{2b\kappa_t\gamma_t}{\beta_t}$. The base learner learning rate (base_lr) is $\eta_t=(\frac{\beta_t}{\kappa_t} + 2)/(K\alpha)$.
>   - We can observe from Figure A.3 that meta_lr $\propto\gamma_t , \frac{1}{\beta_t}$ and base_lr $\propto \beta_t$, which is consistent with their mathematical definitions.

---

> > ### Comment · Reviewer_HjRj · 2023-08-21
> >
> > Thank you for the detailed and comprehensive response to my initial review. Your clarifications, especially the real-world applicability using the online recommendation system, significantly enhanced my understanding of the work. While I am genuinely appreciative of the added clarity and thoroughness of your response, I decided to maintain my score where my hesitations stem from the following:
> >
> > 1. Applicability: Despite the elucidation of the usefulness of CML, the applicability still appears somewhat narrow to me, especially when gauging its broader impact outside specialized use cases.
> >
> > 2. Intuitive Explanations: I must admit, as someone not deeply entrenched in meta-learning, I found it challenging to grasp the key concept behind continual meta-learning upon my initial reading. Your response was instrumental in clarifying this for me. However, I believe that others, especially non-experts, might face similar challenges. It would greatly benefit the paper to incorporate intuitive explanations and examples throughout, including the methods section (what each component is adjusting). Such additions would elucidate what each component does, aiding in a clearer and more intuitive understanding. Also, it would be beneficial for readers if a comparison of practical scenarios such as meta-learning, continual-learning, meta-continual-learning, and continual-meta-learning was presented, possibly in an appendix.
> >
> > I believe that addressing this concern in future iterations or research might provide a more compelling case for the broader use and significance of CML. Once again, I appreciate the hard work and dedication you've put into this research and the effort to clarify doubts.

---

> > > ### Author Response · Authors · 2023-08-21
> > >
> > > Thank you for acknowledging our response and efforts. We sincerely appreciate your valuable suggestions on incorporating intuitive explanations. We will add these in the revised version to improve the paper.

---

### Official Review · Reviewer_9fyt · 2023-06-27

**Soundness:** 2 fair
**Presentation:** 2 fair
**Contribution:** 2 fair
**Rating:** 6
**Confidence:** 3

**Summary:**

The paper considers a continual meta-learning setting with both static and dynamically changing environments and seeks to study the trade-off between learning and forgetting. It derives an upper bound on the excess risk, i.e. the expected gap between the true risk of the predictor and the optimal true risk and uses the average excess risk across tasks as a meta-objective to develop an algorithm that can meta-learn online. The resulting algorithm is empirically evaluated on a synthetic task and a subset of the OSAKA continual meta-learning benchmark.

**Strengths:**

The paper derives novel upper bounds on the average excess risk across a number of methods.

**Weaknesses:**

- Despite its framing, the general approach of using a bound to derive a meta-objective is not completely novel and related work should be more prominently referenced (e.g. [1]).

- The method introduces many hyperparameters and it is unclear where improvements result from. An empirical verification of the theory in the considered settings is missing and ablations on different components of the algorithm that are not necessarily prescribed by the theory (e.g. annealing $\gamma_t$) are missing.

- The presentation of the paper needs to be improved by better motivating and contextualizing theorems and ensuring that all symbols are defined in the main text.

[1] Amit, Ron, and Ron Meir. "Meta-learning by adjusting priors based on extended PAC-Bayes theory." International Conference on Machine Learning. PMLR, 2018.

**Questions:**

1. Could you please comment to what extent the assumption of $\alpha$-quadratic-growth in Theorem 3.1 is justified in the context of the loss of neural networks and the settings considered empirically?
2. In Figure 2b is the plotted AER computed from the equation in Theorem 5.1 or is it measured independently? I am asking to understand whether this entails an empirical verification of the theory or a visualisation of it.
3. In Figure 2b BOCD only detects few change points and seems to do so mostly incorrectly. Could you clarify where your algorithm uses these? The plot over the "meta lr" suggests that the ground truth change points are used and neither "oracle" nor "window" rely on empirically detected change points.
4. Why does "window" perform better than "oracle" in Figure 3? Would we not expect the opposite?
5. In line 358-359 you state "MAML and ANIL do not suffer forgetting since they never update meta-parameters." In light of this, why does their performance vary over time in Figure 4? Does this also hold true for Figure 3? If so have they been pre-trained for the Moving 2D Gaussian task?
6. In Figure 4, why is there already such a clear gap between algorithms at $t=0$?
7. How are the hyperparameters tuned? Specifically how have they been tuned for baselines for the modified OSAKA tasks? Could you please report these details?


**Limitations:**

Limitations have been addressed appropriately.

---

> ### Author Rebuttal · Authors · 2023-08-09
>
> We thank the reviewer for the insightful comments on the experimental details.  Below are our responses.
>
> > Weakness 1. Novelty and citation
>
> - We appreciate the suggestion for a more prominent citation of [1], which used a bound to derive the meta-objective for ***statistical meta-learning***. Due to the page limit, we have briefly discussed [1] in the appendix, which will be moved to the main paper in the revised version.
> - In contrast to [1], we focus on the more challenging problem of learning from **non-i.i.d. sequential** tasks originating from a **possiblely shifting** environment. [1] assumes a **static** task environment, and the algorithm derived from the meta-objective requires a joint training process (simultaneously updating the meta-parameter and task parameters), which cannot be extended to the **sequential** learning scenario.
> - Moreover, we want to highlight our theoretical contribution of the first formal analysis of the **bi-level trade-off** for CML in shifting environments, which provides a better understanding of the non-trivial online statistical mixture learning problem.
>
> > Weakness 2. Unclarities
>
> * Thanks for mentioning these points. We will add corresponding discussions in the revised version.
>   * Due to the formulation of a unified theory, it seems that we have introduced many hyper-parameters. However, by setting the base learner as SGD, we only need to tune the hyper-parameters shown in Table 2 (**rebuttal file**), no more than the previous methods.
>   * The improvements come from the adaptive updates of the meta-parameter and its learning rate that considers the sample number and the environment change, providing a better forgetting-generalization trade-off, as discussed in Sec.4 and 6.
>   * Annealing $\gamma_t$ is theoretically grounded with online convex optimization and proved in Theorem 5.1, where the theoretical optimal value of $\rho$ and $\gamma_0$ are presented in the proof. In the experiments, we directly tune $\rho$ and use the theoretical optimal value for $\gamma_0$, which is tuned with other hyper-parameters like the slot diameter $\hat{D}$.
>
> > Weakness 3. Presentation
>
> * Thanks for the suggestion. We will add a notation table for better presentation.
>
> > Q1. $\alpha$-quadratic-growth
>
> * As discussed in Appendix E.3.1, we have $\alpha=2$ for a moving 2D Gaussian since the square loss is 2-strongly convex. The lasso, sparse logic regression also satisfies the quadratic-growth condition. Whether the condition can be satisfied for deep learning depends on the loss function and the hypothesis set $\mathcal{W}$. However, tuning $\alpha$ as a hyper-parameter can achieve satisfactory results.
>
> >Q2. Figure 2b, AER
>
> * In Figure 2(b), we have empirically verified the inequality in Theorem 5.1 by estimating both sides for SGD.
>   - For the left-hand side, we have $AER_{\mathcal{A}}^T = \frac{1}{T}\sum\_{t=1}^T R_{excess}(\mathcal{A}, u_t) ≈ \frac{1}{T} \sum\_{t=1}^T \mathcal{L}\_{\mu_t}(w_t) - \frac{1}{T}\sum_{t=1}^T\mathcal{L}\_{\mu_t}(w_t^*)$, where the first term is estimated by the ***average test loss*** (the first subplot). In addition, we have $X\sim \mathcal{N}(w_t^*, 0.1 I_2)$. Hence, $\mathcal{L}_{\mu_t}(w_t^*) = \min_w \mathbb{E}_X \|\|w -X\|\|^2 = \mathbb{E}_X[\|\|\mathbb{E}[X] - X\|\|^2] = 0.2$.
>   - The right-hand side (the AER bound) was computed by empirically estimating the slot diameter (of the base learner outputs) $D_n$, the slot variance $V_n$, and the path length.
>
> >Q3. BOCD
>
> * Sorry for the confusion caused. In Figure 2b, we did not use BOCD. We used BOCD and compared different detection methods in Figure 6 in Appendix E.2.1.
>
> > Q4. Window better than oracle
>
> * This result is related to the algorithm, which does not necessarily hold for other methods that maintain multiple meta-models. By updating a **single meta-model online**, the context switch cost exists (Line 373) for reconstructing the meta-knowledge in each slot. Setting a fixed window size can ensure the meta-knowledge quality of each slot and yield good performance on average.
> * In addition, the phenomenon is related to the **overlap between the consecutive environment distributions** and the **environment shift probability $p$**. We empirically tested these factors, and the result is presented in Figure A.2 (**rebuttal file**).
>   - In Figure A.2, "oracle" is better than "window" when $p < 0.1$, and the gap becomes more evident when the distribution overlap is smaller.
>
> > Q5. MAML, ANIL
>
> * Figure 4. illustrates the *running performance* for **each environment**.
>   - Given a fixed meta-parameter $u$, the average performance inside each environment is a proxy of the expectation, e.g., $\frac{1}{T} \sum_{t=1}^T \mathcal{L}\_{S_t}(w_t) ≈ \mathbb{E}\_{\mu \sim \tau_{mnist}} \mathbb{E}\_{S\sim \mu}\mathcal{L}\_{\mu}(\mathcal{A}(u, S))$.
>   - In the figure, MAML and ANIL only slightly vary at the early stage where $T$ is small. The variation decreases as $T$ grows according to the law of large numbers.
> * This does not hold for Figure 3 since it reports the *final performance* w.r.t different $p$ on the **overall environment $\tau$**, where $\tau$ changes with $p$.
> * We did not use a pre-trained model for the Moving 2D Gaussian, since the data is very simple.
>
> > Q6. Why clear gap at $t=0$?
>
> - Figure 4. compares the average performances in **each environment** over ten episodes.
> - The initial **learning rates of the base learner** of compared methods are different since they are tuned to achieve their best performance for the **overall environment**.
> - Hence, as observed, there is a gap at t=0 though the meta-model is the same.
>
> > Q7. Hyper-parameter tuning
>
> * Following OSAKA, the hyper-parameters were tuned by random search, and the same number of trials were allocated for each algorithm.
> * For each trial, we sampled hyper-parameter combinations uniformly from the search space presented in Table 1 for baselines and in Table 2 for DCML (see the **rebuttal file**).

---

> > ### Comment · Reviewer_9fyt · 2023-08-15
> >
> > Thank you for your detailed response that has helped clarify my open questions. I am raising my score accordingly.
> >
> > One more follow-up question regarding the hyperparameter search space: Is there a reason why you allow your own method a larger batch size of 50 while restricting baselines to small batch sizes (1,2,4,8,16)? If not I would suggest including batch size 50 for your baselines HP grid to ensure this is not the reason for performance differences.

---

> > > ### Author Response · Authors · 2023-08-15
> > > **Response**
> > >
> > > We appreciate your responses and raising the scores.
> > >
> > > W.r.t the follow-up question, thanks for the careful check, and sorry for the confusion made. The two batch sizes in the two tables have different meanings.
> > > Table 1 is adapted from Table 6 in the Appendix of OSAKA, where the batch size is the *number of tasks in the outer batch*, which is only needed for *pre-training*. Since we use the same pre-train model, the final setting of this hyper-parameter is the same for all the methods.
> > >
> > > The batch size in Table 2 is the *inner batch size* that we introduced to implement mini-batch SGD. However, to make a fair comparison with the baselines, whose inner-batch size is hard-coded as all the data samples of each few-shot task, we set it to $50$ for OMF data (5-shot, 10-way).
> > >
> > > In the experiments, all the methods use the same inner batch size, which is decided by the setting for each dataset (n_shots, n_ways). Hence, this hyper-parameter can be removed from the hyper-parameter search space.

---

### Official Review · Reviewer_iCpz · 2023-07-06

**Soundness:** 3 good
**Presentation:** 3 good
**Contribution:** 3 good
**Rating:** 7
**Confidence:** 4

**Summary:**

This paper theoretically studies the stability and plasticity dilemma in Continual Meta-Learning (CML).

This paper formulates the CML objective as selecting u_t at each time t to ensure a small Average Excess Risk (AER) upper bound for the task sequence. Based on the AER objective, it proposes a continual meta-learning framework in both static and shifting environments.

The proposed Dynamic Continual Meta-Learning (DCML) algorithm can quickly reconstruct meta-knowledge to alleviate forgetting and fast adapt to new environments when change occurs.

The corresponding theory provides tighter bounds and more flexible base learner selections.

Experiments demonstrate the superiority of the proposed method.

**Strengths:**

This paper first introduces a novel and unified theoretical framework for CML in both static and shifting task environments.
Adaptively learning the meta-parameter can facilitate the training process in deep learning.

The paper is written and organized well.

The source code is provided.

The paper has good and relevant citations.

The results are quite extensive and convincing.

**Weaknesses:**

The paper lacks a clear figure that contextualizes the base model, learner, and how continual meta-learning fits into the setup. This visual representation would enhance understanding for readers.



**Questions:**

Can you discuss what applications this paper will have to memory-based approaches and what theoretical explorations can be done in the future?

**Limitations:**

The proposed theory works for non-convex loss when using Gibbs algorithm. However, this paper does not offer an analysis for SGD with non-convex loss.

This paper does not provide theoretical guarantees for memory-based approaches, which can better address the meta-level forgetting with additional memory cost.

---

> ### Author Rebuttal · Authors · 2023-08-08
>
>
> We sincerely thank the reviewer for appreciating the strength of our work and the constructive suggestions. Below are our responses to your comments.
>
>
> > The paper lacks a clear figure that contextualizes the base model, learner, and how continual meta-learning fits into the setup. This visual representation would enhance understanding for readers.
> * Thanks for the helpful suggestion. We have ***added a figure for the corresponding visualization in the rebuttal file***. Please refer to ***Figure A.1***.
>
> * At each time t, $\mathcal{A}\_{\text{CML}}$ takes the knowledge $u_t$ learned from previous tasks and current task data $S_t$ as input, then outputs the learned hypothesis $W_t$ for the current task and the updated prior knowledge $u_{t+1}$ for the next task. The task environment is static if $\tau_t = \tau, \forall t \in [T]$.
>
> * The whole CML algorithm $\mathcal{A}\_{\text{CML}}$ is a composition of the meta learner and base learner. In the paper , we have $\mathcal{A}\_{\text{base}} \in \\{\text{SGD, SGLD, Gibbs, RLM}\\}$. However, the illustrated continual meta-learning process is valid for other algorithm choices.
>
>
>
> > Can you discuss what applications this paper will have to memory-based approaches and what theoretical explorations can be done in the future?
>
> * The DCML algorithm proposed in the paper maintains only one meta-model. The primary idea for incorporating memory-based approaches is to maintain multiple meta-models and choose the closest meta-model or the weighted majority one as initialization for each task. We can derive regret bounds for this setting.
> * However, the challenge lies in deciding whether to grow a new meta-model or to update previous meta-models when an environment shift is detected within the predefined memory budget. Providing the corresponding theoretical guarantees for this would be valuable.
> * Another exciting direction for theoretical exploration is to quantify the amount of memory needed for successful continual meta-learning as those conducted for continual learning in [1,2].
>
> [1]  Jeremias Knoblauch, Hisham Husain, and Tom Diethe. Optimal
> continual learning has perfect memory and is NP-hard. In
> International Conference on Machine Learning, pages 5327–
> 5337, 2020.
>
> [2] Chen, Xi, Christos Papadimitriou, and Binghui Peng. "Memory bounds for continual learning." 2022 IEEE 63rd Annual Symposium on Foundations of Computer Science (FOCS). IEEE, 2022.

---

### Official Review · Reviewer_Rxnp · 2023-07-06

**Soundness:** 3 good
**Presentation:** 3 good
**Contribution:** 3 good
**Rating:** 7
**Confidence:** 3

**Summary:**

The authors present a framework for continual meta-learning, that is the training of meta-learning models through exposure to a sequence of tasks. The paper presents a formalism for the setting which allows for non-stationary task distributions.
Theoretical analysis is provided both for the framework and for the proposed method, and experimental validation is reported for a reasonable set of continual learning tasks.


**Strengths:**


- The problem setting is interesting, and the work is thorough.
- The experimental verification is convincing.


**Weaknesses:**


- I am not sure if the comparison with non-continual meta-learning method is fair.


**Questions:**


N/A

---

> ### Author Rebuttal · Authors · 2023-08-08
>
> We sincerely thank the reviewer for appreciating our work and the constructive comments. Below are our responses.
>
> > "...if comparison with non-continual meta-learning method is fair."
>
> * Thanks for mentioning this point. Due to the page limit, the detailed description for the baselines was not included in the main paper, and was deferred to Appendix E.1.1. We will move these discussions to the main paper in a revised version.
>
>   - Following OSAKA , we compared the proposed algorithm with different baselines. For a ***fair comparison***, we used the *same meta-model* pre-trained with MAML for all the methods. During the continual meta-learning phase, all the methods adapt with few-shot data and are then evaluated on separate test data for each task.
>     - *ANIL* is a variation of *MAML* that only adapts the network's head w.r.t new tasks. Following OSAKA, MAML, and ANIL, we *do not update* the meta-model during the CML phase.
>     - *CMAML* includes an update modulation phase and a prolonged adaptation phase, which updates the meta-model with buffered tasks when the environment changes or a task switch is detected. Otherwise, it keeps updating w.r.t the previous model.
>     - *MetaBGD and MetaCOG* perform CML based on MAML and Bayesian Gradient Descent (BGD), where *BGD* considers randomized parameters with factorized Gaussian, slowly updating the parameters of small variance to alleviate catastrophic forgetting. In addition, Meta-COG introduced a per-parameter mask.
>     - *Fine-tuning* uses the meta-model as initialization and consistently updates this model w.r.t the tasks encountered.
>   - As we can see, the **non-continual meta-learning** methods of the aforementioned baselines are MAML, ANIL, and Fine-tuning.
>     - MAML and ANIL do not update the meta-knowledge (so they will not suffer from forgetting) but lack the plasticity to learn from tasks in new environments. They are compared to show the problem with static representations in shifting environments.
>     - Fine-tuning is a trivial method that continually learns from data, which can be considered a lower bound in the setting to illustrate catastrophic forgetting.

---

### Official Review · Reviewer_JcdX · 2023-07-07

**Soundness:** 3 good
**Presentation:** 3 good
**Contribution:** 3 good
**Rating:** 5
**Confidence:** 2

**Summary:**

The paper presents a comprehensive theoretical framework for continuous meta-learning in both static and shifting task environments. It also conducts a theoretical analysis of the bi-level trade-off in shifting environments. Furthermore, the paper proposes a method to adapt meta-parameters and the corresponding learning rate.

**Strengths:**

1. The main contribution of the paper is the theoretical analysis of continual meta-learning, particularly in the shifting task environment section, which illustrates the concept of meta-level forgetting.
2. The paper provides theoretical evidence of the task-level trade-off and meta-level trade-off in continual meta-learning, offering valuable insights for optimizing the sequence of meta-parameters.
3. The paper introduces a novel AER objective and presents the DCML algorithm, which effectively mitigates forgetting and accelerates adaptation to new environments.


**Weaknesses:**

1. In my opinion, both continual meta-learning and continual learning share the common objective of addressing the stability-plasticity dilemma, albeit in different settings.  What is the key difference between them? And are there any real applications where the settings of continual meta-learning are more suitable?
2. It is unclear about the training efficiency compared to other methods.


**Questions:**

1. See Weakness 1
2. To better evaluate this method, it would be valuable to provide a comparison of the computation costs with other existing methods.


**Limitations:**

adequately addressed

---

> ### Author Rebuttal · Authors · 2023-08-08
>
> We sincerely appreciate the reviewer's efforts to provide constructive comments which we can incorporate in the revision. Below are our responses to your comments.
>
> > Q1. "Both continual meta-learning and continual learning share the common objective," "What is the key difference between them?"
>
> * Yes, the two settings share the common objective of addressing the stability-plasticity dilemma, which is the typical objective for learning from non-stationary data.
>
> * Their key differences are:
>   - Continual Learning (CL)
>     - Standard CL methods sequentially adapt a ***task model*** and aim for a final model that performs well on all the tasks encountered. (Some memory-based approaches like dynamic architecture grow a subnet for each ***task***.)
>     - The current task model is adapted from the ***previous task model***.
>     - Standard CL reports the ***final performance*** of the agent on all the tasks at the end of its lifetime.
>     - Standard CL approaches mainly focus on minimizing catastrophic forgetting without considering quick generalization for tasks with few-shot data, e.g., the replay-based and regularization-based methods.
>   - Continual Meta-Learning (CML)
>     - CML methods sequentially adapt a ***meta-model*** and aim to recover the performance on previous tasks by adapting the meta-model with few additional samples. Therefore, it's more suitable for learning ***few-shot*** tasks. (Some memory-based approaches grow a subnet for each ***environment***.)
>     - The current task model is adapted from the ***meta-model***.
>     - CML reports the ***cumulative performance*** of the agent throughout its lifetime.
>     - CML focuses on quick generalization for new tasks and fast recovering performance of previous tasks with few-shot data.
>
> > Q2. "And are there any real applications where the settings of continual meta-learning are more suitable?"
>
> * Yes. Maintaining a separate meta-model provides more plasticity to quickly generalize, which is more suitable for some online systems using few-shot labeled data or real-time feedback.
> * A representative real-world application is the online recommendation system, where we aim to predict each user's preference for various products.
>   - Different users' preferences are considered as different task distributions, and the distribution over users can be regarded as a task environment.
>   - The recommendation system maintains a meta-model that predicts the recent preferences shared across users and adapts task-specific models that predict the user-specific preferences.
>
> * The queries for displaying products of each user arrive sequentially, and the system randomly distributes several (e.g., $m_t = 200$) products in response to the query. Denote the products and the corresponding preferences of the $t$-th user as $(X_i, Y_i) \sim \mu_t, \forall i \in [m_t]$, which are i.i.d.. The labels are initially unknown for this setting, but few labels can be acquired through the following interaction:
>   - The meta-model predicts the preferences over products and displays them according to the predicted preference order.
>   - Normally, the user will only view a small part ($m_t^\prime=50$) of the distributed products, and the preference can be determined with the clicks or view time over these products, which gives the few-shot labeled samples.
>
> * Then, the task-specific model is adapted from the meta-model with these examples. The hope is that the model can generalize on the rest of unseen examples with a correct preference prediction, which can provide new recommendations.
>
> * Finally, the task-specific model of each user is used for updating the meta-model. The task environment is shifting since the shared preferences change with the season, fashion trends, and accidental events.
>
> > Q3. "Computation costs compared to other methods."
>
> * The computational complexities of all the methods are provided in the following table, where the run time is measured in terms of the number of gradient computations. All the methods are assumed to use a $K$-step SGD as the inner algorithm.
>   - Fine-tuning uses the meta-model as initialization and consistently updates this model w.r.t the tasks encountered. MAML and ANIL do not update the meta-model during the learning phase. Therefore, these methods need to compute $T*K$ gradients for the $T$ encountered tasks.
>   - MetaBGD and MetaCOG [8] perform CML based on MAML and use Bayesian Gradient Descent (BGD) for meta-model adaptation, which requires $M$ Monte Carlo samplings for the meta-gradient computation. Hence, the computation complexity is $\mathcal{O}(T*(K+M))$.
>   - DCML and CMAML need one meta-gradient and $K$ inner gradients computations for each task, so the complexity is $\mathcal{O}(T*(K+1))$.
>
>
>
> | Methods | Computational complexity |
> | --- | --- |
> | Fine-tuning | $\mathcal{O}(T*K)$ |
> | MetaCOG | $\mathcal{O}(T*(K+M))$ |
> | MetaBGD | $\mathcal{O}(T*(K+M))$ |
> | ANIL | $\mathcal{O}(T*K)$ |
> | MAML | $\mathcal{O}(T*K)$ |
> | CMAML | $\mathcal{O}(T*(K + 1))$ |
> | DCML | $\mathcal{O}(T*(K + 1))$ |

---

### Author Rebuttal · Authors · 2023-08-09


We appreciate the diligent efforts of the reviewers in thoroughly reviewing our paper and providing insightful feedback. In response to the valuable comments, we have included supplementary figures and tables to better address the raised concerns.

---

### Decision · Program_Chairs · 2023-09-21

**Decision:**

Accept (poster)

**Comment:**

The paper presents a framework for continual meta-learning, a setting that trains meta-learning models via a sequence of non-stationary tasks. Theoretical analysis and a practical method is proposed for the framework. Experimental validation is reported for a reasonable set of continual learning tasks.

After the rebuttal, all reviewers are in favor of accepting the paper. Most concerns from the reviewers are sufficiently addressed (a minor outstanding issue is to improve the writing for a wider audience).